# Joint Model and Data Sparsification via the Marginal Likelihood

Alexander Timans[1]  Thomas Möllenhoff[2]  Christian A. Naesseth[1]
Mohammad Emtiyaz Khan[*,2]  Eric Nalisnick[*,3]

## Abstract

Sparse recovery in linear systems underpins applications from signal processing to high-dimensional regression. Sparse Bayesian Learning, grounded in the principle of automatic relevance determination (ARD), offers a practical Bayesian mechanism for feature sparsity via marginal likelihood optimization. Yet, its reliance on a homoscedastic noise model renders it sensitive to data contaminations such as outliers or misspecified noise, harming model fit and predictions. Instead, we propose jointly learning individual feature and sample relevancies, enabling simultaneous model and data sparsification via a single Bayesian objective. This symmetric pruning of model and data offers a natural extension that preserves conjugacy, admits closed-form updates for standard optimization procedures, and aligns with perspectives from robust regression and influence functions. Empirical results across diverse regression tasks affirm that a joint ARD approach consistently yields both sparse and robust prediction models.

## 1. Introduction

Recovering sparse representations in linear systems is a fundamental learning problem, with applications stretching from generic system resolution in ill-posed high-dimensional settings (Bühlmann & Van De Geer, 2011) to signal processing (Giacobello et al., 2012), compressed sensing (Ji et al., 2008; Chen et al., 2001), and imaging (Meer et al., 1991; Lustig et al., 2007). As such, a multitude of proposals from traditional regularization (Tibshirani, 1996) to greedy matching pursuit (Tropp & Gilbert, 2007) tackle the task of identifying a sparse and informa-

tive feature subset. In a Bayesian context, the principle of *automatic relevance determination* (ARD) leverages a data-driven, sparsity-inducing weight prior to suppress superfluous weights during model fitting (MacKay, 1992; 1995). Centered around marginal likelihood maximization, the approach is operationally also known as *sparse Bayesian learning* (SBL) (Tipping, 2001; Palmer et al., 2003).

Here, the canonical linear system is given as

$$\mathbf{y} = \mathbf{X}\boldsymbol{\theta} + \boldsymbol{\epsilon}, \tag{1}$$

where $\mathbf{y} \in \mathbb{R}^n$ denotes the target signal, $\mathbf{X} \in \mathbb{R}^{n \times d}$ a potentially overcomplete feature dictionary, $\boldsymbol{\theta} \in \mathbb{R}^d$ the unknown model weights, and $\epsilon_i \sim \mathcal{N}(0, \lambda_i)$ independent Gaussian noise. The ARD prior is then characterized by $\theta_j \sim \mathcal{N}(0, \gamma_j^{-1})$, with $\gamma_j$ the learned precision parameter of the $j$-th weight, dictating its concentration around zero. If $\gamma_j \to \infty$, the weight is driven to zero and its respective feature column in $\mathbf{X}$ is effectively pruned. We denote features as $\mathbf{X}$ for notational simplicity, but nonlinear mappings $\boldsymbol{\Phi}(\mathbf{X})$ are certainly permitted, and used throughout our experiments (§ 5).

The standard SBL formulation imposes *i.i.d.* Gaussian noise, corresponding to a homoscedastic noise model with scalar variance $\lambda$ shared across terms (*i.e.*, $\lambda_i = \lambda \, \forall i$). It can be estimated alongside weight precisions $\boldsymbol{\gamma} = (\gamma_1, \dots, \gamma_d)$, but is usually treated as a secondary nuisance parameter (Neal, 1996). As a result, the framework can be sensitive to potential data contamination through outliers, heavy tails, and varying dispersion, rendering the underlying noise model $p(\epsilon \mid \lambda)$ misspecified with significant potential to impact model fit and subsequent predictive performance (Rousseeuw & Leroy, 2003). Yet, approaches to equip Eq. 1 and related designs with more flexible noise modelling within a Bayesian framework tend to either *(i)* impose additional noise or model structure conditions, *(ii)* leverage 'robust' distributions that compromise Gaussian conjugacy and necessitate approximate inference, or *(iii)* do not integrate naturally into the SBL *modus operandi* of evidence maximization.

Instead, we propose a straightforward extension to the ARD principle: the inclusion of per-sample noise variances $\lambda_i$ as primary parameters. With a drop-in replacement of $\lambda$

*Equal contribution [1]UvA-Bosch Delta Lab, University of Amsterdam [2]RIKEN Center for AI Project, Tokyo, Japan [3]Department of Computer Science, Johns Hopkins University. Correspondence to: Alexander Timans <a.r.timans@uva.nl>.

*Proceedings of the 43rd International Conference on Machine Learning*, Seoul, South Korea. PMLR 306, 2026. Copyright 2026 by the author(s).

*Figure 1.* An illustration of the proposed sparsification approach for polynomial regression on the *mcycle* dataset (Silverman, 1985) ($d = 12$, $n = 128$). *Left:* Model-only sparsification identifies two prunable features (marked in red), but treats every sample as equally relevant for model fit, including six known outliers (●). *Right:* Joint learning of per-weight ($\boldsymbol{\gamma}$) and per-sample ($\boldsymbol{\lambda}$) parameters via the marginal likelihood not only sparsifies features akin to model-only ARD, but additionally identifies and automatically downweighs noisy samples. This permits data diagnostics and leads to better posterior predictive fit (here, for 10-fold cross-validation with EM, an RMSE of 23.6 *vs.* 24.6 and NLL of 4.61 *vs.* 4.67).

by $\boldsymbol{\lambda} = (\lambda_1, \ldots, \lambda_n)$ we optimize $(\boldsymbol{\gamma}, \boldsymbol{\lambda})$ simultaneously to identify both shrunken weights via $\gamma_j$ and noisy samples for which $\lambda_i$ is large. This hinges on the intuitive framing of model robustification as a data *sparsification* task (Jin & Rao, 2010), and ensures the automatic downweighting of samples deemed uninformative or counterproductive during model fitting (see Fig. 1 for an example). By viewing the model and data as complementary contributions to the marginal likelihood, we obtain a single unifying objective that naturally promotes both sparsity and robustness. We demonstrate consistent improvements over homoscedastic baselines across different regression problems, and link our approach to existing robust regression techniques. In summary, our contributions include:

- Extending the ARD principle to heteroscedastic noise, with closed-form update rules for several well-known SBL optimization schemes (§ 3);

- Connecting the resulting robustification mechanism to established ideas in robust regression, influence functions, and Gaussian Processes (§ 4);

- Demonstrating consistent benefits in prediction and sparsification over homoscedastic baselines across a range of non-linear regression tasks (§ 5).

## 2. Background and Related Work

We begin by revisiting underlying concepts and prior work on SBL, as well as broader related sparsification and robustification (Tab. 1). More can be found in App. A. Regarding

notation, matrices $\mathbf{M}$, vectors $\mathbf{v}$ and scalars $s$ are distinguished by case and bolding.

**Sparse Bayesian Learning.** Denoting model and data parameters $\boldsymbol{\gamma}$ and $\boldsymbol{\lambda}$ within a two-tier Bayesian hierarchical model[1], the joint probability factorizes as

$$p(\mathbf{y}, \boldsymbol{\theta}, \boldsymbol{\gamma}, \boldsymbol{\lambda}) = p(\mathbf{y} \mid \boldsymbol{\theta}, \boldsymbol{\lambda}) \cdot p(\boldsymbol{\theta} \mid \boldsymbol{\gamma}) \cdot p(\boldsymbol{\gamma}) \cdot p(\boldsymbol{\lambda}),$$

with $p(\mathbf{y} \mid \boldsymbol{\theta}, \boldsymbol{\lambda})$ the likelihood function, $p(\boldsymbol{\theta} \mid \boldsymbol{\gamma})$ the ARD prior and $p(\boldsymbol{\gamma}), p(\boldsymbol{\lambda})$ denoting hyperpriors placed on the parameters of interest. By Bayes' rule, the posteriors over weights $\boldsymbol{\theta}$ and parameters $(\boldsymbol{\gamma}, \boldsymbol{\lambda})$ satisfy

$$p(\boldsymbol{\theta} \mid \mathbf{y}, \boldsymbol{\gamma}, \boldsymbol{\lambda}) \propto p(\mathbf{y} \mid \boldsymbol{\theta}, \boldsymbol{\lambda}) \cdot p(\boldsymbol{\theta} \mid \boldsymbol{\gamma}),$$
$$p(\boldsymbol{\gamma}, \boldsymbol{\lambda} \mid \mathbf{y}) \propto p(\mathbf{y} \mid \boldsymbol{\gamma}, \boldsymbol{\lambda}) \cdot p(\boldsymbol{\gamma}) \cdot p(\boldsymbol{\lambda}).$$

Employing uninformative (or flat) hyperpriors $p(\boldsymbol{\gamma}) \propto 1, p(\boldsymbol{\lambda}) \propto 1$ then indicates that $p(\boldsymbol{\gamma}, \boldsymbol{\lambda} \mid \mathbf{y}) \propto p(\mathbf{y} \mid \boldsymbol{\gamma}, \boldsymbol{\lambda})$, and hence a *maximum a posteriori* (MAP) parameter estimate can be obtained by maximizing the marginal likelihood instead (Tipping, 2001), given by

$$p(\mathbf{y} \mid \boldsymbol{\gamma}, \boldsymbol{\lambda}) = \int p(\mathbf{y} \mid \boldsymbol{\theta}, \boldsymbol{\lambda}) \, p(\boldsymbol{\theta} \mid \boldsymbol{\gamma}) \, d\boldsymbol{\theta}. \qquad (2)$$

This is optionally referred to as evidence maximization or Type-II maximum likelihood (Neal, 1996). Estimates of $(\boldsymbol{\gamma}, \boldsymbol{\lambda})$ can subsequently be plugged in to (iteratively) update the weight posterior $p(\boldsymbol{\theta} \mid \mathbf{y}, \boldsymbol{\gamma}, \boldsymbol{\lambda})$ and perform predictive inference on new observations.

---
[1] Where $\boldsymbol{\lambda} = (\lambda, \ldots, \lambda)$ in the homoscedastic case.

*Table 1.* Comparison of model and data (or joint) ARD to related concepts in the literature. We desire a learning approach to satisfy both model sparsity and robustness to contaminated data or misspecified noise.

| Property | Model & Data ARD | Model ARD, RVM | Sparse GPs | Bayesian Pruning | Robust GPs | Robust Regression |
|---|---|---|---|---|---|---|
| Sparsity | ✓ | ✓ | ✓ | ✓ | ✗ | ✗ |
| Robustness | ✓ | ✗ | ✗ | ✗ | ✓ | ✓ |

Under Gaussian conjugacy—as is the case for SBL—the term in Eq. 2 results in a closed-form objective (see § 3.1) but remains jointly non-convex in $(\gamma, \lambda)$, requiring iterative re-estimation procedures. Standard approaches include expectation maximization and MacKay's updates employing a fixed-point condition (MacKay, 1995), as well as a fruitful string of research connecting SBL to standard MAP estimation and iterative reweighted least squares (Wipf & Nagarajan, 2007; 2010; Wu & Wipf, 2012; Candes et al., 2008; Chartrand & Yin, 2008). Recent work has also explored alternative ARD priors (Giri & Rao, 2016; Zhou et al., 2021; Ray & Szabó, 2022) and surrogate objectives (Wang et al., 2024; Zhang et al., 2025).

**Relevance Vector Machine.** As a particular instantiation of SBL, the *relevance vector machine* (RVM) replaces the feature-based dictionary $\mathbf{X} \in \mathbb{R}^{n \times d}$ in Eq. 1 with a kernel matrix $\mathbf{\Phi} \in \mathbb{R}^{n \times n}$ centered around samples, such that $\mathbf{\Phi}_{*,j} = k(\mathbf{x}_*, \mathbf{x}_j)$ evaluates $\mathbf{x}_*$ at the $j$-th basis function (Tipping, 2001; Tipping & Faul, 2003). Thus, model ARD now corresponds to basis pruning and results in a sparse set of 'relevance vectors' dictating its functional form[2]. A correspondence can be drawn to sparsifying Gaussian Processes (GPs) in explicit weight-space form, further connecting to a broader literature on kernel learning and inducing points (Liu et al., 2020b; Quinonero-Candela & Rasmussen, 2005; Titsias, 2009). We also point out links to sparse recovery via stepwise regression and pursuit algorithms (Chen et al., 2001; Keerthi & Chu, 2005; Ament & Gomes, 2021).

**Robust RVM and GPs.** Extensions to more flexible noise models have primarily targeted the RVM by *(i)* introducing inlier and outlier noise components with separate parametrizations (Mitra et al., 2010; Sundin et al., 2015; Nannuru et al., 2019), or *(ii)* imposing a different functional structure on the model (as a mixture, Faul & Tipping (2001b)) and noise term (as a separate GP, Khashabi et al. (2013); Gerstoft et al. (2018)). This usually breaks conjugacy and requires variational approximations or tailored update rules.

Similar ideas on functional noise models can be found in the literature on robust GPs (Goldberg et al., 1997; Kersting et al., 2007; Lázaro-Gredilla & Titsias, 2011; Liu et al.,

[2]Akin to a Bayesian treatment of the well-known Support Vector Machine (Cortes & Vapnik, 1995).

2020a). This includes the incorporation of heavier-tailed likelihoods such as mixtures, the Laplace, or Student-$t$ (Vanhatalo et al., 2009; Jylänki et al., 2011; Shah et al., 2014; Lindfors et al., 2020; Ament et al., 2024) and relies almost exclusively on approximate inference. In contrast, our symmetric treatment of noise parameters provides closed-form updates, requires no additional structural assumptions, and retains a simple model design. The approach is related via the RVM to robustifying Gaussian Processes in § 4.

**Robust regression.** Established approaches in robust regression include the Huber loss, M-estimation, trimmed least squares, LAD, or median-of-means (Rousseeuw & Leroy, 2003; Lugosi & Mendelson, 2019; Loh, 2024), with strong ties to data diagnostics via influence functions (Hampel et al., 2011; McWilliams et al., 2014). From a Bayesian perspective, early work on robust linear models includes O'Hagan (1979); West (1984); Geweke (1993), and may also be tied to influence (Berger et al., 1994). We draw upon such connections for our derived update rule in § 4. Recent work in the direction has centered on data pre-selection (Geppert et al., 2017; Campbell & Beronov, 2019), likelihood reweighting (Wang et al., 2017; Dewaskar et al., 2025), and neural network training (Stirn et al., 2023; Immer et al., 2023). We offer perspectives towards integrating SBL with neural networks in § 5.4 and § 6.

## 3. Joint Automatic Relevance Determination

We next state the explicit marginal likelihood objective as well as a more efficient dual formulation. We then discuss heteroscedastic noise updates and place them in the context of standard SBL optimization schemes.

### 3.1. The Marginal Likelihood Objective

Revisiting the notation in § 2, the ARD model prior $p(\theta \mid \gamma)$ and noise prior $p(\epsilon \mid \lambda)$ are instantiated as

$$p(\boldsymbol{\theta} \mid \boldsymbol{\gamma}) = \prod_{j=1}^{d} \mathcal{N}(0, \gamma_j^{-1}) = \mathcal{N}(\mathbf{0}, \Gamma^{-1})$$

$$p(\boldsymbol{\epsilon} \mid \boldsymbol{\lambda}) = \prod_{i=1}^{n} \mathcal{N}(0, \lambda_i) = \mathcal{N}(\mathbf{0}, \Lambda),$$

where $\Gamma = \text{diag}(\boldsymbol{\gamma})$ and $\Lambda = \text{diag}(\boldsymbol{\lambda})$, collapsing to $\Lambda = \lambda \mathbf{I}_n$ for the homoscedastic case. Subsequently the

likelihood function is given by $p(\mathbf{y} \mid \boldsymbol{\theta}, \boldsymbol{\lambda}) = \mathcal{N}(\mathbf{X}\boldsymbol{\theta}, \Lambda)$. Hyperpriors $p(\boldsymbol{\gamma}), p(\boldsymbol{\lambda})$ are chosen to be uninformative, for instance $\gamma_j \sim \text{Gamma}(a, b)$ and $\lambda_i \sim \text{InvGamma}(a, b)$ with $a, b \approx 0$. This choice serves conjugacy and induces Student-$t$ marginals on $\boldsymbol{\theta}$ and $\mathbf{y}$, theoretically motivating effective sparsification (Tipping, 2001).

Leveraging Gaussian conjugacy, the marginal likelihood (Eq. 2) is given as $p(\mathbf{y} \mid \boldsymbol{\gamma}, \boldsymbol{\lambda}) = \mathcal{N}(\mathbf{0}, \Sigma_{\mathbf{y}})$, and we can write out the objective explicitly as

$$
\begin{aligned}
\mathcal{L}(\boldsymbol{\gamma}, \boldsymbol{\lambda}) &= -\log p(\mathbf{y} \mid \boldsymbol{\gamma}, \boldsymbol{\lambda}) \\
&= \frac{n}{2}\log(2\pi) + \frac{1}{2}\log|\Sigma_{\mathbf{y}}| + \frac{1}{2}\mathbf{y}^\top \Sigma_{\mathbf{y}}^{-1}\mathbf{y} \quad (3) \\
&\equiv \log|\Sigma_{\mathbf{y}}| + \mathbf{y}^\top \Sigma_{\mathbf{y}}^{-1}\mathbf{y},
\end{aligned}
$$

where maximization of $p(\mathbf{y} \mid \boldsymbol{\gamma}, \boldsymbol{\lambda})$ is equivalent to minimization of its log-negative, and the final equivalence holds up to additive constants. $\Sigma_{\mathbf{y}} = \Lambda + \mathbf{X}\Gamma^{-1}\mathbf{X}^\top \in \mathbb{R}^{n \times n}$ models the data covariance, with $\mathbf{X}\Gamma^{-1}\mathbf{X}^\top$ capturing model-induced covariance in sample space and noise variance $\Lambda$ marking unreliable data points. The log-determinant $\log|\Sigma_{\mathbf{y}}|$ serves as a complexity penalty, reflecting an 'Occam's Razor' effect of the marginal likelihood (Rasmussen & Ghahramani, 2000).

For a given pair $(\boldsymbol{\gamma}, \boldsymbol{\lambda})$ the weight posterior can be computed as $p(\boldsymbol{\theta} \mid \mathbf{y}, \boldsymbol{\gamma}, \boldsymbol{\lambda}) = \mathcal{N}(\boldsymbol{\mu_\theta}, \Sigma_{\boldsymbol{\theta}})$, where

$$
\boldsymbol{\mu_\theta} = \Sigma_{\boldsymbol{\theta}}\,\mathbf{X}^\top \Lambda^{-1}\,\mathbf{y} \quad \text{and} \quad \Sigma_{\boldsymbol{\theta}} = (\Gamma + \mathbf{X}^\top \Lambda^{-1}\mathbf{X})^{-1}
$$

denote updated posterior mean and covariance. $\Sigma_{\boldsymbol{\theta}} \in \mathbb{R}^{d \times d}$ highlights the parameter's complementary roles, with $\mathbf{X}^\top \Lambda^{-1}\mathbf{X}$ downweighting noisy sample contributions while model precision $\Gamma$ shrinks weakly supported weight directions, enacting the joint ARD effects.

**The dual objective.** Evaluation of $\mathcal{L}(\boldsymbol{\gamma}, \boldsymbol{\lambda})$ requires an $n \times n$ solve in $\Sigma_{\mathbf{y}}$ and can be numerically sensitive when updates drive $\Gamma^{-1}$ to be large. Instead, leveraging standard matrix identities (see § B.1) an equivalent dual formulation can be given as

$$
\begin{aligned}
\tilde{\mathcal{L}}(\boldsymbol{\gamma}, \boldsymbol{\lambda}) \equiv\ & \log|\Sigma_{\boldsymbol{\theta}}^{-1}| - \log|\Gamma| + \log|\Lambda| \\
& + \left(\mathbf{y}^\top \Lambda^{-1}\mathbf{y} - \tilde{\mathbf{y}}^\top \Sigma_{\boldsymbol{\theta}}\,\tilde{\mathbf{y}}\right),
\end{aligned} \quad (4)
$$

where $\tilde{\mathbf{y}} = \mathbf{X}^\top \Lambda^{-1}\mathbf{y} \in \mathbb{R}^d$ projects the data into weight space (and similarly appears in the posterior mean $\boldsymbol{\mu_\theta}$). Efficiency is gained by solving for size $d \times d$ when $d \ll n$, while decomposition of $\log|\Sigma_{\mathbf{y}}|$ highlights individual contributions to complexity: $\log|\Sigma_{\boldsymbol{\theta}}^{-1}| - \log|\Gamma|$ penalizes data-induced precision increases relative to the prior, while $\log|\Lambda|$ discourages excessive noise fitting. The data term promotes model fit by aiming to close the gap between total and model-explained variance.

*Table 2.* A high-level summary of optimization behaviour across investigated SBL procedures.

| Method | Improvement Guarantee | Convergence | Cost |
|---|---|---|---|
| EM | Monotone in $\mathcal{L}(\boldsymbol{\gamma}, \boldsymbol{\lambda})$ | Stationary point | Low |
| MacKay | None, Heuristic | Fixed point | Low |
| $\ell_2$-IRLS | Monotone in surrogate | Local optimum | Medium |
| $\ell_1$-IRLS | Monotone in surrogate | Local optimum | High |
| Grad. | None, Optimizer-dependent | Stationary point | Variable |

**Computational complexity.** Heteroscedasticity introduces $\mathcal{O}(n)$ additional parameters, but per-iteration asymptotic cost is dominated by the same algebraic operations as for a scalar variance $\lambda$. In both cases each iteration forms and factorizes $\Sigma_{\mathbf{y}}$ or $\Sigma_{\boldsymbol{\theta}}$, thus leading costs remain $\mathcal{O}(n^3)$ or $\mathcal{O}(d^3)$ with Cholesky factorization. Only linear-time overhead is added for storing $\boldsymbol{\lambda}$ and operating on $\Lambda^{-1}$. In practice, wall-clock time can increase as the objective is less constrained, with additional degrees of freedom that may slow convergence.

### 3.2. Optimizing for Heteroscedastic Noise

Both $\mathcal{L}(\boldsymbol{\gamma}, \boldsymbol{\lambda})$ and $\tilde{\mathcal{L}}(\boldsymbol{\gamma}, \boldsymbol{\lambda})$ are jointly non-convex and coupled in $(\boldsymbol{\gamma}, \boldsymbol{\lambda})$, necessitating iterative re-estimation. For $\Lambda = \text{diag}(\boldsymbol{\lambda})$, we show that the same closed-form update for $\lambda_i$ emerges from three standard SBL procedures: expectation maximization (EM), MacKay's updates, and $\ell_2$-iterative reweighted least squares ($\ell_2$-IRLS). We also discuss $\ell_1$-IRLS and gradient updates.

We follow with a concise summary of key aspects of each procedure under heteroscedastic noise. Full derivations are deferred to App. B, a table of update rules to § B.7, and pseudocode to App. C; optimization properties are summarized in Tab. 2 above. Since our heteroscedastic extension instantiates from existing SBL procedures and does not alter their algorithmic form, well-known properties follow directly from the corresponding analyses in respective works, *e.g.* Wu (1983); Wipf & Nagarajan (2007); Faul & Tipping (2001a).

**Expectation Maximization.** Treating $\boldsymbol{\theta}$ as latent, the E-step sees computing the intermediate posterior $p(\boldsymbol{\theta} \mid \mathbf{y}, \boldsymbol{\gamma}^t, \boldsymbol{\lambda}^t)$ at current parameters, while maximization under that posterior (M-step) yields the updates

$$
\gamma_j^{t+1} \leftarrow \frac{1}{\mu_{\boldsymbol{\theta},j}^2 + [\Sigma_{\boldsymbol{\theta}}]_{jj}} \quad \text{and} \quad \lambda_i^{t+1} \leftarrow r_i^2 + \mathbf{x}_i^\top \Sigma_{\boldsymbol{\theta}}\,\mathbf{x}_i,
$$

where $\mu_{\boldsymbol{\theta},j}$ and $[\Sigma_{\boldsymbol{\theta}}]_{jj}$ denote the $j$-th (diagonal) entries, $r_i = (y_i - \mathbf{x}_i^\top \boldsymbol{\mu_\theta})$ is the $i$-th training residual, and $\mathbf{x}_i^\top$ indicates the $i$-th row of $\mathbf{X}$.

Both updates follow from exact posterior second moments and take on intuitive interpretations: $\gamma_j$ grows when both weight magnitude and posterior variance are small, while

$\lambda_i$ decomposes learned noise into data fit ($r_i^2$) and model uncertainty ($\mathbf{x}_i^\top \Sigma_{\boldsymbol{\theta}} \mathbf{x}_i$). We find that the squared residual term tends to dominate empirically. EM updates are relatively stable and efficient, with monotone improvements in $\mathcal{L}(\boldsymbol{\gamma}, \boldsymbol{\lambda})$[3]. For $\Lambda = \lambda \mathbf{I}_n$ the noise updates reduce to a scalar computation, and more generally simplify across all SBL methods.

**MacKay's updates.** MacKay's updates (MacKay, 1992; 1995) target $\mathcal{L}(\boldsymbol{\gamma}, \boldsymbol{\lambda})$ via fixed-point iterations derived from stationarity conditions, yielding a similar form for $\gamma_j^{t+1}$ and the same update for $\lambda_i^{t+1}$ as EM. Improvement assurances are traded for faster convergence and increased sensitivity, but we empirically observe broadly consistent behaviour with EM.

**$\ell_2$-Iterative Reweighted Least Squares.** Rather than directly minimizing $\mathcal{L}(\boldsymbol{\gamma}, \boldsymbol{\lambda})$, Wipf & Nagarajan (2007; 2010) suggest a majorization-minimization strategy on an upper-bounding surrogate objective using Fenchel conjugates. Minimizing the surrogate w.r.t. $\boldsymbol{\theta}$ at fixed $(\boldsymbol{\gamma}, \boldsymbol{\lambda})$ yields a weighted ridge-type subproblem

$$\boldsymbol{\theta}^{t+1} \leftarrow \arg\min_{\boldsymbol{\theta}} (\mathbf{y} - \mathbf{X}\boldsymbol{\theta})^\top \Lambda^{-1} (\mathbf{y} - \mathbf{X}\boldsymbol{\theta}) + \sum_{j=1}^d w_j \cdot \theta_j^2,$$

with ridge weights $w_j = \gamma_j$, clarifying the connection to IRLS and admitting a closed-form solution. Subsequent re-estimation of $\gamma_j^{t+1}$ and $\lambda_i^{t+1}$ yields updates consistent with the same stationarity conditions as EM and MacKay, up to algebraic rearrangements.

**$\ell_1$-Iterative Reweighted Least Squares.** Following a similar logic to the $\ell_2$ case, a tractable and separable upper-bounding surrogate induces $\ell_1$-regularization penalties on weights $\theta_j$ and residuals $r_i$ of the form

$$2 \sum_{j=1}^d w_j \cdot |\theta_j| \quad \text{and} \quad 2 \sum_{i=1}^n v_i \cdot |r_i|,$$

with weights given by $w_j = (\mathbf{x}_j^\top \Sigma_{\mathbf{y}}^{-1} \mathbf{x}_j)^{1/2}$ and $v_i = ([\Sigma_{\mathbf{y}}^{-1}]_{ii})^{1/2}$. The resulting double $\ell_1$-regularized subproblem for $\boldsymbol{\theta}^{t+1}$ is convex but no longer admits a closed-form solution since $|r_i|$ is non-separable in $\theta_j$, necessitating more expensive iterative convex solvers (*e.g.* ADMM). Exact updates for $\gamma_j^{t+1}$ and $\lambda_i^{t+1}$ are then obtained using $\boldsymbol{\theta}^{t+1}$ and map-back rules.

**Gradient-based updates.** Setting aside analytical tractability, $\mathcal{L}(\boldsymbol{\gamma}, \boldsymbol{\lambda})$ (or $\tilde{\mathcal{L}}(\boldsymbol{\gamma}, \boldsymbol{\lambda})$) can also be optimized directly via first-order methods on the objective, either

jointly in $(\boldsymbol{\gamma}, \boldsymbol{\lambda})$ or with alternating steps, yielding updates of the shape

$$\gamma_j^{t+1} \leftarrow \gamma_j^t - \eta_\gamma \frac{\partial \mathcal{L}}{\partial \gamma_j} \quad \text{and} \quad \lambda_i^{t+1} \leftarrow \lambda_i^t - \eta_\lambda \frac{\partial \mathcal{L}}{\partial \lambda_i}$$

with parameter-specific learning rates $\eta$. A $\log \gamma_j$ and $\log \lambda_i$ parametrization improves conditioning and enforces positivity, but learning sensitivities can result in inferior fixed points when compared to closed-form.

# 4. Interpreting Data Relevance

We next provide two complementary interpretations of the resulting noise updates, connecting them to data influence and to GP robustness.

**As an influence-based diagnostic.** Influence is commonly understood as a combination of *outlierness* in target space $\mathbf{y}$ and *leverage* in feature space $\mathbf{X}$, jointly determining a sample's impact on the fitted model (Chatterjee & Hadi, 1986). We show that the derived heteroscedastic noise update $\lambda_i^{t+1} = r_i^2 + \mathbf{x}_i^\top \Sigma_{\boldsymbol{\theta}} \mathbf{x}_i$ naturally admits such an influence-based interpretation on model fit, directly relating it to leave-one-out (LOO) residual diagnostics.

Writing the model's fitted mean as $\hat{\mathbf{y}} = \mathbf{X}\boldsymbol{\mu}_{\boldsymbol{\theta}} = \mathbf{H}\mathbf{y}$ in terms of the 'hat' or leverage matrix $\mathbf{H}$ (using the definition of $\boldsymbol{\mu}_{\boldsymbol{\theta}}$), we observe that per-sample leverage under the joint ARD posterior yields

$$h_i = \frac{1}{\lambda_i} \mathbf{x}_i^\top \Sigma_{\boldsymbol{\theta}} \mathbf{x}_i,$$

measuring how strongly sample $y_i$ affects its own fitted value $\hat{y}_i$ under fixed $(\boldsymbol{\gamma}, \boldsymbol{\lambda})$. Influence can be equated with the impact of a point's finite removal[4] (LOO, Cook & Weisberg (1980)), and since the EM noise update follows from $\lambda_i = \mathbb{E}_{p(\boldsymbol{\theta}|\cdot)}[r_i^2]$ (§ B.3) we compare to the matching LOO term $r_{-i,i}^2$. That is, how the $i$-th sample's removal affects its own $i$-th squared residual, a classical diagnostics known as PRESS (Allen, 1974).

Using a rank-one Sherman-Morrison update, the LOO posterior fit is given as $\hat{y}_{-i,i} = \mathbf{x}_i^\top \boldsymbol{\mu}_{-i,\boldsymbol{\theta}}$ and LOO squared residual follows as

$$r_{-i,i}^2 = (y_i - \hat{y}_{-i,i})^2 = \frac{r_i^2}{(1 - h_i)^2},$$

a standard identity for the linear case (Montgomery et al., 2021). It becomes apparent that $r_{-i,i}^2$ is amplified even for small in-sample residuals if $h_i$ is large. To highlight the shared structure with $\lambda_i$ we take a first-order expansion of $r_{-i,i}^2$ (valid for $|h_i| < 1$) yielding $r_{-i,i}^2 \approx r_i^2 + 2h_i r_i^2$. This

---

[3]Under the exact update; in practice we equip methods with damping and clipping for stability, see § D.1.

[4]Alternatively, the theory on infinitesimal perturbations via derivatives is also known as *sensitivity*.

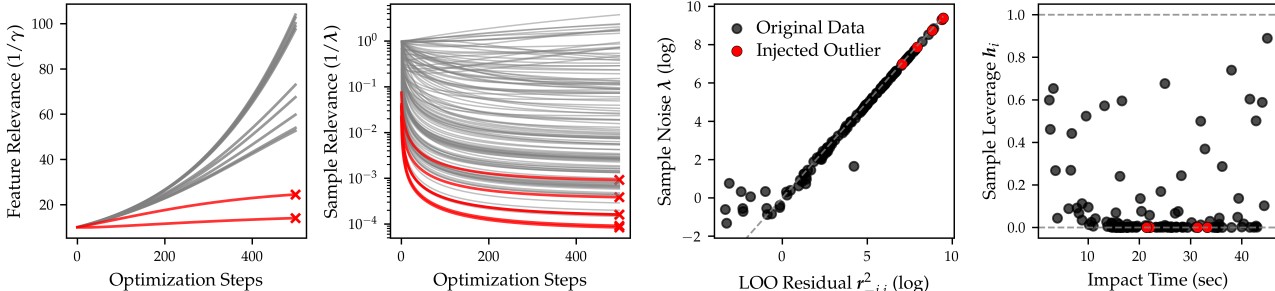

*Figure 2. Left:* The evolution of feature ($1/\gamma$) and sample relevance ($1/\lambda$) over the optimization trajectory for the *mcycle* experiment in Fig. 1. Prunable weights and injected outliers (red trajectories) are deemed superfluous to the model fit. *Right:* Aligning with the interpretation in § 4, learned noise parameters $\lambda_i$ strongly resemble leverage-aware LOO residual terms $r^2_{-i,i}$. Injected outliers record low sample leverage $h_i$ after parameter convergence, stressing their irrelevance for in-sample model fit as opposed to anchoring inlier points.

mirrors the EM update $\lambda_i^{t+1} = r_i^2 + \mathbf{x}_i^\top \Sigma_\theta \mathbf{x}_i = r_i^2 + \lambda_i^t h_i$, revealing $\lambda_i$ as an approximate, leverage-aware residual update. The correspondence is empirically corroborated in Fig. 2.

Importantly, $r^2_{-i,i}$ applies a *multiplicative* magnification $(1 - h_i)^{-2}$ whereas $\lambda_i$ contributes only an *additive* correction. Consequently, high-leverage outliers can remain insufficiently downweighted, an instance of classical residual masking that motivates 'studentized' update variants in future work (see a preliminary test in Fig. 6). The derived heteroscedastic updates we present here primarily respond to data outlierness.

**As a robust Gaussian Process.** Bridging back to the RVM as a particular instantiation of SBL, its design performs model sparsification in a kernel basis $\mathbf{\Phi} \in \mathbb{R}^{n \times n}$ (*i.e.* feature dimensionality $d = n$) and admits a direct connection to weight-space GPs (Tipping, 2001). Specifically, the ARD prior $\theta \sim \mathcal{N}(\mathbf{0}, \Gamma^{-1})$ and the RVM's functional model $f(\mathbf{x}) = \sum_{j=1}^n \theta_j \, k(\mathbf{x}, \mathbf{x}_j)$ together induce a GP prior with finite-rank kernel

$$k(\mathbf{x}, \mathbf{x}') = \sum_{j=1}^n \gamma_j^{-1} \, k(\mathbf{x}, \mathbf{x}_j) \, k(\mathbf{x}', \mathbf{x}_j),$$

and marginalizing $\theta$ yields the GP marginal likelihood $p(\mathbf{y} \mid \gamma, \lambda) = \mathcal{N}(\mathbf{0}, \Lambda + \mathbf{\Phi}\Gamma^{-1}\mathbf{\Phi}^\top)$. In this view, driving $\gamma_j \to \infty$ suppresses the $j$-th basis function (or column of $\mathbf{\Phi}$) in the induced kernel, promoting coefficient sparsity in the weight-space GP.

Crucially, model ARD via basis pruning alone may not suffice since each training datum plays a dual role: as an *observation* through its row in $\mathbf{\Phi}$ and as a *basis* through its column in $\mathbf{\Phi}$. Thus corrupted samples may still impact model fit via their residual and coupling in $\mathbf{\Phi}$ despite basis removal. By expanding $\Lambda = \text{diag}(\lambda)$ to heteroscedastic noise, data ARD can additionally downweight noisy observations directly, counteracting the effect. This yields a

weight-space GP interpretation that is both sparse and *robust*, while preserving conjugacy and closed-form learning.

Finally, we note conceptual distinctions from both *(i)* GP sparsity targeting computational scalability, such as inducing-point methods (Titsias, 2009), and *(ii)* feature sensitivity in GPs expressed via kernel-internal lengthscales rather than basis-function sparsity.

## 5. Empirical Results

We now present our suite of experimental findings, covering synthetic (§ 5.1), tabular (§ 5.2), kernel (§ 5.3) and neural network (§ 5.4) regression tasks. Further protocol details and results are reported in App. D and App. E. Our code is made publicly available at https://github.com/alextimans/robust-sbl.

**Experimental design.** To demonstrate the robustness benefits of data ARD we consider settings where (real-world) inlier data is subject to contamination in target space, perturbing a small fraction of samples to yield plausible response outliers. Model fit is assessed primarily through downstream hold-out predictive performance, using *root mean squared error* (RMSE) as a comparable point-estimate metric and Gaussian *negative log-likelihood* (NLL) for probabilistic predictions. For the NLL we evaluate posterior predictive mean and variance at test points $(\mathbf{x}_*, \mathbf{y}_*)$ as $\mu_* = \mathbf{x}_*^\top \mu_\theta$ and $\lambda_* = \lambda_b + \mathbf{x}_*^\top \Sigma_\theta \mathbf{x}_*$, which requires a base noise level $\lambda_b$. We conservatively set $\lambda_b = \text{mean}(\lambda)$ as the plug-in average training noise learned by data ARD.

To quantify both model and data sparsity without hard thresholding or explicit pruning we report a threshold-free *effective support size* (ESS), a scalar in $(0, 1]$ (or 0–100%) that summarizes the effective fraction of active elements, those being either high-relevance weights (ESS($\theta$)) or high-relevance samples (ESS($\mathbf{y}$)). Concretely, given nonnegative relevance scores ($1/\gamma_j$ or $1/\lambda_i$) we normalize to a probability mass and compute the exponentiated Shannon entropy

*Table 3.* Comparison of heteroscedastic SBL methods against sparse (Ridge, GP) and robust (Student-$t$, Huber) baselines on predictive RMSE and effective support sizes (weights $\boldsymbol{\theta}$ and samples $\mathbf{y}$). Improvements over homoscedastic SBL counterparts are shown as $(-x\%)$, indicating the relative reduction in RMSE from robustness. Results are shown on three tabular UCI regression tasks with 10% outlier contamination (avg. over 10 trials, $\pm 1\sigma$); lowest RMSE in **bold**.

| | Energy | | | Carbon | | | Protein | | |
|---|---|---|---|---|---|---|---|---|---|
| **Method** | RMSE ($\downarrow$) | ESS($\boldsymbol{\theta}$) | ESS($\mathbf{y}$) | RMSE ($\downarrow$) | ESS($\boldsymbol{\theta}$) | ESS($\mathbf{y}$) | RMSE ($\downarrow$) | ESS($\boldsymbol{\theta}$) | ESS($\mathbf{y}$) |
| OLS | $5.73 \pm 0.71$ | 100 | 100 | $0.118 \pm 0.006$ | 100 | 100 | $2932.8 \pm 412.3$ | 100 | 100 |
| Ridge | $2.79 \pm 0.31$ | 96.0 | 100 | $0.053 \pm 0.004$ | 92.6 | 100 | $957.2 \pm 96.0$ | 90.9 | 100 |
| GP | $2.85 \pm 0.29$ | 9.2 | 100 | $0.053 \pm 0.005$ | 16.9 | 100 | $880.5 \pm 70.2$ | 27.7 | 100 |
| Student-$t$ | $2.23 \pm 0.17$ | 100 | 92.5 | $0.028 \pm 0.003$ | 100 | 90.2 | $1060.3 \pm 134.8$ | 100 | 89.7 |
| Huber | $2.14 \pm 0.23$ | 100 | 83.5 | $0.015 \pm 0.004$ | 100 | 90.1 | $653.8 \pm 140.6$ | 100 | 88.5 |
| EM | $\mathbf{1.89 \pm 0.14}$ $(-32.07\%)$ | 29.4 | 91.2 | $0.015 \pm 0.004$ $(-69.91\%)$ | 15.3 | 92.0 | $525.8 \pm 116.3$ $(-44.95\%)$ | 11.7 | 91.7 |
| MacKay | $1.97 \pm 0.16$ $(-29.70\%)$ | 6.7 | 88.4 | $0.016 \pm 0.004$ $(-67.93\%)$ | 2.7 | 90.8 | $\mathbf{520.6 \pm 129.6}$ $(-45.12\%)$ | 5.2 | 90.2 |
| $\ell_2$-IRLS | $\mathbf{1.89 \pm 0.15}$ $(-31.89\%)$ | 39.5 | 91.0 | $0.014 \pm 0.004$ $(-70.34\%)$ | 22.7 | 91.5 | $524.0 \pm 115.4$ $(-44.91\%)$ | 19.0 | 91.5 |
| $\ell_1$-IRLS | $2.48 \pm 0.54$ $(-25.46\%)$ | 18.5 | 95.4 | $0.020 \pm 0.003$ $(-71.53\%)$ | 1.9 | 93.1 | $601.6 \pm 103.5$ $(-60.88\%)$ | 7.3 | 92.5 |
| Grad. (Primal) | $2.28 \pm 0.27$ $(-18.52\%)$ | 11.8 | 64.4 | $\mathbf{0.013 \pm 0.004}$ $(-72.62\%)$ | 4.2 | 89.6 | $565.1 \pm 129.8$ $(-40.81\%)$ | 9.8 | 87.1 |
| Grad. (Dual) | $2.28 \pm 0.27$ $(-18.42\%)$ | 11.1 | 64.4 | $\mathbf{0.013 \pm 0.004}$ $(-72.63\%)$ | 3.7 | 89.6 | $565.5 \pm 130.1$ $(-40.75\%)$ | 8.2 | 87.0 |

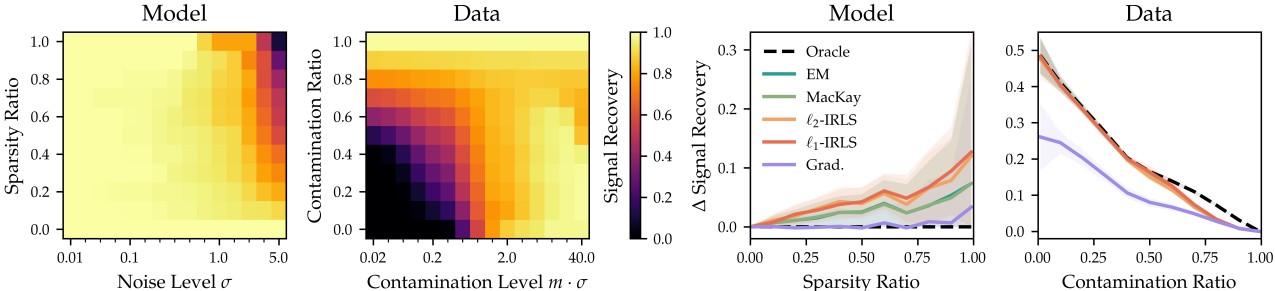

*Figure 3.* We visualize results for the synthetic setting in § 5.1 ($n = 500$, $d = 50$, avg. over 10 trials). *Left:* Representative data and weight recovery behaviour against key design parameters, here for $\ell_2$-IRLS. Recovery degrades as the signal becomes increasingly sparse, or contaminated noise resembles inlier noise ($m \leq 1$). *Right:* Relative gains in signal recovery from heteroscedastic *vs.* homoscedastic modelling, coarsly averaged across different noise levels. Largest benefits are obtained under realistic settings of high weight sparsity and low contamination, with improvements even in weight recovery.

(*i.e.* perplexity), interpreted as the number of active entries; dividing by the total dimension yields ESS (see App. D; Grendar (2006); Martino et al. (2016)). Alternatively, we report *signal recovery* as the fraction of true non-zero weights or outliers contained in the top-$k$ elements ranked by $\boldsymbol{\gamma}$ respective $\boldsymbol{\lambda}$ (*i.e.* top-$k$ recall).

### 5.1. Signal Recovery on Synthetic Data

To probe the behaviour and limitations of joint ARD in a controlled setting, we generate sparse linear regression data and inject response outliers by inflating the noise variance on a small subset of samples, that is $y_i = \mathbf{x}_i^\top \boldsymbol{\theta} + \epsilon_i$ with $\epsilon_i \sim \mathcal{N}(0, \sigma_i^2)$, $\sigma_i \in \{\sigma, m \cdot \sigma\}$. The sparsity ratio controls the fraction of nonzero entries in $\boldsymbol{\theta}$ and the contamination ratio controls the fraction of samples assigned the inflated noise level $m \cdot \sigma$.

Representative recovery heatmaps in Fig. 3 for $\ell_2$-IRLS show strong weight and outlier recovery overall, but degrade when the feature signal becomes weak, *i.e.* large $\sigma$ or very sparse $\boldsymbol{\theta}$, and when outliers are hard to separate, *i.e.* $m \leq 1$ or scarce contamination. Indeed, identifying noisy samples is notably harder than recovering sparse weights for the

given design. Nonetheless, heteroscedastic modelling improves over homoscedastic baselines across settings, with the largest gains in realistic regimes of sparse signals and low contamination. We additionally report an Oracle baseline using true weights and noise levels, and note that under homoscedastic noise the random chance recovery rate $k/n$ bounds attainable data recovery improvements.

### 5.2. Tabular Regression Benchmarks

Next, we evaluate joint ARD in all its variants on nine tabular regression benchmarks from UCI and OpenML (Dua et al., 2017; Bischl et al., 2025). Features are standardized and mapped via random Fourier features to approximate an RBF kernel, yielding a nonlinear and flexible, but correlated design ($d = 256$). We compare against sparse and robust regression baselines, with an exact RBF-kernel GP serving as a predictive reference point on clean data (Tab. 8).

At 10% outlier contamination on three representative datasets, we see in Tab. 3 that joint ARD maintains a strong predictive fit by downweighting corrupted samples. A well-working robustness mechanism should approximately reflect the injected outlier rate; indeed, an ESS($\mathbf{y}$) $\approx 90\%$

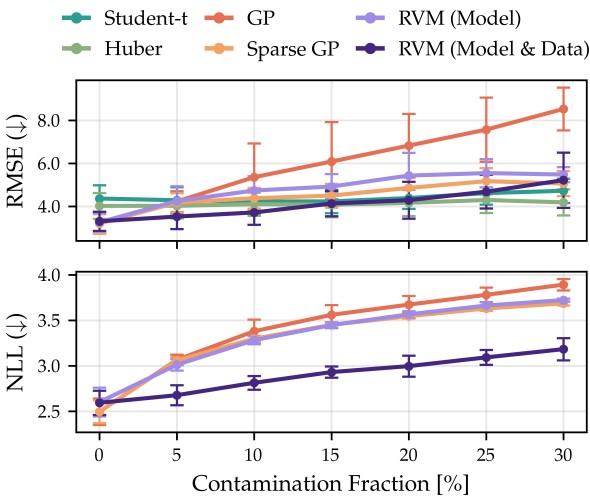

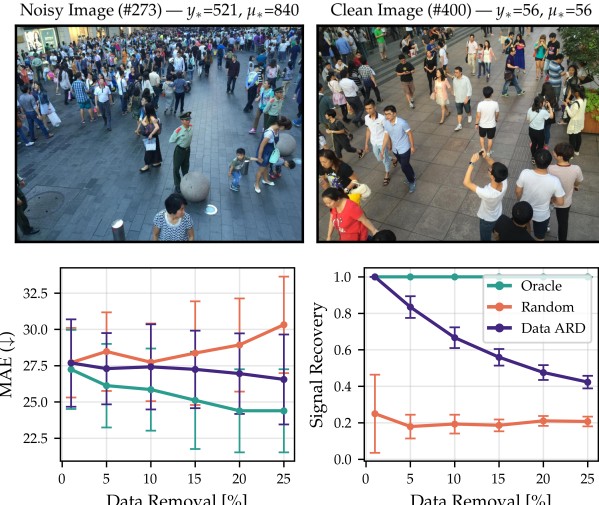

*Figure 4.* Kernel regression performance on *Boston* as a function of data contamination ($n = 506$, 20% test split, avg. over 20 trials, $\pm 1\sigma$). The RVM corresponds to $\ell_2$-IRLS in an RBF kernel basis with fixed scalar lengthscale.

*Figure 5.* Results for crowd counting on *ShanghaiTech* using pretrained DINO features, with joint ARD via EM on the model's output layer ($n = 716$, avg. over 10 trials, $\pm 1\sigma$). *Top:* Unreliable count labels coincide with crowded scenes and tend to be identified as such effectively. *Bottom:* Models fitted on gradually smaller data subsets ranked by learned $\boldsymbol{\lambda}$ maintain stable performance (MAE) despite a shrinking training signal.

indicates joint ARD downweighs roughly the right amount, as do robust Student-$t$ (Geweke, 1993) and Huber (Huber, 1964) baselines. In addition, joint ARD simultaneously yields meaningful feature sparsity, and predictive gains against sparsity-only baselines—homoscedastic variants, Ridge, and GP—are clearly visible. Nonetheless, SBL approaches can be sensitive to learning dynamics, as seen in the gradient-based variants.

### 5.3. Sparse Kernel Regression (RVM)

We conduct a similar regression experiment on the *Boston* dataset (Harrison Jr & Rubinfeld, 1978), replacing Fourier features with an RBF kernel basis and thereby instantiating the model as an RVM. Since the basis size scales with $n$ (here, $d = n = 506$) this constitutes a relatively high-capacity regime in which robustness is nontrivial to achieve. Predictive performance in Fig. 4 compares $\ell_2$-IRLS against robust and sparse baselines for varying contamination fractions, and we highlight a competitive disadvantage for RVM models against GP baselines, whose lengthscales are *optimized* internally rather than fixed at initialization.

Nonetheless, joint ARD consistently shows low RMSE and improved NLL relative to non-robust alternatives. Interestingly, an inducing-point sparse GP outperforms the full GP, suggesting mild robustness benefits from data subsetting. On clean data, the RVM (using $\ell_2$-IRLS) attains $\text{ESS}(\boldsymbol{\theta}) \approx 33\%$, indicating substantial shrinkage within the full set of basis functions. Prediction results across other SBL procedures show broadly comparable behaviour (Tab. 6, Tab. 7).

### 5.4. Neural Network Regression

Our final experiment considers image-based crowd counting on *ShanghaiTech* (Zhang et al., 2016). We extract fixed nonlinear DINO-2 features ($d = 384$, Oquab et al. (2024)) and learn a final linear layer with joint ARD, akin to a neural linear model (Ober & Rasmussen, 2019). Targets are transformed as $\mathbf{z} = \log(1 + \mathbf{y})$ to stabilize count variance and render label noise approximately additive and Gaussian in log-space, better matching our likelihood assumptions. We inject noise by randomly corrupting labels for a subset of high-count images, mimicking annotator inaccuracies in crowded scenes (for 20% effective contamination). This time we additionally act upon learned noise variances (via EM) by repeatedly refitting the model with the top-$k$-th fraction of samples removed, as determined by the ranking of $\boldsymbol{\lambda}$ on the full set. We compare to both an oracle (rank by true outliers) and random data removal.

Quantitative results in Fig. 5 affirm high signal recovery for data ARD initially matching the Oracle, with an expected drop curve as high-noise (and hence signal) samples are gradually pruned and the task hardens; and that removing high-noise samples stabilizes performance (here via *mean absolute error*, MAE) under increasing data scarcity, while random removal degrades performance. That is, joint ARD closer matches Oracle errors which gradually improve as true contaminants are removed, affirming an actionable data signal.

Yet, modest net MAE gains, together with very low $\text{ESS}(\boldsymbol{\theta})$

*Table 4.* We compare joint ARD with EM using the exact, closed-form posterior to EM with the BLR-provided approximate posterior on *Boston*, following the protocol in § 5.2 (avg. over 10 trials, $\pm 1\sigma$). Lowest errors in **bold**.

| Method | Uncontaminated | | 10% Contamination | |
|---|---|---|---|---|
| | RMSE ($\downarrow$) | NLL ($\downarrow$) | RMSE ($\downarrow$) | NLL ($\downarrow$) |
| EM (exact) | $\mathbf{3.31 \pm 0.43}$ | $\mathbf{2.55 \pm 0.11}$ | $4.75 \pm 1.00$ | $2.92 \pm 0.09$ |
| EM (BLR) | $4.05 \pm 0.70$ | $2.84 \pm 0.18$ | $\mathbf{4.29 \pm 0.66}$ | $\mathbf{2.91 \pm 0.11}$ |

(Fig. 7), suggest the injected corruptions are only mildly harmful and that the DINO representation is redundant for this task, with very few feature directions sufficing for good performance. This is plausible, since DINO-2 is trained on vast amounts of data and generalizes exceptionally well (Oquab et al., 2024), hence despite no finetuning crowd-counting likely constitutes a simple task for this powerful feature model.

# 6. Discussion

At its core, data ARD introduces heteroscedastic noise and induces sample-adaptive reweighting, promoting robustness to contaminants while remaining closely aligned with standard model-only ARD, a key strength of the approach. By treating model weights and data samples symmetrically as parameters governed by the marginal likelihood, joint ARD unifies sparsity and robustness within a single objective and highlights previously underexplored model–data correspondences. These insights extend naturally to $\ell_1$- and $\ell_2$-IRLS surrogate objectives, where regularization on both coefficients and residuals emerges. No additional *ad-hoc* structural assumptions are required beyond those inherent to the framework, and our regression experiments corroborate the practical benefits of the idea.

A principal limitation, however, is that our closed-form updates rely on a linear-in-weights parametrization (Eq. 1). While such models remain expressive through nonlinear feature mappings, they do not cover highly nonlinear parameterizations such as end-to-end deep neural networks. This constraint is shared by recent SBL-related proposals (Zhang et al. (2025); Wang et al. (2024); Ament & Gomes (2021), among others), and constitutes a principal angle for future work. Few tailored attempts have been made (Karaletsos & Rätsch, 2015; Kharitonov et al., 2018; Li et al., 2020), and broader applicability will likely trade tractability for scalable variational approximations.

A promising candidate in this direction is the *Bayesian Learning Rule* (BLR; Khan & Rue, 2023; Shen et al., 2024), a natural-gradient variational method whose objective recovers $\mathcal{L}(\boldsymbol{\gamma}, \boldsymbol{\lambda})$ in the linear setting (as we show in § B.8), permits an ARD-type parametrization, and draws its own connections to influence (Nickl et al., 2023; Tailor et al.,

2025). A tentative experiment in Tab. 4 suggests potential for future integration, combining closed-form EM update rules for $(\boldsymbol{\gamma}, \boldsymbol{\lambda})$ with a variational approximation for $p(\boldsymbol{\theta} \mid \mathbf{y}, \boldsymbol{\gamma}, \boldsymbol{\lambda})$. An alternative route may perhaps leverage links between GPs and neural networks (Dutordoir et al., 2021; Khan et al., 2019).

Other directions warranting future treatment include *(i)* a better understanding of $\mathcal{L}(\boldsymbol{\gamma}, \boldsymbol{\lambda})$ and possible tractable surrogates (Lotfi et al., 2022; Zhang et al., 2020), including potential to reduce existing learning sensitivities (§ D.1); *(ii)* the integration of alternative sparsity-inducing priors within SBL, such as the Horseshoe or Spike-and-Slab (Carvalho et al., 2009; Ray & Szabó, 2022); and *(iii)* the expansion to other tasks and settings benefitting from data sparsity, such as continual and active learning (Chang et al., 2023; Hübotter et al., 2024). Clearly, the related literature on sparse and robust models is vast, and many more comparisons can be drawn. In this work, our intent was to provide a simple yet principled foundation for robust SBL, opening up several promising avenues for investigation.

# Impact Statement

This work develops methods for sparse and robust regression by jointly identifying relevant model features and potentially unreliable training samples. Potential benefits include improved predictive reliability, better diagnostics for contaminated datasets, and more interpretable models in scientific or engineering applications. As with other data-weighting and outlier-detection methods, care is needed when applying the approach to socially sensitive data. Samples assigned low relevance should not automatically be interpreted as erroneous or unimportant without domain context and analysis. We do not foresee direct negative societal impacts beyond these general risks of misuse or misinterpretation.

# Acknowledgements

We thank the reviewers for constructive feedback, Putri van der Linden for help with designing the neural network experiment, and Dharmesh Tailor for supportive comments and pointing out insightful connections to influence functions. This project was generously supported by the Bosch Center for Artificial Intelligence and partially supported by JST CREST Grant No. JPMJCR2112.

# Author Contributions

AT led the project, including methodology, proofs, experiments, and paper writing. The idea was co-developed by AT, MEK and TM; MEK highlighted pruning dualities and connections to the BLR. CN and EN advised and facilitated the project; EN helped frame and place the paper in context.

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

# Joint Model and Data Sparsification via the Marginal Likelihood
## — Supplementary Materials —

## Contents

# A. Additional Background

**Bayesian pruning.** More generally, the introduction of sparsity-inducing and shrinkage priors including Laplace (Seeger et al., 2007), Horseshoe (Carvalho et al., 2009), Spike-and-slab (Ray & Szabó, 2022) and complexity-based (Castillo & van der Vaart, 2012; Castillo et al., 2015; Martin et al., 2017) has been studied extensively for Bayesian pruning, variable, and model selection. Perhaps as a well-known bridge between frequentist and Bayesian ideas, the LASSO estimator is both interpreted as an M-estimator for high-dimensional settings or a Laplace shrinkage prior for sparsification (Hastie et al., 2015). Recent work includes shrinkage priors on model weights in combination with Bayesian neural networks (Louizos et al., 2017; Molchanov et al., 2017; Neklyudov et al., 2017; Ghosh et al., 2019; Nalisnick et al., 2021; 2019), and has also seen explicit use of the marginal likelihood for model sparsification (Immer et al., 2021; Bouchiat et al., 2023; Dhahri et al., 2024), albeit necessitating approximations via the Laplace.

**Further related works.** Additional related works on SBL include Titsias & Lázaro-Gredilla (2014) for a variational inference approach and Rudy & Sapsis (2021) for a discussion on threshold design to strictly nullify down-weighted features. On heteroscedastic modelling, Helgøy et al. (2024) consider the RVM with a Laplace approximation, while Altamirano et al. (2024); Algikar & Mili (2023); Andrade & Takeda (2023) present interesting takes on robust GPs using generalized inference or non-standard likelihoods. Park et al. (2022) present a GP model with additional bias term for outlier identification.

Another interesting conceptual bridge can be drawn to robustness in *principal component analysis* (PCA), such as Candès et al. (2011), when viewed through a shared regularization pattern. Both approaches encourage parsimonious signal representation (via a low-rank component resp. sparse model parameters $\boldsymbol{\gamma}$) and structured explanation of corruptions (via a sparse component resp. per-sample noise $\boldsymbol{\lambda}$), and also share overlap in IRLS-type solver procedures. Nonetheless, robust PCA (Candès et al., 2011) requires distinct low-rank and corrupt components for (unsupervised) identifiability, while in our (supervised) setting each sample may be an important contributor *and* corrupted, with the marginal likelihood balancing model fit and robustness.

**Further placement in existing literature.** An overview pertaining to key properties of interest is given in Tab. 1. The proposed joint ARD framework can be positioned at the intersection of two lines of research: sparse Bayesian learning and robust regression. On the model side, the ARD principle provides a well-established route to weight or feature sparsity and variable selection, with ties to compressed sensing (*e.g.* via pursuit algorithms), sparse GPs, and more broadly Bayesian pruning (*e.g.* regarding prior choices). On the data side, heteroscedastic noise modelling includes Bayesian and non-Bayesian robust regression, robust GPs, and heavy-tailed likelihoods and noise models. We ensure robustness by leveraging principles from the latter line of research, while operating within the ARD framework initially designed for the former, sparsity-focused line of research.

# B. Mathematical Details

We relate $\mathcal{L}(\boldsymbol{\gamma}, \boldsymbol{\lambda})$ to the dual objective $\tilde{\mathcal{L}}(\boldsymbol{\gamma}, \boldsymbol{\lambda})$, provide detailed derivations for all heteroscedastic SBL update rules (including a summary table in Tab. 5) and discuss how marginal likelihood optimization connects to the Bayesian learning rule (Khan & Rue, 2023) in linear models (§ B.8).

## B.1. Formulation of the dual objective

We enumerate the primal objective from Eq. 3 as follows

$$\underbrace{- \log p(\mathbf{y} \mid \boldsymbol{\gamma}, \boldsymbol{\lambda})}_{\text{(III)}} = \frac{n}{2} \log(2\pi) + \underbrace{\frac{1}{2} \log |\Sigma_{\mathbf{y}}|}_{\text{(II)}} + \underbrace{\frac{1}{2} \mathbf{y}^{\top} \Sigma_{\mathbf{y}}^{-1} \mathbf{y}}_{\text{(I)}}$$

and re-express each part following manipulations.

(I) Using the Woodbury identity $\boxed{(A + CBC^{\top})^{-1} = A^{-1} - A^{-1}C(B^{-1} + C^{\top}A^{-1}C)^{-1}C^{\top}A^{-1}}$ on $\Sigma_{\mathbf{y}}^{-1}$ we obtain

$$\Sigma_{\mathbf{y}}^{-1} = (\Lambda + \mathbf{X}\,\Gamma^{-1}\,\mathbf{X}^{\top})^{-1} = \Lambda^{-1} - \Lambda^{-1}\mathbf{X}\underbrace{(\Gamma + \mathbf{X}^{\top}\Lambda^{-1}\mathbf{X})^{-1}}_{= \Sigma_{\boldsymbol{\theta}} \in \mathbb{R}^{d \times d}}\mathbf{X}^{\top}\Lambda^{-1} = \Lambda^{-1} - \Lambda^{-1}\mathbf{X}\Sigma_{\boldsymbol{\theta}}\mathbf{X}^{\top}\Lambda^{-1}.$$

Plugging into the expression, we then have

$$\mathbf{y}^\top \Sigma_{\mathbf{y}}^{-1} \mathbf{y} = \mathbf{y}^\top \left(\Lambda^{-1} - \Lambda^{-1}\mathbf{X}\Sigma_{\boldsymbol{\theta}}\mathbf{X}^\top\Lambda^{-1}\right)\mathbf{y} = \mathbf{y}^\top\Lambda^{-1}\mathbf{y} - \mathbf{y}^\top\Lambda^{-1}\mathbf{X}\Sigma_{\boldsymbol{\theta}}\underbrace{\mathbf{X}^\top\Lambda^{-1}\mathbf{y}}_{= \tilde{\mathbf{y}} \in \mathbb{R}^d} = \mathbf{y}^\top\Lambda^{-1}\mathbf{y} - \tilde{\mathbf{y}}^\top\Sigma_{\boldsymbol{\theta}}\tilde{\mathbf{y}},$$

where $\tilde{\mathbf{y}} = \mathbf{X}^\top\Lambda^{-1}\mathbf{y}$ becomes the data projection into weight space.

(II) Applying the matrix determinant lemma as $\boxed{|A + UWV^\top| = |W^{-1} + V^\top A^{-1}U| \cdot |W| \cdot |A|}$ we obtain

$$\log|\Sigma_{\mathbf{y}}| = \log\left(|\Gamma + \mathbf{X}^\top\Lambda^{-1}\mathbf{X}| \cdot |\Gamma^{-1}| \cdot |\Lambda|\right) = \log\underbrace{|\Gamma + \mathbf{X}^\top\Lambda^{-1}\mathbf{X}|}_{= \Sigma_{\boldsymbol{\theta}}^{-1}} - \log|\Gamma| + \log|\Lambda|,$$

where again $\Sigma_{\boldsymbol{\theta}}$ emerges. Combining (I) and (II) to re-express (III), the dual objective (Eq. 4) is given by

$$\begin{aligned}
\tilde{\mathcal{L}}(\boldsymbol{\gamma}, \boldsymbol{\lambda}) &= -\log p(\tilde{\mathbf{y}} \mid \boldsymbol{\gamma}, \boldsymbol{\lambda}) \\
&= \frac{n}{2}\log(2\pi) + \frac{1}{2}\left(\log|\Sigma_{\boldsymbol{\theta}}^{-1}| - \log|\Gamma| + \log|\Lambda|\right) + \frac{1}{2}\left(\mathbf{y}^\top\Lambda^{-1}\mathbf{y} - \tilde{\mathbf{y}}^\top\Sigma_{\boldsymbol{\theta}}\tilde{\mathbf{y}}\right) \\
&\equiv \log|\Sigma_{\boldsymbol{\theta}}^{-1}| - \log|\Gamma| + \log|\Lambda| + \left(\mathbf{y}^\top\Lambda^{-1}\mathbf{y} - \tilde{\mathbf{y}}^\top\Sigma_{\boldsymbol{\theta}}\tilde{\mathbf{y}}\right).
\end{aligned}$$

## B.2. Update rules from first-order optimality conditions

Consider the primal objective (Eq. 3) as

$$\mathcal{L}(\boldsymbol{\gamma}, \boldsymbol{\lambda}) = \log|\Sigma_{\mathbf{y}}| + \mathbf{y}^\top\Pi_{\mathbf{y}}\,\mathbf{y},$$

where we define $\Pi_{\mathbf{y}} = \Sigma_{\mathbf{y}}^{-1}$ for notational convenience moving forward.

**Update for $\gamma_j$.** A first-order optimality condition for $\gamma_j$ is given by

$$\frac{\partial\mathcal{L}(\boldsymbol{\gamma}, \boldsymbol{\lambda})}{\partial\gamma_j} \overset{!}{=} 0.$$

Differentiation of the individual components yields that

$$\frac{\partial\log|\Sigma_{\mathbf{y}}|}{\partial\gamma_j} = \text{trace}(\Pi_{\mathbf{y}}\frac{\partial\Sigma_{\mathbf{y}}}{\partial\gamma_j}) = -\frac{1}{\gamma_j^2}\mathbf{x}_j^\top\Pi_{\mathbf{y}}\mathbf{x}_j, \qquad \frac{\partial\Pi_{\mathbf{y}}}{\partial\gamma_j} = -\Pi_{\mathbf{y}}\frac{\partial\Sigma_{\mathbf{y}}}{\partial\gamma_j}\Pi_{\mathbf{y}} = -\frac{1}{\gamma_j^2}\Pi_{\mathbf{y}}\mathbf{x}_j^\top\mathbf{x}_j\Pi_{\mathbf{y}},$$

where $\mathbf{x}_j$ denotes the $j$-th column of $\mathbf{X}$. Combined, we obtain that

$$\frac{\partial\mathcal{L}(\boldsymbol{\gamma}, \boldsymbol{\lambda})}{\partial\gamma_j} = -\frac{1}{\gamma_j^2}\left(\mathbf{x}_j^\top\Pi_{\mathbf{y}}\mathbf{x}_j - \mathbf{y}^\top\left(\Pi_{\mathbf{y}}\mathbf{x}_j^\top\mathbf{x}_j\Pi_{\mathbf{y}}\right)\mathbf{y}\right) = -\frac{1}{\gamma_j^2}\left(\mathbf{x}_j^\top\Pi_{\mathbf{y}}\mathbf{x}_j - \left(\mathbf{x}_j^\top\Pi_{\mathbf{y}}\mathbf{y}\right)^2\right) \overset{!}{=} 0.$$

We now derive two identities to obtain more convenient expressions for the condition. First, using the posterior mean and covariance definitions we see that

$$\boldsymbol{\mu}_{\boldsymbol{\theta}} = \Sigma_{\boldsymbol{\theta}}\mathbf{X}^\top\Lambda^{-1}\mathbf{y} \;\Leftrightarrow\; \Sigma_{\boldsymbol{\theta}}^{-1}\boldsymbol{\mu}_{\boldsymbol{\theta}} = \mathbf{X}^\top\Lambda^{-1}\mathbf{y} \;\Leftrightarrow\; \Gamma\boldsymbol{\mu}_{\boldsymbol{\theta}} = \mathbf{X}^\top\Lambda^{-1}(\mathbf{y} - \mathbf{X}\boldsymbol{\mu}_{\boldsymbol{\theta}}),$$

and using $\Pi_{\mathbf{y}} = \Lambda^{-1} - \Lambda^{-1}\mathbf{X}\Sigma_{\boldsymbol{\theta}}\mathbf{X}^\top\Lambda^{-1}$ via Woodbury we observe that $\Pi_{\mathbf{y}}\mathbf{y} = \Lambda^{-1}(\mathbf{y} - \mathbf{X}\boldsymbol{\mu}_{\boldsymbol{\theta}})$ (see update derivation for $\lambda_i$ below), which can be plugged into the right-hand side. Taking the $j$-th component, we obtain the first identity as $\mathbf{x}_j^\top\Pi_{\mathbf{y}}\mathbf{y} = \gamma_j \cdot \mu_{\boldsymbol{\theta},j}$.

Next, we re-use the identity for $\Pi_{\mathbf{y}}$ and see that the inner product on the $j$-th column is given by

$$\mathbf{x}_j^\top\Pi_{\mathbf{y}}\mathbf{x}_j = \mathbf{x}_j^\top\Lambda^{-1}\mathbf{x}_j - \mathbf{x}_j^\top\Lambda^{-1}\mathbf{X}\Sigma_{\boldsymbol{\theta}}\mathbf{X}^\top\Lambda^{-1}\mathbf{x}_j = u_j - \mathbf{u}^\top\Sigma_{\boldsymbol{\theta}}\mathbf{u},$$

where we define $\mathbf{u} = \mathbf{X}^\top\Lambda^{-1}\mathbf{x}_j$. Additionally defining $\mathbf{s}_j = \Sigma_{\boldsymbol{\theta}}\mathbf{e}_j$ as the $j$-th column of $\Sigma_{\boldsymbol{\theta}}$, left-multiplying by $\mathbf{u}^\top$ and solving we find the expression $\mathbf{u}^\top\mathbf{s}_j = 1 - \gamma_j \cdot s_{jj} = 1 - \gamma_j \cdot [\Sigma_{\boldsymbol{\theta}}]_{jj}$. Furthermore, substituting $\Pi_{\mathbf{y}}$ via Woodbury and

simplifying using the identity $\Sigma_{\boldsymbol{\theta}}(\Gamma + \mathbf{X}^\top \Lambda^{-1}\mathbf{X}) = \mathbf{I}_d$ we find the expression $\Pi_{\mathbf{y}} \mathbf{X} = \Lambda^{-1}\mathbf{X}\,\Sigma_{\boldsymbol{\theta}}\,\Gamma$, and for the $j$-th column $\Pi_{\mathbf{y}}\mathbf{x}_j = \Lambda^{-1}\mathbf{X}\,\Sigma_{\boldsymbol{\theta}}\,\Gamma\,\mathbf{e}_j = \gamma_j \cdot \Lambda^{-1}\mathbf{X}\,\mathbf{s}_j$. Therefore we obtain the final identity

$$\mathbf{x}_j^\top \Pi_{\mathbf{y}}\,\mathbf{x}_j = \gamma_j(\mathbf{X}^\top \Lambda^{-1}\mathbf{x}_j)^\top \mathbf{s}_j = \gamma_j \cdot \mathbf{u}^\top \mathbf{s}_j = \gamma_j(1 - \gamma_j \cdot s_{jj}) = \gamma_j(1 - \gamma_j \cdot [\Sigma_{\boldsymbol{\theta}}]_{jj}).$$

Plugging in both identities into the optimality condition, we find the update as

$$\gamma_j(1 - \gamma_j \cdot [\Sigma_{\boldsymbol{\theta}}]_{jj}) = \gamma_j^2 \cdot \mu_{\boldsymbol{\theta},j}^2 \;\Leftrightarrow\; (1 - \gamma_j \cdot [\Sigma_{\boldsymbol{\theta}}]_{jj}) = \gamma_j \cdot \mu_{\boldsymbol{\theta},j}^2 \;\Leftrightarrow\; \gamma_j^{t+1} = \frac{1 - \gamma_j^t \cdot [\Sigma_{\boldsymbol{\theta}}]_{jj}}{\mu_{\boldsymbol{\theta},j}^2},$$

where $\gamma_j^t$ denotes the previous iterate on the right-hand side (and equates MacKay's update).

**Update for $\lambda_i$.** We follow the same general steps as above. A first-order optimality condition for $\lambda_i$ is given by

$$\frac{\partial \mathcal{L}(\boldsymbol{\gamma}, \boldsymbol{\lambda})}{\partial \lambda_i} \overset{!}{=} 0.$$

Differentiation of the individual components yields that

$$\frac{\partial \log |\Sigma_{\mathbf{y}}|}{\partial \lambda_i} = \mathrm{trace}(\Pi_{\mathbf{y}}\,\mathbf{E}_{ii}) = [\Pi_{\mathbf{y}}]_{ii}, \qquad \frac{\partial \Pi_{\mathbf{y}}}{\partial \lambda_i} = -\Pi_{\mathbf{y}}\,\mathbf{E}_{ii}\,\Pi_{\mathbf{y}}$$

since $\Lambda = \mathrm{diag}(\boldsymbol{\lambda})$ and $\frac{\partial \Sigma_{\mathbf{y}}}{\partial \lambda_i} = \mathbf{E}_{ii}$, with $\mathbf{E}_{ii} \in [0,1]^{n \times n}$ denoting the basis with a single 1 at $(i,i)$. Combined, we obtain

$$\frac{\partial \mathcal{L}(\boldsymbol{\gamma}, \boldsymbol{\lambda})}{\partial \lambda_i} = [\Pi_{\mathbf{y}}]_{ii} - \mathbf{y}^\top (\Pi_{\mathbf{y}}\,\mathbf{E}_{ii}\,\Pi_{\mathbf{y}})\,\mathbf{y} = [\Pi_{\mathbf{y}}]_{ii} - ([\Pi_{\mathbf{y}}\,\mathbf{y}]_i)^2 \overset{!}{=} 0.$$

We now re-use the Woodbury identity for $\Pi_{\mathbf{y}}$ to obtain more convenient expressions for the condition. Since $\Pi_{\mathbf{y}} = \Lambda^{-1} - \Lambda^{-1}\mathbf{X}\Sigma_{\boldsymbol{\theta}}\mathbf{X}^\top \Lambda^{-1}$, we multiply with $\mathbf{y}$ and observe that

$$\Pi_{\mathbf{y}}\,\mathbf{y} = \Lambda^{-1}\,\mathbf{y} - \Lambda^{-1}\mathbf{X}\Sigma_{\boldsymbol{\theta}}\mathbf{X}^\top \Lambda^{-1}\,\mathbf{y} = \Lambda^{-1}\,\mathbf{y} - \Lambda^{-1}\mathbf{X}\,\boldsymbol{\mu_\theta} = \Lambda^{-1}(\mathbf{y} - \mathbf{X}\,\boldsymbol{\mu_\theta}),$$

using the definition of the posterior mean. Defining $\mathbf{r} = (\mathbf{y} - \mathbf{X}\,\boldsymbol{\mu_\theta})$ we see that $[\Pi_{\mathbf{y}}\,\mathbf{y}]_i = [\Lambda^{-1}\,\mathbf{r}]_i = \lambda_i^{-1} \cdot r_i$. Taking the diagonal of the Woodbury identity, we see that

$$[\Pi_{\mathbf{y}}]_{ii} = [\Lambda^{-1} - \Lambda^{-1}\mathbf{X}\Sigma_{\boldsymbol{\theta}}\mathbf{X}^\top \Lambda^{-1}]_{ii} = \lambda_i^{-1} - \lambda_i^{-2} \cdot q_i, \quad q_i = \mathbf{x}_i^\top \Sigma_{\boldsymbol{\theta}}\,\mathbf{x}_i$$

where $\mathbf{x}_i^\top$ indicates the $i$-th row of $\mathbf{X}$. Plugging the two expressions into the condition, we obtain

$$\frac{\partial \mathcal{L}(\boldsymbol{\gamma}, \boldsymbol{\lambda})}{\partial \lambda_i} = [\Pi_{\mathbf{y}}]_{ii} - ([\Pi_{\mathbf{y}}\,\mathbf{y}]_i)^2 = \lambda_i^{-1} - \lambda_i^{-2} \cdot q_i - (\lambda_i^{-1} \cdot r_i)^2 \overset{!}{=} 0.$$

Multiplication with $\lambda_i^2$ and re-arranging yields the final update form

$$\lambda_i^{t+1} = r_i^2 + q_i = (y_i - \mathbf{x}_i^\top \boldsymbol{\mu_\theta})^2 + \mathbf{x}_i^\top \Sigma_{\boldsymbol{\theta}}\,\mathbf{x}_i.$$

### B.3. Update rules for Expectation Maximization

We target the complete-data joint log-likelihood of $(\mathbf{y}, \boldsymbol{\theta})$, *i.e.*

$$\log p(\mathbf{y}, \boldsymbol{\theta} \mid \boldsymbol{\gamma}, \boldsymbol{\lambda}) \propto \log p(\mathbf{y} \mid \boldsymbol{\theta}, \boldsymbol{\lambda}) + \log p(\boldsymbol{\theta} \mid \boldsymbol{\gamma}),$$

where $\boldsymbol{\theta}$ forms the latent variable. The E-step sees computing the posterior $q_t(\boldsymbol{\theta}) = p(\boldsymbol{\theta} \mid \mathbf{y}, \boldsymbol{\gamma}^t, \boldsymbol{\lambda}^t)$, where we condition on fixed parameters $\boldsymbol{\gamma}^t, \boldsymbol{\lambda}^t$ from the previous step. Updates for $\boldsymbol{\gamma}, \boldsymbol{\lambda}$ are obtained in the M-step by maximizing the expected joint under the current posterior, given by

$$Q(\boldsymbol{\gamma}, \boldsymbol{\lambda}) = \mathbb{E}_{q_t(\boldsymbol{\theta})}[\log p(\mathbf{y}, \boldsymbol{\theta} \mid \boldsymbol{\gamma}, \boldsymbol{\lambda})] = \mathbb{E}_{q_t(\boldsymbol{\theta})}[\log p(\mathbf{y} \mid \boldsymbol{\theta}, \boldsymbol{\lambda})] + \mathbb{E}_{q_t(\boldsymbol{\theta})}[\log p(\boldsymbol{\theta} \mid \boldsymbol{\gamma})].$$

**Update for $\gamma_j$.** Considering dependencies on $\gamma$ and since $p(\boldsymbol{\theta} \mid \boldsymbol{\gamma}) = \prod_{j=1}^{d} \mathcal{N}(0, \gamma_j^{-1})$ we obtain that

$$Q(\boldsymbol{\gamma}, \cdot) = \mathbb{E}_{q_t(\boldsymbol{\theta})}[\log p(\boldsymbol{\theta} \mid \boldsymbol{\gamma})] \equiv \sum_{j=1}^{d} \left( \log \gamma_j - \gamma_j \cdot \mathbb{E}_{q_t(\boldsymbol{\theta})}[\theta_j^2] \right),$$

such that first-order optimality implies

$$\frac{\partial Q(\boldsymbol{\gamma}, \cdot)}{\partial \gamma_j} = \frac{1}{\gamma_j} - \mathbb{E}_{q_t(\boldsymbol{\theta})}[\theta_j^2] \overset{!}{=} 0 \Leftrightarrow \gamma_j^{t+1} = \frac{1}{\mathbb{E}_{q_t(\boldsymbol{\theta})}[\theta_j^2]} = \frac{1}{\mu_{\boldsymbol{\theta},j}^2 + [\Sigma_{\boldsymbol{\theta}}]_{jj}},$$

since $q_t(\boldsymbol{\theta}) = \mathcal{N}(\boldsymbol{\mu_\theta}, \Sigma_{\boldsymbol{\theta}})$ and we match its second moment.

**Update for $\lambda_i$.** Considering dependencies on $\boldsymbol{\lambda}$ and since $p(\mathbf{y} \mid \boldsymbol{\theta}, \boldsymbol{\lambda}) = \prod_{i=1}^{n} \mathcal{N}(\mathbf{x}_i^\top \boldsymbol{\theta}, \lambda_i)$ we obtain that

$$Q(\cdot, \boldsymbol{\lambda}) = \mathbb{E}_{q_t(\boldsymbol{\theta})}[\log p(\mathbf{y} \mid \boldsymbol{\theta}, \boldsymbol{\lambda})] \equiv \sum_{i=1}^{n} \left( \log \lambda_i + \frac{1}{\lambda_i} \cdot \mathbb{E}_{q_t(\boldsymbol{\theta})}[(y_i - \mathbf{x}_i^\top \boldsymbol{\theta})^2] \right),$$

such that first-order optimality implies

$$\frac{\partial Q(\cdot, \boldsymbol{\lambda})}{\partial \lambda_i} = \frac{1}{\lambda_i} - \frac{1}{\lambda_i^2} \cdot \mathbb{E}_{q_t(\boldsymbol{\theta})}[(y_i - \mathbf{x}_i^\top \boldsymbol{\theta})^2] \overset{!}{=} 0 \Leftrightarrow \lambda_i^{t+1} = \mathbb{E}_{q_t(\boldsymbol{\theta})}[(y_i - \mathbf{x}_i^\top \boldsymbol{\theta})^2] = (y_i - \mathbf{x}_i^\top \boldsymbol{\mu_\theta})^2 + \mathbf{x}_i^\top \Sigma_{\boldsymbol{\theta}} \mathbf{x}_i,$$

since $q_t(\boldsymbol{\theta}) = \mathcal{N}(\boldsymbol{\mu_\theta}, \Sigma_{\boldsymbol{\theta}})$ and $\mathbb{E}[z^2] = \mathbb{E}[z]^2 + \mathrm{Var}(z)$ by law of total variance applied to $z = (y_i - \mathbf{x}_i^\top \boldsymbol{\theta})$.

**Update for $\lambda$ in the homoscedastic case.** Similarly to the heteroscedastic case above we obtain

$$Q(\cdot, \boldsymbol{\lambda}) = \mathbb{E}_{q_t(\boldsymbol{\theta})}[\log p(\mathbf{y} \mid \boldsymbol{\theta}, \boldsymbol{\lambda})] \equiv n \log \lambda + \frac{1}{\lambda} \cdot \mathbb{E}_{q_t(\boldsymbol{\theta})} \left[ \sum_{i=1}^{n} (y_i - \mathbf{x}_i^\top \boldsymbol{\theta})^2 \right] = n \log \lambda + \frac{1}{\lambda} \cdot \mathbb{E}_{q_t(\boldsymbol{\theta})}[\|\mathbf{y} - \mathbf{X}\boldsymbol{\theta}\|_2^2],$$

and by first-order optimality we have

$$\frac{\partial Q(\cdot, \boldsymbol{\lambda})}{\partial \lambda} = \frac{n}{\lambda} - \frac{1}{\lambda^2} \cdot \mathbb{E}_{q_t(\boldsymbol{\theta})}[\|\mathbf{y} - \mathbf{X}\boldsymbol{\theta}\|_2^2] \overset{!}{=} 0 \Leftrightarrow \lambda^{t+1} = \frac{1}{n} \mathbb{E}_{q_t(\boldsymbol{\theta})}[\|\mathbf{y} - \mathbf{X}\boldsymbol{\theta}\|_2^2] = \frac{1}{n} \left( \|\mathbf{y} - \mathbf{X}\boldsymbol{\mu_\theta}\|_2^2 + \mathrm{trace}(\mathbf{X}\Sigma_{\boldsymbol{\theta}}\mathbf{X}^\top) \right),$$

or more explicitly $\lambda^{t+1} = \frac{1}{n} \sum_{i=1}^{n} \left[ (y_i - \mathbf{x}_i^\top \boldsymbol{\mu_\theta})^2 + \mathbf{x}_i^\top \Sigma_{\boldsymbol{\theta}} \mathbf{x}_i \right]$ as a simplified, sample-averaged estimate.

## B.4. Update rules for MacKay's updates

MacKay's updates can be directly obtained from first-order optimality conditions, and we detail those derivations in § B.2. Alternatively, we can start from the EM updates in § B.3 and leverage simple fixed-point rearrangements (*i.e.* assuming convergence to the EM optima) to arrive at the same rules, which we show next.

**Update for $\gamma_j$.** Starting from the EM update and multiplying by $\gamma_j$ we obtain

$$\gamma_j^{t+1} = \frac{1}{\mu_{\boldsymbol{\theta},j}^2 + [\Sigma_{\boldsymbol{\theta}}]_{jj}} \Leftrightarrow \frac{1}{\gamma_j^{t+1}} = \mu_{\boldsymbol{\theta},j}^2 + [\Sigma_{\boldsymbol{\theta}}]_{jj} \Leftrightarrow \frac{\gamma_j}{\gamma_j^{t+1}} = \gamma_j \cdot \mu_{\boldsymbol{\theta},j}^2 + \gamma_j \cdot [\Sigma_{\boldsymbol{\theta}}]_{jj}.$$

At a fixed point we then have $\gamma_j^{t+1} = \gamma_j$ and thus

$$1 = \gamma_j \cdot \mu_{\boldsymbol{\theta},j}^2 + \gamma_j \cdot [\Sigma_{\boldsymbol{\theta}}]_{jj} \Leftrightarrow (1 - \gamma_j \cdot [\Sigma_{\boldsymbol{\theta}}]_{jj}) = \gamma_j \cdot \mu_{\boldsymbol{\theta},j}^2 \Leftrightarrow \gamma_j^{t+1} = \frac{1 - \gamma_j^t \cdot [\Sigma_{\boldsymbol{\theta}}]_{jj}}{\mu_{\boldsymbol{\theta},j}^2},$$

recovering the optimality-based update.

**Update for $\lambda_i$.** For the heteroscedastic case there is no simplification via fixed-point conditions, and so the update takes the same direct form as for EM, that is

$$\lambda_i^{t+1} = (y_i - \mathbf{x}_i^\top \boldsymbol{\mu_\theta})^2 + \mathbf{x}_i^\top \Sigma_{\boldsymbol{\theta}} \mathbf{x}_i.$$

**Update for $\lambda$ in the homoscedastic case.** We revisit the EM update given by $\lambda^{t+1} = \frac{1}{n} \left( \|\mathbf{y} - \mathbf{X}\boldsymbol{\mu_\theta}\|_2^2 + \text{trace}(\mathbf{X}\Sigma_\theta\mathbf{X}^\top) \right)$ and observe that simplifications can be done to the trace term. In the homoscedastic setting the posterior covariance simplifies to $\Sigma_\theta = (\Gamma + \frac{1}{\lambda}\mathbf{X}^\top\mathbf{X})^{-1}$, and thus we see that $\mathbf{X}^\top\mathbf{X} = \lambda(\Sigma_\theta^{-1} - \Gamma)$. It follows for the trace term that

$$\text{trace}(\mathbf{X}\Sigma_\theta\mathbf{X}^\top) = \text{trace}(\Sigma_\theta\mathbf{X}^\top\mathbf{X}) = \lambda \cdot \left( \text{trace}(\Sigma_\theta\Sigma_\theta^{-1}) - \text{trace}(\Sigma_\theta\Gamma) \right) = \lambda \cdot \left( d - \sum_{j=1}^{d} \gamma_j \cdot [\Sigma_\theta]_{jj} \right),$$

where $\text{trace}(\Sigma_\theta\Sigma_\theta^{-1}) = \text{trace}(\mathbf{I}_d) = d$. Plugging into the update, invoking the fixed-point condition such that $\lambda^{t+1} = \lambda$, and re-arranging we obtain

$$\lambda \cdot \left( n - d + \sum_{j=1}^{d} \gamma_j \cdot [\Sigma_\theta]_{jj} \right) = \|\mathbf{y} - \mathbf{X}\boldsymbol{\mu_\theta}\|_2^2 \;\Leftrightarrow\; \lambda^{t+1} = \frac{\|\mathbf{y} - \mathbf{X}\boldsymbol{\mu_\theta}\|_2^2}{n - \sum_{j=1}^{d} \left( 1 - \gamma_j^t \cdot [\Sigma_\theta]_{jj} \right)},$$

which aligns with the classical MacKay update found in the literature, *e.g.* see Tipping (2001), App. A.2.

## B.5. Update rules for $\ell_2$-IRLS

Rather than directly operating on $\mathcal{L}(\boldsymbol{\gamma}, \boldsymbol{\lambda})$ (Eq. 3), Wipf & Nagarajan (2007; 2010) employ a majorization-minimization strategy on an upper-bounding surrogate objective via IRLS. Following notation in Wipf & Nagarajan (2010), $\mathcal{L}(\boldsymbol{\gamma}, \boldsymbol{\lambda})$ can be equivalently expressed (up to constants) as

$$\mathcal{L}(\boldsymbol{\gamma}, \boldsymbol{\lambda}) = (\mathbf{y} - \mathbf{X}\boldsymbol{\theta})^\top \Lambda^{-1} (\mathbf{y} - \mathbf{X}\boldsymbol{\theta}) + g_{\text{SBL}}(\boldsymbol{\theta}),$$

where $g_{\text{SBL}}(\boldsymbol{\theta}) \equiv \min_{\boldsymbol{\gamma} \geq 0} \{\boldsymbol{\theta}^\top \Gamma \boldsymbol{\theta} + \log|\Sigma_\mathbf{y}|\}$ forms a non-separable penalty term. As $g_{\text{SBL}}(\boldsymbol{\theta})$ is (componentwise) non-decreasing and concave in $\boldsymbol{\theta}^2$ (and similarly for $|\boldsymbol{\theta}|$, see $\ell_1$-IRLS) minimization can be accomplished by iterative $\ell_2$-reweighted least squares. For tractable updates, a quadratic upper bound on $g_{\text{SBL}}(\boldsymbol{\theta})$ is derived as

$$g_{\text{SBL}}(\boldsymbol{\theta}) \leq \boldsymbol{\theta}^\top \Gamma \boldsymbol{\theta} + \log|\Sigma_\mathbf{y}| \leq \boldsymbol{\theta}^\top \Gamma \boldsymbol{\theta} + \log|\Sigma_\theta^{-1}| - \log|\Gamma| + \log|\Lambda| \leq \boldsymbol{\theta}^\top \Gamma \boldsymbol{\theta} + (\mathbf{z}^\top\boldsymbol{\gamma} - h^*(\mathbf{z})) - \log|\Gamma| + \log|\Lambda|,$$

leveraging the determinant lemma and (concave) Fenchel duality of $\log|\Sigma_\theta^{-1}|$ to admit a separable upper bound, and yielding the IRLS-style surrogate

$$\mathcal{L}^{\text{IRLS}}(\boldsymbol{\gamma}, \boldsymbol{\lambda}; \mathbf{z}) = (\mathbf{y} - \mathbf{X}\boldsymbol{\theta})^\top \Lambda^{-1} (\mathbf{y} - \mathbf{X}\boldsymbol{\theta}) - h^*(\mathbf{z}) + \sum_{j=1}^{d} \left( \gamma_j \cdot (\theta_j^2 + z_j) - \log\gamma_j \right) + \log|\Lambda| \geq \mathcal{L}(\boldsymbol{\gamma}, \boldsymbol{\lambda}).$$

We refer to Wipf & Nagarajan (2010; 2007) for more details on the employed concave conjugate $h^*(\mathbf{z})$ and the auxiliary-bound viewpoint (for the homoscedastic setting).

**Updates for $\mathbf{z}, \boldsymbol{\theta}$.** Treating $\boldsymbol{\lambda}$ as fixed, the optimal value for auxiliary variables $\mathbf{z}$ is given by the slope of $\log|\Sigma_\theta^{-1}|$ at current iterate $\boldsymbol{\gamma}$ (Fenchel optimality condition), *i.e.*

$$z_j^{t+1} = \frac{\partial \log|\Sigma_\theta^{-1}|}{\partial \gamma_j} = \text{trace}(\Sigma_\theta \frac{\partial \Sigma_\theta^{-1}}{\partial \gamma_j}) = \text{trace}(\Sigma_\theta \mathbf{E}_{jj}) = [\Sigma_\theta]_{jj}.$$

Minimizing the surrogate w.r.t. $\boldsymbol{\theta}$ at fixed $(\boldsymbol{\gamma}, \boldsymbol{\lambda}, \mathbf{z})$ and dropping constants yields the weighted ridge update

$$\boldsymbol{\theta}^{t+1} = \arg\min_{\boldsymbol{\theta}} (\mathbf{y} - \mathbf{X}\boldsymbol{\theta})^\top \Lambda^{-1} (\mathbf{y} - \mathbf{X}\boldsymbol{\theta}) + \sum_{j=1}^{d} w_j \cdot \theta_j^2$$

with ridge weights given by $w_j = \gamma_j$, which admits a closed-form solution.

**Update for $\gamma_j$.** For fixed $(\boldsymbol{\theta}, \boldsymbol{\lambda}, \mathbf{z})$ minimizing each separable term dependent on $\boldsymbol{\gamma}$ yields the optimality condition

$$\frac{\partial}{\partial \gamma_j}(\gamma_j \cdot (\theta_j^2 + z_j) - \log \gamma_j) = (\theta_j^2 + z_j) - \frac{1}{\gamma_j} \overset{!}{=} 0 \;\Leftrightarrow\; \gamma_j^{t+1} = \frac{1}{\theta_j^2 + z_j} = \frac{1}{\theta_j^2 + [\Sigma_{\boldsymbol{\theta}}]_{jj}},$$

characterizing the same optimality condition as the EM update rule in § B.3 (with $\theta_j = \mu_{\boldsymbol{\theta},j}$ at convergence). To recover the iterative update found in Wipf & Nagarajan (2010), we employ the Woodbury identity on $\Sigma_{\boldsymbol{\theta}}$ to obtain

$$\Sigma_{\boldsymbol{\theta}} = (\Gamma + \mathbf{X}^\top \Lambda^{-1} \mathbf{X})^{-1} = \Gamma^{-1} - \Gamma^{-1}\mathbf{X}^\top(\Lambda + \mathbf{X}\Gamma^{-1}\mathbf{X}^\top)^{-1}\mathbf{X}\Gamma^{-1} = \Gamma^{-1} - \Gamma^{-1}\mathbf{X}^\top \Pi_{\mathbf{y}} \mathbf{X}\Gamma^{-1},$$

and taking the diagonal element we see that

$$[\Sigma_{\boldsymbol{\theta}}]_{jj} = [\Gamma^{-1} - \Gamma^{-1}\mathbf{X}^\top \Pi_{\mathbf{y}} \mathbf{X}\Gamma^{-1}]_{jj} = \gamma_j^{-1} - \gamma_j^{-2} \cdot q_j, \quad q_j = \mathbf{x}_j^\top \Pi_{\mathbf{y}} \mathbf{x}_j$$

where $\mathbf{x}_j$ denotes the $j$-th column of $\mathbf{X}$. Plugging in the expression we obtain the update

$$\gamma_j^{t+1} = (\theta_j^2 + [\Sigma_{\boldsymbol{\theta}}]_{jj})^{-1} = (\theta_j^2 + (\gamma_j^t)^{-1} - (\gamma_j^t)^{-2} \cdot q_j)^{-1},$$

which matches the update rule found in Wipf & Nagarajan (2010), Eq. 29. Note that above steps do not rely on a particular parametrization of $\boldsymbol{\lambda}$, permitting the same IRLS-style updates to hold in the heteroscedastic case.

**Update for $\lambda_i$.** Based on characterization of the same optimality conditions as the EM update rules, we start from a true stationary point of $\mathcal{L}(\boldsymbol{\gamma}, \boldsymbol{\lambda})$ and apply the Woodbury identity to $\Sigma_{\mathbf{y}}$ (as also done in § B.2) to obtain the expression

$$[\Pi_{\mathbf{y}}]_{ii} = [\Lambda^{-1} - \Lambda^{-1}\mathbf{X}\Sigma_{\boldsymbol{\theta}}\mathbf{X}^\top \Lambda^{-1}]_{ii} = \lambda_i^{-1} - \lambda_i^{-2} \cdot q_i, \quad q_i = \mathbf{x}_i^\top \Sigma_{\boldsymbol{\theta}} \mathbf{x}_i$$

where $\mathbf{x}_i^\top$ indicates the $i$-th row of $\mathbf{X}$. Thus $q_i = \lambda_i - \lambda_i^2 [\Pi_{\mathbf{y}}]_{ii}$ plugged into the EM update yields

$$\lambda_i^{t+1} = (y_i - \mathbf{x}_i^\top \boldsymbol{\theta})^2 + q_i = (y_i - \mathbf{x}_i^\top \boldsymbol{\theta})^2 + \lambda_i^t - (\lambda_i^t)^2 [\Pi_{\mathbf{y}}]_{ii}$$

as an iterative heteroscedastic update.

**Update for $\lambda$ in the homoscedastic case.** As for EM, the homoscedastic case collapses to a simple sample-averaged estimate, which using the above expression for $q_i$ is given by

$$\lambda^{t+1} = \frac{1}{n}\left(\|\mathbf{y} - \mathbf{X}\boldsymbol{\theta}\|_2^2 + \lambda^t - (\lambda^t)^2 \cdot \mathrm{trace}(\Pi_{\mathbf{y}})\right) = \frac{1}{n}\sum_{i=1}^{n}(y_i - \mathbf{x}_i^\top \boldsymbol{\theta})^2 + \lambda^t - \frac{(\lambda^t)^2}{n}\sum_{i=1}^{n}[\Pi_{\mathbf{y}}]_{ii}.$$

### B.6. Update rules for $\ell_1$-IRLS

As for the $\ell_2$ case, a majorization-minimization strategy on an upper-bounding surrogate objective is employed. First, using Wipf & Nagarajan (2010) we observe the same IRLS formulation of $\mathcal{L}(\boldsymbol{\gamma}, \boldsymbol{\lambda})$ as

$$\mathcal{L}(\boldsymbol{\gamma}, \boldsymbol{\lambda}) = (\mathbf{y} - \mathbf{X}\boldsymbol{\theta})^\top \Lambda^{-1}(\mathbf{y} - \mathbf{X}\boldsymbol{\theta}) + g_{\mathrm{SBL}}(\boldsymbol{\theta}),$$

where $g_{\mathrm{SBL}}(\boldsymbol{\theta}) \equiv \min_{\boldsymbol{\gamma} \geq 0}\{\boldsymbol{\theta}^\top \Gamma \boldsymbol{\theta} + \log|\Sigma_{\mathbf{y}}|\}$ forms a non-separable penalty term. $g_{\mathrm{SBL}}(\boldsymbol{\theta})$ is componentwise non-decreasing and concave in $|\boldsymbol{\theta}|$ (as we show more explicitly below), and thus permits minimization by iterative $\ell_1$-reweighted least squares.

**Obtaining a separable upper bound.** In contrast to the setting in Wipf & Nagarajan (2010) where $\boldsymbol{\lambda}$ is ignored, we now desire a tractable and separable upper bound on $g_{\mathrm{SBL}}(\boldsymbol{\theta})$ that induces $\ell_1$-regularization terms in both $\boldsymbol{\gamma}, \boldsymbol{\lambda}$, achievable by similarly majorizing the concave term $\log|\Sigma_{\mathbf{y}}|$ with an affine Fenchel bound. For the concave function $f(\mathbf{X}) = \log|\mathbf{X}|$ on a symmetric positive-definite matrix $\mathbf{X}$ of size $n$, we leverage the Fenchel identity

$$\log|\mathbf{X}| = \min_{\mathbf{P} \succ 0}\{\mathrm{trace}(\mathbf{P}\mathbf{X}) - \log|\mathbf{P}| - n\},$$

whose minimizer is $\mathbf{P}^* = \mathbf{X}^{-1}$ (Boyd & Vandenberghe, 2004). This follows directly from the definition of the concave conjugate of $f(\mathbf{X})$ as $f^*(\mathbf{P}) = \min_{\mathbf{X} \succ 0}\{\langle \mathbf{P}, \mathbf{X}\rangle - f(\mathbf{X})\}$ with minimizer $\mathbf{X}^* = \mathbf{P}^{-1}$, yielding $f^*(\mathbf{P}) = n + \log|\mathbf{P}|$;

and the equivalent relation $f(\mathbf{X}) = \min_{\mathbf{P} \succ 0} \{\langle \mathbf{P}, \mathbf{X} \rangle - f^*(\mathbf{P})\}$ made explicit. Thus for any fixed $\mathbf{P} \succ 0$, we obtain the upper bound $\log |\mathbf{X}| \leq \text{trace}(\mathbf{PX}) - \log |\mathbf{P}| - n$, with equality at $\mathbf{P}^*$. Applied to $\mathbf{X} = \Sigma_{\mathbf{y}} = \Lambda + \mathbf{X}\,\Gamma^{-1}\,\mathbf{X}^\top$ and $\mathbf{P} = \Sigma_{\mathbf{y}}^{-1} = \Pi_{\mathbf{y}}$, we obtain a tight linear upper bound up to constants[5] given by

$$\log |\Sigma_{\mathbf{y}}| \leq \text{trace}(\Pi_{\mathbf{y}}\,\Lambda) + \text{trace}(\Pi_{\mathbf{y}}\,\mathbf{X}\,\Gamma^{-1}\,\mathbf{X}^\top) = \sum_{i=1}^{n} \left([\Pi_{\mathbf{y}}]_{ii} \cdot \lambda_i\right) + \sum_{j=1}^{d} \left([\mathbf{X}^\top\,\Pi_{\mathbf{y}}\,\mathbf{X}]_{jj} \cdot \gamma_j^{-1}\right) = \sum_{i=1}^{n} z_i \cdot \lambda_i + \sum_{j=1}^{d} q_j \cdot \gamma_j^{-1},$$

with $z_i = [\Pi_{\mathbf{y}}]_{ii}$ and $q_j = \mathbf{x}_j^\top\,\Pi_{\mathbf{y}}\,\mathbf{x}_j$. Note that parameters $\boldsymbol{\gamma}, \boldsymbol{\lambda}$ are now separable, and the coefficients $z_i, q_j$ match the expressions obtained by separately tracing the Fenchel optimality condition for each parameter, *i.e.*

$$z_i^{t+1} = \frac{\partial \log |\Sigma_{\mathbf{y}}|}{\partial \lambda_i} = \text{trace}(\Pi_{\mathbf{y}}\,\mathbf{E}_{ii}) = [\Pi_{\mathbf{y}}]_{ii}, \qquad q_j^{t+1} = \frac{\partial \log |\Sigma_{\mathbf{y}}|}{\partial \gamma_j^{-1}} = \text{trace}(\Pi_{\mathbf{y}}\,\frac{\partial \Sigma_{\mathbf{y}}}{\partial \gamma_j^{-1}}) = \mathbf{x}_j^\top\,\Pi_{\mathbf{y}}\,\mathbf{x}_j.$$

Plugging the expression into $g_{\text{SBL}}(\boldsymbol{\theta})$, a final separable upper bound is obtained as

$$g_{\text{SBL}}(\boldsymbol{\theta}) \leq \boldsymbol{\theta}^\top\,\Gamma\,\boldsymbol{\theta} + \log |\Sigma_{\mathbf{y}}| \leq \sum_{i=1}^{n} z_i \cdot \lambda_i + \sum_{j=1}^{d} \left(\theta_j^2 \cdot \gamma_j + q_j \cdot \gamma_j^{-1}\right),$$

yielding the surrogate

$$\mathcal{L}^{\text{IRLS}}(\boldsymbol{\gamma}, \boldsymbol{\lambda}; \mathbf{z}, \mathbf{q}) = (\mathbf{y} - \mathbf{X}\,\boldsymbol{\theta})^\top\,\Lambda^{-1}\,(\mathbf{y} - \mathbf{X}\,\boldsymbol{\theta}) + \sum_{i=1}^{n} z_i \cdot \lambda_i + \sum_{j=1}^{d} \left(\theta_j^2 \cdot \gamma_j + q_j \cdot \gamma_j^{-1}\right) \geq \mathcal{L}(\boldsymbol{\gamma}, \boldsymbol{\lambda}).$$

**Update for $\gamma_j$.** Fixing all other parameters, minimization of each dependent term on $\boldsymbol{\gamma}$ yields the optimality condition

$$\frac{\partial}{\partial \gamma_j}(\theta_j^2 \cdot \gamma_j + q_j \cdot \gamma_j^{-1}) = \theta_j^2 - \frac{q_j}{\gamma_j^2} \overset{!}{=} 0 \Leftrightarrow \gamma_j^{t+1} = \sqrt{\frac{q_j}{\theta_j^2}} = \frac{\sqrt{q_j}}{|\theta_j|}.$$

The corresponding weight-side contribution of $g_{\text{SBL}}(\boldsymbol{\theta})$ is upper-bounded by the $\ell_1$ penalty

$$\min_{\boldsymbol{\gamma} \geq 0} \left\{ \sum_{j=1}^{d} \left(\theta_j^2 \cdot \gamma_j + q_j \cdot \gamma_j^{-1}\right) \right\} = \sum_{j=1}^{d} \left(\theta_j^2 \cdot \frac{\sqrt{q_j}}{|\theta_j|} + q_j \cdot \frac{|\theta_j|}{\sqrt{q_j}}\right) = 2 \cdot \sum_{j=1}^{d} \sqrt{q_j} \cdot |\theta_j|,$$

which is indeed non-decreasing and concave in each $|\theta_j|$. Leveraging a different upper bound dependent on $\boldsymbol{\gamma}$ only (and ignoring $\boldsymbol{\lambda}$), the same update rule for $\gamma_j^{t+1}$ can be found in Wipf & Nagarajan (2010), Eq. 32.

**Update for $\lambda_i$.** We first observe the appearence of $\boldsymbol{\lambda}$ in the quadratic data term. Denoting residuals $r_i = y_i - \mathbf{x}_i^\top\,\boldsymbol{\theta}$, we equivalently express the surrogate objective as

$$\mathcal{L}^{\text{IRLS}}(\boldsymbol{\gamma}, \boldsymbol{\lambda}; \mathbf{z}, \mathbf{q}) = \sum_{i=1}^{n} \left(r_i^2 \cdot \lambda_i^{-1} + z_i \cdot \lambda_i\right) + \sum_{j=1}^{d} \left(\theta_j^2 \cdot \gamma_j + q_j \cdot \gamma_j^{-1}\right).$$

To obtain a data-side $\ell_1$ penalty, we similarly fix other parameters and pool all terms dependent on $\boldsymbol{\lambda}$ to yield the optimality condition

$$\frac{\partial}{\partial \lambda_i}(r_i^2 \cdot \lambda_i^{-1} + z_i \cdot \lambda_i) = z_i - \frac{r_i^2}{\lambda_i^2} \overset{!}{=} 0 \Leftrightarrow \lambda_i^{t+1} = \sqrt{\frac{r_i^2}{z_i}} = \frac{|r_i|}{\sqrt{z_i}},$$

which results in the upper-bounding $\ell_1$ penalty

$$\min_{\boldsymbol{\lambda} \geq 0} \left\{ \sum_{i=1}^{n} \left(r_i^2 \cdot \lambda_i^{-1} + z_i \cdot \lambda_i\right) \right\} = \sum_{i=1}^{n} \left(r_i^2 \cdot \frac{\sqrt{z_i}}{|r_i|} + z_i \cdot \frac{|r_i|}{\sqrt{z_i}}\right) = 2 \cdot \sum_{i=1}^{n} \sqrt{z_i} \cdot |r_i|,$$

also non-decreasing and concave in $|r_i|$ and $|\theta_j|$. Note that in practice we alternate between updating $\boldsymbol{\theta}$ and $\boldsymbol{\lambda}$, which are respectively fixed at current iterates. Thus the quadratic data term is kept and the derived $\ell_1$ penalty is *added*, rather than eliminating $\boldsymbol{\lambda}$ in the same substep (which would result in a 'pure' $\ell_1$-only surrogate).

---

[5]$\mathbf{P}$ is held fixed within each iteration, hence $-\log |\mathbf{P}|$ acts as a constant and can be omitted during minimization.

**Update for $\boldsymbol{\theta}$.** Under fixed parameters, plugging in the above penalty expressions and minimizing the surrogate w.r.t. $\boldsymbol{\theta}$ then yields the double $\ell_1$-regularized update

$$\boldsymbol{\theta}^{t+1} = \arg\min_{\boldsymbol{\theta}} \left(\mathbf{y} - \mathbf{X}\,\boldsymbol{\theta}\right)^{\top} \Lambda^{-1} \left(\mathbf{y} - \mathbf{X}\,\boldsymbol{\theta}\right) + 2\sum_{j=1}^{d} w_j \cdot |\theta_j| \; + \; 2\sum_{i=1}^{n} v_i \cdot |r_i|,$$

with weights given by $w_j = \sqrt{q_j} = \sqrt{\mathbf{x}_j^{\top}\,\Pi_{\mathbf{y}}\,\mathbf{x}_j}$ and $v_i = \sqrt{z_i} = \sqrt{[\Pi_{\mathbf{y}}]_{ii}}$. Since the residual penalty is non-separable in $\theta_j$ there is no simple closed-form solution (as for the $\ell_2$ case), and the problem can be iteratively solved via split-variable or proximal gradient methods.

**Update for $\lambda$ in the homoscedastic case.** In the case where $\Lambda = \lambda\,\mathbf{I}_n$ the Fenchel bound yields a $\lambda$-dependent part of the surrogate as $\frac{1}{\lambda}\sum_{i=1}^{n} r_i^2 + \lambda\sum_{i=1}^{n} z_i$, and minimization yields the global update

$$\lambda^{t+1} = \sqrt{\frac{\sum_{i=1}^{n} r_i^2}{\sum_{i=1}^{n} z_i}} = \frac{\|\mathbf{y} - \mathbf{X}\,\boldsymbol{\theta}\|_2}{\sqrt{\mathrm{trace}(\Pi_{\mathbf{y}})}}.$$

Using an alternating update scheme, minimization of the surrogate w.r.t. $\boldsymbol{\theta}$ collapses to a weighted LASSO problem of the form

$$\boldsymbol{\theta}^{t+1} = \arg\min_{\boldsymbol{\theta}} \frac{1}{\lambda}\|\mathbf{y} - \mathbf{X}\,\boldsymbol{\theta}\|_2^2 + 2\sum_{j=1}^{d} w_j \cdot |\theta_j|,$$

with weights $w_j = \sqrt{q_j} = \sqrt{\mathbf{x}_j^{\top}\,\Pi_{\mathbf{y}}\,\mathbf{x}_j}$.

### B.7. Summary of update rules

Following derivations for each procedure, a summary of key parameter update rules is presented below in Tab. 5. We highlight that despite taking different perspectives on the target objective, all three methods of EM, MacKay, and $\ell_2$-IRLS recover the same update rules for $\boldsymbol{\gamma}$ and $\boldsymbol{\lambda}$ in the heteroscedastic case, bar algebraic arrangements (fixed-point for MacKay's $\gamma_j$, Woodbury identities for $\ell_2$-IRLS). This stresses the intuitive design of the emerging updates from taking a marginal likelihood perspective. Homoscedastic updates generally follow from collapsing individual components and employing algebraic simplifications, but take a similar, sample-averaged structural form to the heteroscedastic update.

*Table 5.* Summary table presenting the key parameter update rules across considered optimization procedures.

| Method | Objective | Update for $\gamma_j^{t+1}$ | Update for $\lambda_i^{t+1}$ (Heterosced.) | Update for $\lambda^{t+1}$ (Homosced.) |
|---|---|---|---|---|
| EM | $\log p(\mathbf{y}, \boldsymbol{\theta} \mid \boldsymbol{\gamma}, \boldsymbol{\lambda})$ | $(\mu_{\boldsymbol{\theta},j}^2 + [\Sigma_{\boldsymbol{\theta}}]_{jj})^{-1}$ | $(y_i - \mathbf{x}_i^{\top}\,\boldsymbol{\mu}_{\boldsymbol{\theta}})^2 + \mathbf{x}_i^{\top}\,\Sigma_{\boldsymbol{\theta}}\,\mathbf{x}_i$ | $\frac{1}{n}\left(\|\mathbf{y} - \mathbf{X}\,\boldsymbol{\mu}_{\boldsymbol{\theta}}\|_2^2 + \mathrm{trace}(\mathbf{X}\,\Sigma_{\boldsymbol{\theta}}\,\mathbf{X}^{\top})\right)$ |
| MacKay | $\mathcal{L}(\boldsymbol{\gamma}, \boldsymbol{\lambda})$ | $\frac{1 - \gamma_j^t \cdot [\Sigma_{\boldsymbol{\theta}}]_{jj}}{\mu_{\boldsymbol{\theta},j}^2}$ | $(y_i - \mathbf{x}_i^{\top}\,\boldsymbol{\mu}_{\boldsymbol{\theta}})^2 + \mathbf{x}_i^{\top}\,\Sigma_{\boldsymbol{\theta}}\,\mathbf{x}_i$ | $\frac{\|\mathbf{y} - \mathbf{X}\,\boldsymbol{\mu}_{\boldsymbol{\theta}}\|_2^2}{n - \sum_{j=1}^{d}\left(1 - \gamma_j^t \cdot [\Sigma_{\boldsymbol{\theta}}]_{jj}\right)}$ |
| IRLS ($\ell_2$) | $\mathcal{L}^{\mathrm{IRLS}}(\boldsymbol{\gamma}, \boldsymbol{\lambda}; \mathbf{z})$ | $(\theta_j^2 + (\gamma_j^t)^{-1} - (\gamma_j^t)^{-2} \cdot q_j)^{-1}$ | $(y_i - \mathbf{x}_i^{\top}\,\boldsymbol{\theta})^2 + \lambda_i^t - (\lambda_i^t)^2\,[\Pi_{\mathbf{y}}]_{ii}$ | $\frac{1}{n}\left(\|\mathbf{y} - \mathbf{X}\,\boldsymbol{\theta}\|_2^2 + \lambda^t - (\lambda^t)^2 \cdot \mathrm{trace}(\Pi_{\mathbf{y}})\right)$ |
| IRLS ($\ell_1$) | $\mathcal{L}^{\mathrm{IRLS}}(\boldsymbol{\gamma}, \boldsymbol{\lambda}; \mathbf{z}, \mathbf{q})$ | $\frac{\sqrt{\mathbf{x}_j^{\top}\,\Pi_{\mathbf{y}}\,\mathbf{x}_j}}{|\theta_j|}$ | $\frac{|y_i - \mathbf{x}_i^{\top}\,\boldsymbol{\theta}|}{\sqrt{[\Pi_{\mathbf{y}}]_{ii}}}$ | $\frac{\|\mathbf{y} - \mathbf{X}\,\boldsymbol{\theta}\|_2}{\sqrt{\mathrm{trace}(\Pi_{\mathbf{y}})}}$ |
| Grad. | $\mathcal{L}(\boldsymbol{\gamma}, \boldsymbol{\lambda})/\tilde{\mathcal{L}}(\boldsymbol{\gamma}, \boldsymbol{\lambda})$ | $\gamma_j^t - \eta_{\gamma}\,\frac{\partial \mathcal{L}}{\partial \gamma_j}$ | $\lambda_i^t - \eta_{\lambda}\,\frac{\partial \mathcal{L}}{\partial \lambda_i}$ | $\lambda^t - \eta_{\lambda}\,\frac{\partial \mathcal{L}}{\partial \lambda}$ |

### B.8. Variational Learning for Linear Regression

To expand the scope beyond strictly conjugate (Gaussian) and linear models, we consider embedding ARD parameter optimization as part of a larger step-wise optimization procedure utilizing *Variational Learning* (Khan & Rue, 2023; Khan, 2025). To that end, we take a first step by demonstrating that, for our standard Gaussian linear regression model, the variational objective precisely recovers the marginal likelihood given in Eq. 3. This provides a stepping stone and a natural avenue to extensions of the current procedure in future work.

**Notation.** Following above notation we describe the model and noise priors as $p(\boldsymbol{\theta} \mid \boldsymbol{\gamma}) = \mathcal{N}(\mathbf{0}, \Gamma^{-1})$ and $p(\boldsymbol{\epsilon} \mid \boldsymbol{\lambda}) = \mathcal{N}(\mathbf{0}, \Lambda)$, the data likelihood as $p(\mathbf{y} \mid \boldsymbol{\theta}, \boldsymbol{\lambda}) = \mathcal{N}(\mathbf{X}\boldsymbol{\theta}, \Lambda) = \prod_{i=1}^{n} \mathcal{N}(\mathbf{x}_i^{\top}\boldsymbol{\theta}, \lambda_i)$, and the marginal likelihood as $p(\mathbf{y} \mid \boldsymbol{\gamma}, \boldsymbol{\lambda}) = \mathcal{N}(\mathbf{0}, \Sigma_{\mathbf{y}})$. The exact posterior is given as $p(\boldsymbol{\theta} \mid \mathbf{y}, \boldsymbol{\gamma}, \boldsymbol{\lambda}) = \mathcal{N}(\boldsymbol{\mu}_{\boldsymbol{\theta}}, \Sigma_{\boldsymbol{\theta}})$, whereas the approximate variational posterior is denoted $q(\boldsymbol{\theta} \mid \mathbf{y}, \boldsymbol{\gamma}, \boldsymbol{\lambda}) = \mathcal{N}(\mathbf{m}, \mathbf{S})$, or $q(\boldsymbol{\theta})$ in short.

**Background.** $q(\boldsymbol{\theta}) \in \mathcal{Q}$ approximates $p(\boldsymbol{\theta} \mid \mathbf{y}, \boldsymbol{\gamma}, \boldsymbol{\lambda})$ and is rendered tractable by originating from the exponential family $\mathcal{Q}$, whose general p.d.f. takes the form $p(\boldsymbol{\theta}) = h(\boldsymbol{\theta}) \exp(\langle \boldsymbol{\eta}, \mathbf{T}(\boldsymbol{\theta}) \rangle - a(\boldsymbol{\eta}))$, with $\mathbf{T}(\boldsymbol{\theta}) = [\boldsymbol{\theta}, \boldsymbol{\theta}\boldsymbol{\theta}^{\top}]$ the sufficient statistics. For our (full) Gaussian choice, $q(\boldsymbol{\theta})$ can be parametrized in different ways, namely by standard mean and covariance $\boldsymbol{\xi} = (\mathbf{m}, \mathbf{S})$, by *natural* parameters $\boldsymbol{\eta} = (\mathbf{S}^{-1}\mathbf{m}, -\frac{1}{2}\mathbf{S}^{-1})$, and by *expectation* parameters $\boldsymbol{\mu} = \mathbb{E}_{q(\boldsymbol{\theta})}[\mathbf{T}(\boldsymbol{\theta})] = (\mathbf{m}, \mathbf{S} + \mathbf{m}\mathbf{m}^{\top})$. These parameterizations are equivalent and map between each other, which can be leveraged to rewrite our target objective and therein quantities.

**The ELBO as target objective.** As common in variational inference, we consider the evidence lower bound (ELBO) as a surrogate maximization objective to the log-marginal likelihood. Since Eq. 3 targets *minimization* of its negative, we obtain the relation

$$\mathcal{L}(\boldsymbol{\gamma}, \boldsymbol{\lambda}) = -\log p(\mathbf{y} \mid \boldsymbol{\gamma}, \boldsymbol{\lambda}) \leq -\mathbb{E}_{q(\boldsymbol{\theta})}[\log p(\mathbf{y} \mid \boldsymbol{\theta}, \boldsymbol{\lambda})] + D_{\mathrm{KL}}[q(\boldsymbol{\theta}) \| p(\boldsymbol{\theta} \mid \boldsymbol{\gamma})] = \mathcal{L}^{\mathrm{ELBO}}(\boldsymbol{\gamma}, \boldsymbol{\lambda}).$$

Recovery of the *Bayesian learning rule* in generality (see Khan & Rue (2023), Eq. 2) is obtained by rewriting

$$-\mathbb{E}_{q(\boldsymbol{\theta})}[\log p(\mathbf{y} \mid \boldsymbol{\theta}, \boldsymbol{\lambda})] = -\mathbb{E}_{q(\boldsymbol{\theta})}\left[\log \prod_{i=1}^{n} p(y_i \mid \boldsymbol{\theta}, \lambda_i)\right] = \sum_{i=1}^{n} \mathbb{E}_{q(\boldsymbol{\theta})}[-\log p(y_i \mid \boldsymbol{\theta}, \lambda_i)] = \sum_{i=1}^{n} \mathbb{E}_{q(\boldsymbol{\theta})}[\ell_i(\boldsymbol{\theta})],$$

with $\ell_i(\boldsymbol{\theta}) = -\log p(y_i \mid \boldsymbol{\theta}, \lambda_i) = -\log \mathcal{N}(\mathbf{x}_i^{\top}\boldsymbol{\theta}, \lambda_i)$ the likelihood contribution as a per-sample loss term. Alternatively, using the posterior relation $p(\boldsymbol{\theta} \mid \mathbf{y}, \boldsymbol{\gamma}, \boldsymbol{\lambda}) = p(\mathbf{y} \mid \boldsymbol{\theta}, \boldsymbol{\lambda}) \cdot p(\boldsymbol{\theta} \mid \boldsymbol{\gamma}) / \mathcal{Z}(\boldsymbol{\gamma}, \boldsymbol{\lambda})$ the ELBO can be written as

$$\mathcal{L}^{\mathrm{ELBO}}(\boldsymbol{\gamma}, \boldsymbol{\lambda}) = D_{\mathrm{KL}}[q(\boldsymbol{\theta}) \| p(\boldsymbol{\theta} \mid \mathbf{y}, \boldsymbol{\gamma}, \boldsymbol{\lambda})] - \log \mathcal{Z}(\boldsymbol{\gamma}, \boldsymbol{\lambda}),$$

with $\mathcal{Z}(\boldsymbol{\gamma}, \boldsymbol{\lambda}) = p(\mathbf{y} \mid \boldsymbol{\gamma}, \boldsymbol{\lambda})$ the partition function independent of $\boldsymbol{\theta}$. Clearly, for our conjugate Gaussian setting where $\mathcal{Q}$ contains the true posterior, the optimal value $q^*(\boldsymbol{\theta}) = p(\boldsymbol{\theta} \mid \mathbf{y}, \boldsymbol{\gamma}, \boldsymbol{\lambda})$ minimizes the objective as $D_{\mathrm{KL}}[\cdot \| \cdot] = 0$, recovering the exact log-marginal likelihood.

**Reformulation of the ELBO.** We now aim to make the same relation more explicit by parametrizing the above ELBO in terms of natural parameters $\boldsymbol{\eta}$, as employed in variational learning. To that end, following Khan & Rue (2023) we may express $q(\boldsymbol{\theta})$—more specifically, the likelihood terms—using local *site functions* as

$$q(\boldsymbol{\theta}) \propto p(\boldsymbol{\theta} \mid \boldsymbol{\gamma}) \prod_{i=1}^{n} \exp(-t_i(\boldsymbol{\theta})) \text{ with sites } t_i(\boldsymbol{\theta}) = \langle \tilde{\nabla}_{\boldsymbol{\eta}} \mathbb{E}_{q(\boldsymbol{\theta})}[\ell_i(\boldsymbol{\theta})], \mathbf{T}(\boldsymbol{\theta}) \rangle,$$

where $\ell_i(\boldsymbol{\theta})$ as above[6]. Each site $t_i(\boldsymbol{\theta})$ is expressed as an inner product between sufficient statistics $\mathbf{T}(\boldsymbol{\theta})$ and $\tilde{\nabla}_{\boldsymbol{\eta}} \mathbb{E}_{q(\boldsymbol{\theta})}[\ell_i(\boldsymbol{\theta})]$, the *natural gradient* with respect to natural parameters $\boldsymbol{\eta}$ evaluated at $\mathbb{E}_{q(\boldsymbol{\theta})}[\ell_i(\boldsymbol{\theta})]$. This key quantity is rendered tractable later on, but for now consider general $t_i(\boldsymbol{\theta})$. We may then re-write the ELBO by expanding $D_{\mathrm{KL}}[\cdot \| \cdot]$ with the site parameterization of $q(\boldsymbol{\theta})$ to obtain

$$\begin{aligned}
\mathcal{L}^{\mathrm{ELBO}}(\boldsymbol{\gamma}, \boldsymbol{\lambda}) &= -\mathbb{E}_{q(\boldsymbol{\theta})}\left[\log \prod_{i=1}^{n} p(y_i \mid \boldsymbol{\theta}, \lambda_i)\right] + D_{\mathrm{KL}}[q(\boldsymbol{\theta}) \| p(\boldsymbol{\theta} \mid \boldsymbol{\gamma})] \\
&= -\mathbb{E}_{q(\boldsymbol{\theta})}\left[\log \prod_{i=1}^{n} p(y_i \mid \boldsymbol{\theta}, \lambda_i)\right] + \mathbb{E}_{q(\boldsymbol{\theta})}\left[\log \frac{q(\boldsymbol{\theta})}{p(\boldsymbol{\theta} \mid \boldsymbol{\gamma})}\right] \\
&= -\mathbb{E}_{q(\boldsymbol{\theta})}\left[\log \prod_{i=1}^{n} p(y_i \mid \boldsymbol{\theta}, \lambda_i)\right] + \mathbb{E}_{q(\boldsymbol{\theta})}\left[\log \frac{p(\boldsymbol{\theta} \mid \boldsymbol{\gamma}) \prod_{i=1}^{n} \exp(-t_i(\boldsymbol{\theta}))}{p(\boldsymbol{\theta} \mid \boldsymbol{\gamma}) \mathcal{Z}(\boldsymbol{\gamma}, \boldsymbol{\lambda})}\right] \\
&= \mathbb{E}_{q(\boldsymbol{\theta})}\left[\log \frac{\prod_{i=1}^{n} \exp(-t_i(\boldsymbol{\theta}))}{\prod_{i=1}^{n} p(y_i \mid \boldsymbol{\theta}, \lambda_i)}\right] - \mathbb{E}_{q(\boldsymbol{\theta})}[\log \mathcal{Z}(\boldsymbol{\gamma}, \boldsymbol{\lambda})] \\
&= c(\boldsymbol{\theta}) - \log \mathcal{Z}(\boldsymbol{\gamma}, \boldsymbol{\lambda}),
\end{aligned}$$

---

[6]Note that $\tilde{\nabla}_{\boldsymbol{\eta}} \mathbb{E}_{q(\boldsymbol{\theta})}[\ell_i(\boldsymbol{\theta})] = \tilde{\nabla}_{\boldsymbol{\eta}} \mathbb{E}_{q(\boldsymbol{\theta})}[-\log p(y_i \mid \boldsymbol{\theta}, \lambda_i)] = -\tilde{\nabla}_{\boldsymbol{\eta}} \mathbb{E}_{q(\boldsymbol{\theta})}[\log p(y_i \mid \boldsymbol{\theta}, \lambda_i)]$

where $c(\boldsymbol{\theta}) = \sum_{i=1}^{n} \mathbb{E}_{q(\boldsymbol{\theta})}\left[\log\frac{\exp(-t_i(\boldsymbol{\theta}))}{p(y_i \mid \boldsymbol{\theta}, \lambda_i)}\right]$ and $\mathcal{Z}(\boldsymbol{\gamma}, \boldsymbol{\lambda})$ forms the partition function of $q(\boldsymbol{\theta})$. The first term can also be re-arranged to $c(\boldsymbol{\theta}) = \sum_{i=1}^{n} \mathbb{E}_{q(\boldsymbol{\theta})}\left[-\log p(y_i \mid \boldsymbol{\theta}, \lambda_i) + \log\exp(-t_i(\boldsymbol{\theta}))\right] = \sum_{i=1}^{n} \mathbb{E}_{q(\boldsymbol{\theta})}\left[\ell_i(\boldsymbol{\theta}) - t_i(\boldsymbol{\theta})\right]$, and is interpretable as a *posterior correction* term (Khan, 2025). Thus reparametrization in terms of natural parameters is made apparent through $t_i(\boldsymbol{\theta})$, and the expression is rendered tractable if we can compute the natural gradients, as shown next.

**Approximation of natural gradients.** The natural gradient terms can be rendered computable in several steps, as detailed in Khan & Rue (2023). First, reparametrization from $\boldsymbol{\eta}$ to expectation parameters $\boldsymbol{\mu}$ simplifies natural gradients to render *standard* gradients. Next, expression in terms of $(\mathbf{m}, \mathbf{S})$ and use of Bonnet's and Price's theorems returns terms using first and second-order derivatives of $\boldsymbol{\theta}$. Finally, the delta method can be used to approximate the expectations by point estimates evaluated at the mean. That is, computability of the two natural gradients is given by the outlined steps as

$$
\begin{aligned}
\tilde{\nabla}_{\eta_1}\mathbb{E}_{q(\boldsymbol{\theta})}[\ell_i(\boldsymbol{\theta})] &\overset{\text{Reparam.}}{=} \nabla_{\mu_1}\mathbb{E}_{q(\boldsymbol{\theta})}[\ell_i(\boldsymbol{\theta})] \\
&\overset{\text{Reparam.}}{=} \nabla_{\mathbf{m}}\mathbb{E}_{q(\boldsymbol{\theta})}[\ell_i(\boldsymbol{\theta})] - 2\left[\nabla_{\mathbf{S}}\mathbb{E}_{q(\boldsymbol{\theta})}[\ell_i(\boldsymbol{\theta})]\right]\mathbf{m} \\
&\overset{\text{Bonnet's Thm.}}{=} \mathbb{E}_{q(\boldsymbol{\theta})}[\nabla_{\boldsymbol{\theta}}\ell_i(\boldsymbol{\theta})] - \mathbb{E}_{q(\boldsymbol{\theta})}[\nabla^2_{\boldsymbol{\theta}}\ell_i(\boldsymbol{\theta})]\mathbf{m} \\
&\overset{\text{Delta method}}{\approx} \nabla_{\boldsymbol{\theta}}\ell_i(\boldsymbol{\theta})|_{\boldsymbol{\theta}=\mathbf{m}} - \left[\nabla^2_{\boldsymbol{\theta}}\ell_i(\boldsymbol{\theta})|_{\boldsymbol{\theta}=\mathbf{m}}\right]\mathbf{m}, \\
\tilde{\nabla}_{\eta_2}\mathbb{E}_{q(\boldsymbol{\theta})}[\ell_i(\boldsymbol{\theta})] &\overset{\text{Reparam.}}{=} \nabla_{\mu_2}\mathbb{E}_{q(\boldsymbol{\theta})}[\ell_i(\boldsymbol{\theta})] \\
&\overset{\text{Reparam.}}{=} \nabla_{\mathbf{S}}\mathbb{E}_{q(\boldsymbol{\theta})}[\ell_i(\boldsymbol{\theta})] \\
&\overset{\text{Price's Thm.}}{=} \frac{1}{2}\mathbb{E}_{q(\boldsymbol{\theta})}[\nabla^2_{\boldsymbol{\theta}}\ell_i(\boldsymbol{\theta})] \\
&\overset{\text{Delta method}}{\approx} \frac{1}{2}\nabla^2_{\boldsymbol{\theta}}\ell_i(\boldsymbol{\theta})|_{\boldsymbol{\theta}=\mathbf{m}}.
\end{aligned}
$$

Thus, final approximations require merely gradient and Hessian evaluations at the variational posterior's mean $\mathbf{m}$, which are relatively straightforward to obtain.

**Recovering the marginal likelihood objective.** Given pratically computable terms, we can now show that for our choice of Gaussian variational posterior and Gaussian linear regression model the reformulated ELBO objective also exactly recovers the marginal likelihood. We start by explicitly computing the natural gradients using above approximations. For our Gaussian linear model we then have

$$
\ell_i(\boldsymbol{\theta}) = -\log p(y_i \mid \boldsymbol{\theta}, \lambda_i) = -\log\mathcal{N}(\mathbf{x}_i^\top\boldsymbol{\theta}, \lambda_i) = \frac{1}{2}\log(2\pi\lambda_i) + \frac{1}{2\lambda_i}(y_i - \mathbf{x}_i^\top\boldsymbol{\theta})^2,
$$

$$
\nabla_{\boldsymbol{\theta}}\ell_i(\boldsymbol{\theta}) = -\frac{1}{\lambda_i}(y_i - \mathbf{x}_i^\top\boldsymbol{\theta})\mathbf{x}_i, \qquad \nabla^2_{\boldsymbol{\theta}}\ell_i(\boldsymbol{\theta}) = \frac{1}{\lambda_i}\mathbf{x}_i\mathbf{x}_i^\top.
$$

To compute the natural gradients, we follow the rules outlined above. As $\ell_i(\boldsymbol{\theta})$ is quadratic in $\boldsymbol{\theta}$, the expectations of $\nabla_{\boldsymbol{\theta}}\ell_i(\boldsymbol{\theta})$ and $\nabla^2_{\boldsymbol{\theta}}\ell_i(\boldsymbol{\theta})$ under Gaussian $q(\boldsymbol{\theta})$ are exact (affine/constant), thus we can skip the delta approximation. Plugging in the derivatives then yields the natural gradient terms

$$
\tilde{\nabla}_{\eta_1}\mathbb{E}_{q(\boldsymbol{\theta})}[\ell_i(\boldsymbol{\theta})] = \mathbb{E}_{q(\boldsymbol{\theta})}\left[-\frac{1}{\lambda_i}(y_i - \mathbf{x}_i^\top\boldsymbol{\theta})\mathbf{x}_i\right] - \mathbb{E}_{q(\boldsymbol{\theta})}\left[\frac{1}{\lambda_i}\mathbf{x}_i\mathbf{x}_i^\top\right]\mathbf{m} = -\frac{1}{\lambda_i}(y_i - \mathbf{x}_i^\top\mathbf{m})\mathbf{x}_i - (\frac{1}{\lambda_i}\mathbf{x}_i\mathbf{x}_i^\top)\mathbf{m} = -\frac{y_i}{\lambda_i}\mathbf{x}_i,
$$

$$
\tilde{\nabla}_{\eta_2}\mathbb{E}_{q(\boldsymbol{\theta})}[\ell_i(\boldsymbol{\theta})] = \frac{1}{2}\mathbb{E}_{q(\boldsymbol{\theta})}\left[\frac{1}{\lambda_i}\mathbf{x}_i\mathbf{x}_i^\top\right] = \frac{1}{2\lambda_i}\mathbf{x}_i\mathbf{x}_i^\top.
$$

We can now verify a given site function to observe the form

$$
\log\exp(-t_i(\boldsymbol{\theta})) = -\langle\tilde{\nabla}_{\boldsymbol{\eta}}\mathbb{E}_{q(\boldsymbol{\theta})}[\ell_i(\boldsymbol{\theta})], \mathbf{T}(\boldsymbol{\theta})\rangle = -(-\frac{y_i}{\lambda_i}\mathbf{x}_i^\top\boldsymbol{\theta} + \frac{1}{2\lambda_i}\boldsymbol{\theta}^\top\mathbf{x}_i\mathbf{x}_i^\top\boldsymbol{\theta}) = \frac{y_i}{\lambda_i}\mathbf{x}_i^\top\boldsymbol{\theta} - \frac{1}{2\lambda_i}\boldsymbol{\theta}^\top\mathbf{x}_i\mathbf{x}_i^\top\boldsymbol{\theta},
$$

and expanding the quadratic likelihood term we also see that

$$
\begin{aligned}
\log p(y_i \mid \boldsymbol{\theta}, \lambda_i) &= -\frac{1}{2}\log(2\pi\lambda_i) - \frac{1}{2\lambda_i}(y_i - \mathbf{x}_i^\top\boldsymbol{\theta})^2 \\
&= -\frac{1}{2}\log(2\pi\lambda_i) - \frac{1}{2\lambda_i}(y_i^2 - 2y_i\mathbf{x}_i^\top\boldsymbol{\theta} + \boldsymbol{\theta}^\top\mathbf{x}_i\mathbf{x}_i^\top\boldsymbol{\theta}) \\
&= \left(-\frac{1}{2}\log(2\pi\lambda_i) - \frac{y_i^2}{2\lambda_i}\right) + \frac{y_i}{\lambda_i}\mathbf{x}_i^\top\boldsymbol{\theta} - \frac{1}{2\lambda_i}\boldsymbol{\theta}^\top\mathbf{x}_i\mathbf{x}_i^\top\boldsymbol{\theta} \\
&= C_i + \frac{y_i}{\lambda_i}\mathbf{x}_i^\top\boldsymbol{\theta} - \frac{1}{2\lambda_i}\boldsymbol{\theta}^\top\mathbf{x}_i\mathbf{x}_i^\top\boldsymbol{\theta},
\end{aligned}
$$

with $\boldsymbol{\theta}$-independent constant term $C_i$. Thus we immediately observe that $\exp(-t_i(\boldsymbol{\theta})) \propto p(y_i \mid \boldsymbol{\theta}, \lambda_i)$, and therefore $c(\boldsymbol{\theta}) = \sum_{i=1}^n \mathbb{E}_{q(\boldsymbol{\theta})}\left[\log\frac{\exp(-t_i(\boldsymbol{\theta}))}{p(y_i\mid\boldsymbol{\theta},\lambda_i)}\right] = 0$ up to constants $C_1, \ldots, C_n$. Since the sites parametrized by natural gradients match the true likelihood factors, $q(\boldsymbol{\theta})$ matches the optimal $q^*(\boldsymbol{\theta})$ whose factorization yields

$$
q^*(\boldsymbol{\theta}) \propto p(\boldsymbol{\theta} \mid \boldsymbol{\gamma}) \prod_{i=1}^n \exp(-t_i(\boldsymbol{\theta})) \propto p(\boldsymbol{\theta} \mid \boldsymbol{\gamma}) \prod_{i=1}^n p(y_i \mid \boldsymbol{\theta}, \lambda_i) = p(\boldsymbol{\theta}, \mathbf{y} \mid \boldsymbol{\gamma}, \boldsymbol{\lambda}),
$$

and whose corresponding partition function is $\mathcal{Z}(\boldsymbol{\gamma}, \boldsymbol{\lambda}) = p(\mathbf{y} \mid \boldsymbol{\gamma}, \boldsymbol{\lambda})$. It follows directly for the reformulated ELBO that $\mathcal{L}^{\text{ELBO}}(\boldsymbol{\gamma}, \boldsymbol{\lambda}) = c(\boldsymbol{\theta}) - \log\mathcal{Z}(\boldsymbol{\gamma}, \boldsymbol{\lambda}) = 0 - \log p(\mathbf{y} \mid \boldsymbol{\gamma}, \boldsymbol{\lambda}) = \mathcal{L}(\boldsymbol{\gamma}, \boldsymbol{\lambda})$ recovers the exact marginal likelihood objective (Eq. 3), affirming that variational learning is amenable to ARD parameter optimization.

# C. Algorithmic Details

---

**Algorithm 1** Joint ARD via Expectation Maximization

---

1: **Input:** Data $\mathbf{X}, \mathbf{y}$, initialized parameters $\boldsymbol{\gamma}^0, \boldsymbol{\lambda}^0$
2: **Output:** Estimated ARD parameters $\hat{\boldsymbol{\gamma}}, \hat{\boldsymbol{\lambda}}$ and posterior parameters $\hat{\boldsymbol{\mu}}_{\boldsymbol{\theta}}, \hat{\Sigma}_{\boldsymbol{\theta}}$
   ▷ Run until convergence
3: **for** $t = 0, \ldots, T-1$ **do**
4:    Set $\Gamma^t \leftarrow \mathrm{diag}(\boldsymbol{\gamma}^t)$, $\Lambda^t \leftarrow \mathrm{diag}(\boldsymbol{\lambda}^t)$
5:    Weight posterior at current parameters:        ▷ E-Step
$$\Sigma_{\boldsymbol{\theta}}^t \leftarrow (\Gamma^t + \mathbf{X}^\top (\Lambda^t)^{-1} \mathbf{X})^{-1}$$
$$\boldsymbol{\mu}_{\boldsymbol{\theta}}^t \leftarrow \Sigma_{\boldsymbol{\theta}}^t \mathbf{X}^\top (\Lambda^t)^{-1} \mathbf{y}$$
6:    Update parameters using posterior moments:        ▷ M-Step
$$\boldsymbol{\gamma}^{t+1} \leftarrow [(\boldsymbol{\mu}_{\boldsymbol{\theta}}^t)^2 + \mathrm{diag}(\Sigma_{\boldsymbol{\theta}}^t)]^{-1}$$
$$\boldsymbol{\lambda}^{t+1} \leftarrow (\mathbf{y} - \mathbf{X}\boldsymbol{\mu}_{\boldsymbol{\theta}}^t)^2 + \mathrm{diag}(\mathbf{X}\Sigma_{\boldsymbol{\theta}}^t \mathbf{X}^\top)$$
7: **end for**
8: Re-compute weight posterior at convergence: $\boldsymbol{\mu}_{\boldsymbol{\theta}}^{t+1}, \Sigma_{\boldsymbol{\theta}}^{t+1}$        ▷ Final assignments
9: Set $\hat{\boldsymbol{\gamma}} \leftarrow \boldsymbol{\gamma}^{t+1}$, $\hat{\boldsymbol{\lambda}} \leftarrow \boldsymbol{\lambda}^{t+1}$, $\hat{\boldsymbol{\mu}}_{\boldsymbol{\theta}} \leftarrow \boldsymbol{\mu}_{\boldsymbol{\theta}}^{t+1}$, $\hat{\Sigma}_{\boldsymbol{\theta}} \leftarrow \Sigma_{\boldsymbol{\theta}}^{t+1}$
10: **return** $\hat{\boldsymbol{\gamma}}, \hat{\boldsymbol{\lambda}}, \hat{\boldsymbol{\mu}}_{\boldsymbol{\theta}}, \hat{\Sigma}_{\boldsymbol{\theta}}$

---

---

**Algorithm 2** Joint ARD via MacKay's updates

---

1: **Input:** Data $\mathbf{X}, \mathbf{y}$, initialized parameters $\boldsymbol{\gamma}^0, \boldsymbol{\lambda}^0$
2: **Output:** Estimated ARD parameters $\hat{\boldsymbol{\gamma}}, \hat{\boldsymbol{\lambda}}$ and posterior parameters $\hat{\boldsymbol{\mu}}_{\boldsymbol{\theta}}, \hat{\Sigma}_{\boldsymbol{\theta}}$
   ▷ Run until convergence
3: **for** $t = 0, \ldots, T-1$ **do**
4:    Set $\Gamma^t \leftarrow \mathrm{diag}(\boldsymbol{\gamma}^t)$, $\Lambda^t \leftarrow \mathrm{diag}(\boldsymbol{\lambda}^t)$
5:    Weight posterior at current parameters:
$$\Sigma_{\boldsymbol{\theta}}^t \leftarrow (\Gamma^t + \mathbf{X}^\top (\Lambda^t)^{-1} \mathbf{X})^{-1}$$
$$\boldsymbol{\mu}_{\boldsymbol{\theta}}^t \leftarrow \Sigma_{\boldsymbol{\theta}}^t \mathbf{X}^\top (\Lambda^t)^{-1} \mathbf{y}$$
6:    Update parameters using rules:        ▷ EM updates with fixed-point cond.
$$\boldsymbol{\gamma}^{t+1} \leftarrow \frac{1 - \boldsymbol{\gamma}^t \cdot \mathrm{diag}(\Sigma_{\boldsymbol{\theta}}^t)}{(\boldsymbol{\mu}_{\boldsymbol{\theta}}^t)^2}$$
$$\boldsymbol{\lambda}^{t+1} \leftarrow (\mathbf{y} - \mathbf{X}\boldsymbol{\mu}_{\boldsymbol{\theta}}^t)^2 + \mathrm{diag}(\mathbf{X}\Sigma_{\boldsymbol{\theta}}^t \mathbf{X}^\top)$$
7: **end for**
8: Re-compute weight posterior at convergence: $\boldsymbol{\mu}_{\boldsymbol{\theta}}^{t+1}, \Sigma_{\boldsymbol{\theta}}^{t+1}$        ▷ Final assignments
9: Set $\hat{\boldsymbol{\gamma}} \leftarrow \boldsymbol{\gamma}^{t+1}$, $\hat{\boldsymbol{\lambda}} \leftarrow \boldsymbol{\lambda}^{t+1}$, $\hat{\boldsymbol{\mu}}_{\boldsymbol{\theta}} \leftarrow \boldsymbol{\mu}_{\boldsymbol{\theta}}^{t+1}$, $\hat{\Sigma}_{\boldsymbol{\theta}} \leftarrow \Sigma_{\boldsymbol{\theta}}^{t+1}$
10: **return** $\hat{\boldsymbol{\gamma}}, \hat{\boldsymbol{\lambda}}, \hat{\boldsymbol{\mu}}_{\boldsymbol{\theta}}, \hat{\Sigma}_{\boldsymbol{\theta}}$

---

---

**Algorithm 3** Joint ARD via $\ell_2$-IRLS

---

1: **Input:** Data $\mathbf{X}, \mathbf{y}$, initialized parameters $\boldsymbol{\gamma}^0, \boldsymbol{\lambda}^0$
2: **Output:** Estimated ARD parameters $\hat{\boldsymbol{\gamma}}, \hat{\boldsymbol{\lambda}}$ and posterior parameters $\hat{\boldsymbol{\mu}}_{\boldsymbol{\theta}}, \hat{\Sigma}_{\boldsymbol{\theta}}$
    ▷ Run until convergence
3: **for** $t = 0, \ldots, T-1$ **do**
4:    Set $\Gamma^t \leftarrow \text{diag}(\boldsymbol{\gamma}^t), \ \Lambda^t \leftarrow \text{diag}(\boldsymbol{\lambda}^t)$
5:    Data covariance at current parameters:
$$\Sigma_{\mathbf{y}}^t \leftarrow \Lambda^t + \mathbf{X}\,(\Gamma^t)^{-1}\,\mathbf{X}^{\top}$$
$$\Pi_{\mathbf{y}}^t \leftarrow (\Sigma_{\mathbf{y}}^t)^{-1}$$
6:    Surrogate ridge solution for weights:         ▷ Posterior mean with Woodbury id.
$$\boldsymbol{\theta}^{t+1} \leftarrow (\Gamma^t)^{-1}\,\mathbf{X}^{\top}\,\Pi_{\mathbf{y}}^t\,\mathbf{y}$$
7:    Update parameters using rules:         ▷ EM updates with Woodbury id.
$$\boldsymbol{\gamma}^{t+1} \leftarrow [(\boldsymbol{\theta}^{t+1})^2 + (\boldsymbol{\gamma}^t)^{-1} - (\boldsymbol{\gamma}^t)^{-2} \odot \text{diag}(\mathbf{X}^{\top}\,\Pi_{\mathbf{y}}^t\,\mathbf{X})]^{-1}$$
$$\boldsymbol{\lambda}^{t+1} \leftarrow (\mathbf{y} - \mathbf{X}\,\boldsymbol{\theta}^{t+1})^2 + \boldsymbol{\lambda}^t - (\boldsymbol{\lambda}^t)^2 \odot \text{diag}(\Pi_{\mathbf{y}}^t)$$
8: **end for**
9: Compute weight posterior at convergence: $\boldsymbol{\mu}_{\boldsymbol{\theta}}^{t+1}, \Sigma_{\boldsymbol{\theta}}^{t+1}$        ▷ Final assignments
10: Set $\hat{\boldsymbol{\gamma}} \leftarrow \boldsymbol{\gamma}^{t+1}, \ \hat{\boldsymbol{\lambda}} \leftarrow \boldsymbol{\lambda}^{t+1}, \ \hat{\boldsymbol{\mu}}_{\boldsymbol{\theta}} \leftarrow \boldsymbol{\mu}_{\boldsymbol{\theta}}^{t+1}, \ \hat{\Sigma}_{\boldsymbol{\theta}} \leftarrow \Sigma_{\boldsymbol{\theta}}^{t+1}$
11: **return** $\hat{\boldsymbol{\gamma}}, \hat{\boldsymbol{\lambda}}, \hat{\boldsymbol{\mu}}_{\boldsymbol{\theta}}, \hat{\Sigma}_{\boldsymbol{\theta}}$

---

---

**Algorithm 4** Joint ARD via $\ell_1$-IRLS

---

1: **Input:** Data $\mathbf{X}, \mathbf{y}$, initialized parameters $\boldsymbol{\gamma}^0, \boldsymbol{\lambda}^0$
2: **Output:** Estimated ARD parameters $\hat{\boldsymbol{\gamma}}, \hat{\boldsymbol{\lambda}}$ and posterior parameters $\hat{\boldsymbol{\mu}}_{\boldsymbol{\theta}}, \hat{\Sigma}_{\boldsymbol{\theta}}$
    ▷ Run until convergence
3: **for** $t = 0, \ldots, T-1$ **do**
4:    Set $\Gamma^t \leftarrow \text{diag}(\boldsymbol{\gamma}^t), \ \Lambda^t \leftarrow \text{diag}(\boldsymbol{\lambda}^t)$
5:    Data covariance at current parameters:
$$\Sigma_{\mathbf{y}}^t \leftarrow \Lambda^t + \mathbf{X}\,(\Gamma^t)^{-1}\,\mathbf{X}^{\top}$$
$$\Pi_{\mathbf{y}}^t \leftarrow (\Sigma_{\mathbf{y}}^t)^{-1}$$
6:    Weights for $\ell_1$-regularized surrogate:         ▷ From auxiliary variables
$$\mathbf{w}^t \leftarrow \sqrt{\text{diag}(\mathbf{X}^{\top}\,\Pi_{\mathbf{y}}^t\,\mathbf{X})}$$
$$\mathbf{v}^t \leftarrow \sqrt{\text{diag}(\Pi_{\mathbf{y}}^t)}$$
7:    Surrogate solution for weights:    ▷ Run inner loop for convex solver (*e.g.* ADMM)
$$\boldsymbol{\theta}^{t+1} = \arg\min_{\boldsymbol{\theta}} (\mathbf{y} - \mathbf{X}\,\boldsymbol{\theta})^{\top}\,(\Lambda^t)^{-1}\,(\mathbf{y} - \mathbf{X}\,\boldsymbol{\theta}) + 2\,\langle \mathbf{w}^t, |\boldsymbol{\theta}| \rangle + 2\,\langle \mathbf{v}^t, |\mathbf{y} - \mathbf{X}\,\boldsymbol{\theta}| \rangle$$
8:    Update parameters using map-back rules:
$$\boldsymbol{\gamma}^{t+1} \leftarrow \frac{\mathbf{w}^t}{|\boldsymbol{\theta}^{t+1}|}$$
$$\boldsymbol{\lambda}^{t+1} \leftarrow \frac{|\mathbf{y} - \mathbf{X}\,\boldsymbol{\theta}^{t+1}|}{\mathbf{v}^t}$$
9: **end for**
10: Compute weight posterior at convergence: $\boldsymbol{\mu}_{\boldsymbol{\theta}}^{t+1}, \Sigma_{\boldsymbol{\theta}}^{t+1}$        ▷ Final assignments
11: Set $\hat{\boldsymbol{\gamma}} \leftarrow \boldsymbol{\gamma}^{t+1}, \ \hat{\boldsymbol{\lambda}} \leftarrow \boldsymbol{\lambda}^{t+1}, \ \hat{\boldsymbol{\mu}}_{\boldsymbol{\theta}} \leftarrow \boldsymbol{\mu}_{\boldsymbol{\theta}}^{t+1}, \ \hat{\Sigma}_{\boldsymbol{\theta}} \leftarrow \Sigma_{\boldsymbol{\theta}}^{t+1}$
12: **return** $\hat{\boldsymbol{\gamma}}, \hat{\boldsymbol{\lambda}}, \hat{\boldsymbol{\mu}}_{\boldsymbol{\theta}}, \hat{\Sigma}_{\boldsymbol{\theta}}$

---

# D. Additional Experiment Details

The code to reproduce experiments is made available at <https://github.com/alextimans/robust-sbl>. Additional clarifying details are given below.

**Effective support size (ESS).**   Given generic nonnegative relevance scores $\{r_i\}_{i=1}^m$, *i.e.* $r_i = 1/\gamma_i$ for weight relevance and $r_i = 1/\lambda_i$ for data relevance, we first convert them into a probability mass function by normalization,

$$p_i = \frac{r_i}{\sum_{j=1}^m r_j},$$

optionally using a small $\epsilon > 0$ and clipping for numerical stability. We then compute the Shannon entropy

$$H(p) = -\sum_{i=1}^m p_i \log(p_i)$$

quantifying how dispersed the relevance mass is. Exponentiating yields *perplexity* as $N_{\text{eff}} = \exp(H(p))$, denoting an effective number of active elements, and we report its normalized quantity $\text{ESS} = N_{\text{eff}}/m \in (0, 1]$ as the *effective support size*. ESS approaches 1 for uniform relevance and becomes small when relevance concentrates on few elements, matching our sparsity-inducing interpretation for $(\boldsymbol{\gamma}, \boldsymbol{\lambda})$. Finally, entropy-based ESS is related but distinct from the *effective sample size* commonly used in Monte Carlo methods and importance sampling: our definition corresponds to the Shannon (order-1) effective number, whereas a popular alternative is the order-2 quantity $N_{\text{eff}}^{(2)} = 1/\sum_i p_i^2$ (Martino et al., 2017).

**Posterior predictive noise estimate.**   The posterior predictive distribution for a new test input $\mathbf{x}_* \in \mathbb{R}^{d \times 1}$ is given by

$$p(\mathbf{y}_* \mid \mathbf{x}_*, \mathbf{X}, \mathbf{y}) = \mathcal{N}(\mathbf{y}_* \mid \mu_*, \lambda_*) \quad \text{with} \quad \mu_* = \mathbf{x}_*^\top \boldsymbol{\mu_\theta} \quad \text{and} \quad \lambda_* = \lambda_b + \mathbf{x}_*^\top \Sigma_{\boldsymbol{\theta}} \, \mathbf{x}_*$$

denoting posterior predictive mean and variance, respectively. In particular, the latter requires selecting a base noise level $\lambda_b$ that summarizes expected observation noise at test time, and which is *a priori* unknown. Assuming no distribution shift, we use a plug-in estimate based on the learned training sample variances $\boldsymbol{\lambda}$ via data ARD. Setting $\lambda_b = \text{mean}(\boldsymbol{\lambda})$ is conservative in the sense that it aggregates across all samples, thus a small number of large $\lambda_i$ values (flagged as unreliable by data ARD) can only *increase* $\lambda_b$ and hence inflate predictive uncertainty rather than overstate confidence. A useful alternative is to employ a trimmed mean plug-in estimate to reduce sensitivity to extreme values when data contamination is subsumed, *i.e.* $\lambda_b = \text{mean}(\{\lambda_i : \lambda_i \in [q_\alpha, q_{1-\alpha}]\})$ where $q_\alpha$ and $q_{1-\alpha}$ may be pre-selected fixed empirical quantiles of $\{\lambda_i\}_{i=1}^n$ (*e.g.* 5% and 95%). Importantly, both choices are computed solely from learned variances $\boldsymbol{\lambda}$ and do not require any outlier ground truth labels or even prior knowledge of the contamination rate.

## D.1. Practical implementation of update rules

Stable learning of closed-form SBL updates benefits from several practical safeguards. First, all matrix operations required for posterior moments (inversions, diagonals, and quadratic forms) are implemented via Cholesky factorization and triangular solves, yielding both improved numerical stability and computational efficiency over explicit inversion. Second, we damp iterative parameter updates using an exponential moving average as

$$\boldsymbol{\gamma}^{t+1} \leftarrow (1 - \eta_\gamma)\boldsymbol{\gamma}^t + \eta_\gamma \boldsymbol{\gamma}_{\text{new}}^{t+1}, \qquad \boldsymbol{\lambda}^{t+1} \leftarrow (1 - \eta_\lambda)\boldsymbol{\lambda}^t + \eta_\lambda \boldsymbol{\lambda}_{\text{new}}^{t+1}, \qquad \eta_\gamma \in (0, 1], \eta_\lambda \in (0, 1],$$

which improves conditioning and smooths optimization trajectories, analogous to the learning rate in gradient-based schemes. In practice we found strong damping in the range of $\eta \in [5 \cdot 10^{-4}, 2 \cdot 10^{-2}]$ to be necessary. Third, we clip extreme values to preserve positivity and avoid ill-conditioning, *e.g.* $\epsilon_{\min} \in [10^{-6}, 10^{-3}]$ and $\epsilon_{\max} \in [10^2, 10^6]$, and add a small magnitude-adaptive jitter to $\Sigma_{\mathbf{y}}$ and $\Sigma_{\boldsymbol{\theta}}$ when required for robust factorization. Fourth, optimization convergence is monitored via a relative $\ell_\infty$ change criterion on log-parameters for scale-invariance,

$$\frac{\|\log \boldsymbol{\gamma}^{t+1} - \log \boldsymbol{\gamma}^t\|_\infty}{1 + \|\log \boldsymbol{\gamma}^{t+1}\|_\infty} < \epsilon_{\text{rel}} \quad \text{and} \quad \frac{\|\log \boldsymbol{\lambda}^{t+1} - \log \boldsymbol{\lambda}^t\|_\infty}{1 + \|\log \boldsymbol{\lambda}^{t+1}\|_\infty} < \epsilon_{\text{rel}},$$

combined with a small patience window ($\epsilon_{\text{rel}} = 10^{-6}$ for 5 patience steps). Empirically, we found this to be more reliable than loss-based stopping on the objective. For $\ell_1$-IRLS we warm-start the inner convex subproblem (solved via ADMM)

from the current iterate $\boldsymbol{\theta}^t$ rather than re-optimizing from scratch. Finally, we also warm-start data ARD by initially updating only $\boldsymbol{\gamma}$ for several iterations before enabling $\boldsymbol{\lambda}$ (in the range of $50-300$ steps), and subsequently updating $\boldsymbol{\lambda}$ only every $K \in [2, 5]$ outer iterations (rather than at every step), which yields a stronger initial fit and reduces the risk of overfit.

Introducing heteroscedastic noise expands the parameterization from a single scalar noise level to $n$ additional free variance parameters, substantially increasing flexibility and, with it, the risk of overfitting. In particular, the model risks prematurely 'explaining away' residual structure by inflating individual $\lambda_i$ values instead of improving the predictive mean, a well-known tendency in heteroscedastic regression that can be exacerbated in highly expressive models (Cawley & Talbot, 2010; Grünwald et al., 2017; Wong-Toi et al., 2024). As a result, the learning dynamics of joint ARD can be sensitive to hyperparameters and numerical choices, and the above regularization mechanisms (damping, clipping, jittering, and robust stopping) are often necessary to obtain well-conditioned posteriors and meaningful convergence results in practice. Strongly misspecified priors can have a notable impact on convergence speed and recovery, an effect more pronounced for noise variances $\boldsymbol{\lambda}$ (which live on the residual scale) and less so for weight precisions $\boldsymbol{\gamma}$ (which live on their own scale). Nonetheless, we found weakly-informative choices to consistently work well across different algorithms and experiments, which in practice may look like a fixed scalar initial value (*e.g.* $s = 0.1$) across parameters.

### D.2. Experiment design protocols

**Synthetic experiment (§ 5.1).** We generate linear regression data with sparse ground-truth weights and uncorrelated Gaussian features. For each trial, we sample a support set $S \subseteq \{1, \dots, d\}$ uniformly at random and draw nonzero weights $\theta_j \sim \mathcal{N}(0, 1)$ for $j \in S$, with $\theta_j = 0$ otherwise. Features are sampled *i.i.d.* as $\mathbf{X} \in \mathbb{R}^{n \times d}$ with rows $\mathbf{x}_i \sim \mathcal{N}(\mathbf{0}, \mathbf{I})$. Observation noise is Gaussian with a base scale $\sigma$. We then sample an outlier index set $\mathcal{O} \subset \{1, \dots, n\}$ uniformly without replacement of size $|\mathcal{O}| = \lfloor \rho \cdot n \rfloor$, where $\rho \in [0, 1]$ is the contamination fraction. This set is assigned inflated variance by a multiplier $m$. The targets are then generated as

$$y_i = \mathbf{x}_i^\top \boldsymbol{\theta} + \epsilon_i, \qquad \epsilon_i \sim \begin{cases} \mathcal{N}(0, \sigma^2), & i \notin \mathcal{O}, \\ \mathcal{N}(0, (m\sigma)^2), & i \in \mathcal{O}. \end{cases}$$

We return $(\mathbf{X}, \mathbf{y})$ together with the ground-truth supports $S$ and $\mathcal{O}$ for evaluation. To generate the weight recovery heatmap in Fig. 3 we fix $\rho = 0.2, m = 10$ and vary $|S|$ and $\sigma$, whereas to generate the data recovery heatmap we fix $|S| = 0.2 \cdot d, \sigma = 0.2$ and vary $\rho$ and $m$. Throughout, $n = 500, d = 50$ remain fixed and test samples $n_{\text{test}} = 1000$ for predictive evaluation are generated entirely uncorrupted.

**Tabular regression benchmarks (§ 5.2).** Training targets $\mathbf{y}$ are standardized to zero mean and unit variance. We then sample an outlier index set $\mathcal{O} \subset \{1, \dots, n\}$ uniformly without replacement of size $|\mathcal{O}| = \lfloor \rho \cdot n \rfloor$, where $\rho \in [0, 1]$ is the contamination fraction. For each $i \in \mathcal{O}$ we draw a sign $s_i \in \{-1, +1\}$ *i.i.d.* Rademacher and add a signed perturbation of amplitude $a > 0$:

$$\tilde{y}_i = y_i + a\, s_i\, \delta_i, \qquad i \in \mathcal{O},$$

where $\delta_i \sim \mathcal{N}(1, 0.25^2)$, yielding heterogeneous outlier magnitudes around $a$ while most $\tilde{y}_i = y_i$ for $i \notin \mathcal{O}$ remain uncorrupted. The outlier amplitude is fixed at $a = 3.0$, and we report results for $\rho = 0$ (uncontaminated) and $\rho = 0.1$ (mild contamination).

The evaluated datasets in original sample size and feature dimension are: *Boston* ($n = 506, d = 13$), *Carbon* ($n = 10721, d = 5$), *Concrete* ($n = 1030, d = 8$), *Elevators* ($n = 16599, d = 18$), *Energy* ($n = 768, d = 8$), *Kin8nm* ($n = 8192, d = 8$), *Power* ($n = 9568, d = 4$), *Protein* ($n = 45730, d = 9$), and *Yacht* ($n = 308, d = 6$). We consistently randomly split 20% for hold-out test evaluation (uncontaminated) and cap datasets with large training splits to a fixed maximum $n_{\max} = 2000$ for computational efficiency (again, sampled at random). Otherwise we use the full training set. For datasets with multiple targets (*Carbon* with 3 targets, *Energy* with 2 targets) we only regress on the first target and omit the others.

**Sparse kernel regression (§ 5.3).** The experiment follows the same general protocol as for tabular regression above, the key difference being a switch from random Fourier features to a kernel-based feature matrix $\boldsymbol{\Phi} \in \mathbb{R}^{n \times n}$ instantiating the RVM.

**Neural network regression (§ 5.4).** Denoting the log-transformed count label as $z_i = \log(1 + y_i)$, we form a high-count pool $\mathcal{P}$ as the indices of the top $\lfloor \rho_{\text{pool}} \cdot n \rfloor$ values of $\{z_i\}_{i=1}^n$, and sample an outlier set $\mathcal{O} \subseteq \mathcal{P}$ uniformly at random with

$|\mathcal{O}| = \lfloor \rho_{\text{out}} \cdot |\mathcal{P}| \rfloor$. We then contaminate targets in log-space additively via

$$\tilde{z}_i \;=\; z_i + \epsilon_i, \qquad \epsilon_i \sim \begin{cases} \mathcal{N}(0, \sigma_{\text{in}}^2), & i \notin \mathcal{O}, \\ \mathcal{N}(0, \sigma_{\text{out}}^2), & i \in \mathcal{O}. \end{cases}$$

The pool fraction is fixed at $\rho_{\text{pool}} = 0.4$ and the contamination fraction at $\rho_{\text{out}} = 0.5$, yielding an effective contamination rate of $\rho_{\text{pool}} \cdot \rho_{\text{out}} = 0.2$. The inlier scale is set to $\sigma_{\text{in}} = 0.0$ (no contamination) while the outlier scale is fixed at $\sigma_{\text{out}} = 0.45$.

Regarding the *ShanghaiTech* dataset, we only make use of the consistent 'part B' set ($n = 716$) to avoid test-time distribution shifts caused by web-crawled crowd scenes in 'part A'. The data is randomly split into 80% train and 20% test samples (uncontaminated). For DINO-2 features we employ a fixed pre-trained backbone model `DINOv2-ViT-S/14` ($d = 384$, see https://github.com/facebookresearch/dinov2) with no additional finetuning and access image features from the penultimate layer with a mean aggregation.

**Method implementations.**    We predominantly employ `PyTorch` for implementing all SBL updates and experiments (Paszke et al., 2019), with gradient-based updates making use of the Adam optimizer. We implement random Fourier features as well as Ridge and Huber regression baselines via `scikit-learn` (Pedregosa et al., 2011), and the exact GP and sparse GP (via inducing points, $n_{\text{ind}} = 256$) using `GPyTorch` (Gardner et al., 2018). Robust regression with Student-$t$ likelihood is manually implemented using EM-style iterative steps with a map-back to Gaussian variance. The BLR is implemented using the IVON optimizer (Shen et al., 2024) to obtain a variational posterior and is iterated with EM steps.

# E. Additional Experiment Results

We report additional results, including *(i)* a comparison of SBL methods on the kernel regression task (Tab. 6, Tab. 7), *(ii)* full experimental results for tabular regression on all nine datasets for uncontaminated and 10% contamination cases (Tab. 8, Tab. 9, Tab. 10, Tab. 11, Tab. 12), *(iii)* an additional experiment relating to influence and leverage (Fig. 6), and *(iv)* baseline results and visual examples for neural network regression (Fig. 7, Fig. 8).

*Table 6.* Comparison of heteroscedastic SBL methods on the kernel regression task. **Performance measured via RMSE** as a function of data contamination on *Boston* ($n = 506$, 20% test split, avg. over 20 trials, $\pm 1\sigma$).

| Method | 0% | 5% | 10% | 15% | 20% | 25% | 30% |
|---|---|---|---|---|---|---|---|
| EM | $3.305 \pm 0.446$ | $3.562 \pm 0.608$ | $3.840 \pm 0.755$ | $4.829 \pm 0.124$ | $5.096 \pm 0.278$ | $5.581 \pm 0.196$ | $6.067 \pm 0.115$ |
| MacKay | $3.456 \pm 0.506$ | $3.569 \pm 0.559$ | $3.707 \pm 0.559$ | $4.051 \pm 0.503$ | $4.066 \pm 0.868$ | $4.519 \pm 0.693$ | $5.037 \pm 1.179$ |
| $\ell_2$-IRLS | $3.303 \pm 0.448$ | $3.521 \pm 0.580$ | $3.713 \pm 0.572$ | $4.126 \pm 0.590$ | $4.284 \pm 0.860$ | $4.679 \pm 0.783$ | $5.216 \pm 1.284$ |
| $\ell_1$-IRLS | $3.369 \pm 0.432$ | $3.988 \pm 0.655$ | $4.617 \pm 0.562$ | $5.035 \pm 0.629$ | $5.670 \pm 0.813$ | $6.060 \pm 0.761$ | $6.424 \pm 0.868$ |

*Table 7.* Comparison of heteroscedastic SBL methods on the kernel regression task. **Performance measured via NLL** as a function of data contamination on *Boston* ($n = 506$, 20% test split, avg. over 20 trials, $\pm 1\sigma$).

| Method | 0% | 5% | 10% | 15% | 20% | 25% | 30% |
|---|---|---|---|---|---|---|---|
| EM | $2.593 \pm 0.131$ | $2.673 \pm 0.127$ | $2.785 \pm 0.114$ | $2.944 \pm 0.023$ | $3.040 \pm 0.015$ | $3.157 \pm 0.018$ | $3.273 \pm 0.052$ |
| MacKay | $2.654 \pm 0.165$ | $2.734 \pm 0.085$ | $2.877 \pm 0.068$ | $3.005 \pm 0.063$ | $3.077 \pm 0.076$ | $3.153 \pm 0.088$ | $3.265 \pm 0.083$ |
| $\ell_2$-IRLS | $2.594 \pm 0.133$ | $2.678 \pm 0.109$ | $2.814 \pm 0.076$ | $2.931 \pm 0.062$ | $2.997 \pm 0.116$ | $3.093 \pm 0.081$ | $3.183 \pm 0.122$ |
| $\ell_1$-IRLS | $2.629 \pm 0.143$ | $2.849 \pm 0.242$ | $3.112 \pm 0.251$ | $3.308 \pm 0.288$ | $3.654 \pm 0.410$ | $3.855 \pm 0.431$ | $4.070 \pm 0.492$ |

*Table 8.* Comparison of heteroscedastic SBL methods against sparse and robust baselines in predictive RMSE and effective support sizes (weights $\theta$ and samples $\mathbf{y}$). Improvements over homoscedastic counterparts are shown as $(-x\%)$. **Results on Energy, Carbon, Protein with no contamination** (avg. over 10 trials, $\pm 1\sigma$).

| | Energy | | | Carbon | | | Protein | | |
|---|---|---|---|---|---|---|---|---|---|
| Method | RMSE ($\downarrow$) | ESS($\theta$) | ESS($\mathbf{y}$) | RMSE ($\downarrow$) | ESS($\theta$) | ESS($\mathbf{y}$) | RMSE ($\downarrow$) | ESS($\theta$) | ESS($\mathbf{y}$) |
| OLS | $1.67 \pm 0.22$ | 100 | 100 | $0.014 \pm 0.003$ | 100 | 100 | $460.6 \pm 73.1$ | 100 | 100 |
| Ridge | $2.13 \pm 0.13$ | 89.1 | 100 | $0.026 \pm 0.002$ | 69.9 | 100 | $742.3 \pm 118.7$ | 71.3 | 100 |
| GP | $\mathbf{1.43 \pm 0.24}$ | 3.9 | 100 | $\mathbf{0.013 \pm 0.004}$ | 6.2 | 100 | $\mathbf{356.4 \pm 83.7}$ | 6.9 | 100 |
| Student-t | $2.40 \pm 0.22$ | 100 | 96.7 | $0.032 \pm 0.003$ | 100 | 96.4 | $1498.8 \pm 131.7$ | 100 | 93.9 |
| Huber | $2.14 \pm 0.24$ | 100 | 88.4 | $0.015 \pm 0.004$ | 100 | 99.3 | $634.3 \pm 128.7$ | 100 | 97.6 |
| EM | $1.86 \pm 0.17$ $(+1.73\%)$ | 38.4 | 100 | $0.014 \pm 0.004$ $(+1.16\%)$ | 24.9 | 100 | $453.4 \pm 106.7$ $(-0.23\%)$ | 19.4 | 100 |
| MacKay | $1.92 \pm 0.15$ $(+3.17\%)$ | 7.7 | 98.6 | $0.015 \pm 0.003$ $(+10.45\%)$ | 3.1 | 100 | $487.5 \pm 114.2$ $(+7.03\%)$ | 6.3 | 100 |
| $\ell_2$-IRLS | $1.86 \pm 0.17$ $(+2.51\%)$ | 44.6 | 100 | $0.014 \pm 0.004$ $(+0.36\%)$ | 18.7 | 100 | $451.4 \pm 111.5$ $(-0.96\%)$ | 15.9 | 100 |
| $\ell_1$-IRLS | $1.87 \pm 0.14$ $(+9.03\%)$ | 9.6 | 100 | $0.015 \pm 0.003$ $(+9.34\%)$ | 5.7 | 100 | $481.8 \pm 109.9$ $(+6.18\%)$ | 10.8 | 100 |
| Grad. (Primal) | $2.36 \pm 0.36$ $(+25.88\%)$ | 8.2 | 72.8 | $\mathbf{0.013 \pm 0.004}$ $(-6.58\%)$ | 3.9 | 100 | $532.5 \pm 133.1$ $(+16.48\%)$ | 9.1 | 96.5 |
| Grad. (Dual) | $2.36 \pm 0.36$ $(+25.83\%)$ | 8.0 | 72.8 | $\mathbf{0.013 \pm 0.004}$ $(-6.57\%)$ | 3.3 | 100 | $532.3 \pm 133.2$ $(+16.39\%)$ | 7.7 | 96.5 |

*Table 9.* Comparison of heteroscedastic SBL methods against sparse and robust baselines in predictive RMSE and effective support sizes (weights $\boldsymbol{\theta}$ and samples $\mathbf{y}$). Improvements over homoscedastic counterparts are shown as ($-x\%$). **Results on Boston, Yacht, Concrete with no contamination** (avg. over 10 trials, $\pm 1\sigma$).

| | Boston | | | Yacht | | | Concrete | | |
|---|---|---|---|---|---|---|---|---|---|
| Method | RMSE ($\downarrow$) | ESS($\boldsymbol{\theta}$) | ESS($\mathbf{y}$) | RMSE ($\downarrow$) | ESS($\boldsymbol{\theta}$) | ESS($\mathbf{y}$) | RMSE ($\downarrow$) | ESS($\boldsymbol{\theta}$) | ESS($\mathbf{y}$) |
| OLS | $8.47 \pm 2.04$ | 100 | 100 | $0.89 \pm 0.24$ | 100 | 100 | $7.13 \pm 0.60$ | 100 | 100 |
| Ridge | $3.89 \pm 0.54$ | 96.2 | 100 | $7.43 \pm 1.09$ | 91.1 | 100 | $8.04 \pm 0.32$ | 71.2 | 100 |
| GP | $\mathbf{3.24 \pm 0.47}$ | 10.3 | 100 | $\mathbf{0.48 \pm 0.15}$ | 5.9 | 100 | $\mathbf{5.14 \pm 0.41}$ | 15.4 | 100 |
| Student-t | $4.61 \pm 0.68$ | 100 | 95.9 | $9.52 \pm 1.57$ | 100 | 93.7 | $8.35 \pm 0.35$ | 100 | 97.7 |
| Huber | $3.93 \pm 0.63$ | 100 | 93.7 | $7.54 \pm 1.26$ | 100 | 89.5 | $7.79 \pm 0.34$ | 100 | 95.7 |
| EM | $3.31 \pm 0.43$ ($-9.93\%$) | 42.8 | 100 | $2.88 \pm 0.37$ ($+45.73\%$) | 30.4 | 100 | $7.10 \pm 0.53$ ($-0.85\%$) | 26.0 | 98.6 |
| MacKay | $3.58 \pm 0.50$ ($+1.54\%$) | 14.1 | 96.0 | $3.17 \pm 0.45$ ($+48.40\%$) | 10.4 | 98.8 | $7.25 \pm 0.60$ ($+1.57\%$) | 20.3 | 94.5 |
| $\ell_2$-IRLS | $3.28 \pm 0.41$ ($-11.24\%$) | 59.0 | 100 | $2.83 \pm 0.38$ ($+47.27\%$) | 37.4 | 100 | $7.10 \pm 0.55$ ($-0.71\%$) | 27.9 | 98.1 |
| $\ell_1$-IRLS | $3.44 \pm 0.49$ ($-11.64\%$) | 17.5 | 100 | $3.24 \pm 0.49$ ($+107.75\%$) | 19.8 | 100 | $7.11 \pm 0.56$ ($+0.07\%$) | 33.9 | 99.4 |
| Grad. (Primal) | $3.77 \pm 0.69$ ($+6.69\%$) | 11.9 | 58.4 | $4.25 \pm 0.51$ ($+100.18\%$) | 8.3 | 70.8 | $7.86 \pm 0.58$ ($+10.37\%$) | 20.0 | 42.7 |
| Grad. (Dual) | $3.77 \pm 0.69$ ($+6.59\%$) | 11.9 | 58.3 | $4.28 \pm 0.56$ ($+105.23\%$) | 8.0 | 70.6 | $7.86 \pm 0.58$ ($+10.34\%$) | 19.1 | 42.6 |

*Table 10.* Comparison of heteroscedastic SBL methods against sparse and robust baselines in predictive RMSE and effective support sizes (weights $\boldsymbol{\theta}$ and samples $\mathbf{y}$). Improvements over homoscedastic counterparts are shown as ($-x\%$). **Results on Boston, Yacht, Concrete with 10% outlier contamination** (avg. over 10 trials, $\pm 1\sigma$).

| | Boston | | | Yacht | | | Concrete | | |
|---|---|---|---|---|---|---|---|---|---|
| Method | RMSE ($\downarrow$) | ESS($\boldsymbol{\theta}$) | ESS($\mathbf{y}$) | RMSE ($\downarrow$) | ESS($\boldsymbol{\theta}$) | ESS($\mathbf{y}$) | RMSE ($\downarrow$) | ESS($\boldsymbol{\theta}$) | ESS($\mathbf{y}$) |
| OLS | $26.85 \pm 10.45$ | 100 | 100 | $24.83 \pm 6.04$ | 100 | 100 | $9.89 \pm 1.03$ | 100 | 100 |
| Ridge | $4.43 \pm 0.53$ | 96.5 | 100 | $8.62 \pm 0.90$ | 92.9 | 100 | $8.90 \pm 0.46$ | 80.4 | 100 |
| GP | $6.41 \pm 1.37$ | 25.7 | 100 | $7.84 \pm 1.93$ | 6.3 | 100 | $8.46 \pm 0.62$ | 18.7 | 100 |
| Student-t | $\mathbf{4.29 \pm 0.66}$ | 100 | 93.7 | $8.23 \pm 1.28$ | 100 | 94.1 | $8.36 \pm 0.42$ | 100 | 95.1 |
| Huber | $3.96 \pm 0.61$ | 100 | 88.6 | $7.62 \pm 1.24$ | 100 | 85.0 | $7.99 \pm 0.41$ | 100 | 91.5 |
| EM | $4.75 \pm 1.00$ ($-15.91\%$) | 27.1 | 91.3 | $3.95 \pm 0.63$ ($-48.57\%$) | 23.8 | 92.5 | $\mathbf{7.55 \pm 0.74}$ ($-15.01\%$) | 22.0 | 90.5 |
| MacKay | $\mathbf{4.29 \pm 0.69}$ ($-25.38\%$) | 13.9 | 85.5 | $\mathbf{3.75 \pm 0.43}$ ($-51.59\%$) | 9.4 | 89.8 | $\mathbf{7.55 \pm 0.70}$ ($-15.41\%$) | 20.2 | 87.2 |
| $\ell_2$-IRLS | $5.26 \pm 1.14$ ($-6.83\%$) | 37.8 | 92.2 | $4.07 \pm 0.73$ ($-47.42\%$) | 38.7 | 92.5 | $\mathbf{7.55 \pm 0.76}$ ($-15.03\%$) | 27.5 | 90.2 |
| $\ell_1$-IRLS | $7.33 \pm 1.91$ ($-6.50\%$) | 23.1 | 93.8 | $7.37 \pm 1.32$ ($-10.48\%$) | 27.9 | 98.6 | $8.02 \pm 0.75$ ($-12.41\%$) | 27.9 | 94.8 |
| Grad. (Primal) | $5.02 \pm 0.81$ ($-11.98\%$) | 15.7 | 54.1 | $4.30 \pm 0.54$ ($-43.42\%$) | 9.5 | 60.2 | $8.78 \pm 0.77$ ($-2.10\%$) | 20.8 | 37.4 |
| Grad. (Dual) | $5.01 \pm 0.78$ ($-12.13\%$) | 15.5 | 54.0 | $4.29 \pm 0.54$ ($-43.55\%$) | 9.1 | 60.1 | $8.78 \pm 0.77$ ($-2.10\%$) | 20.1 | 37.5 |

*Table 11.* Comparison of heteroscedastic SBL methods against sparse and robust baselines in predictive RMSE and effective support sizes (weights $\boldsymbol{\theta}$ and samples $\mathbf{y}$). Improvements over homoscedastic counterparts are shown as ($-x\%$). **Results on Power, Kin8nm, Elevators with no contamination** (avg. over 10 trials, $\pm 1\sigma$).

| | Power | | | Kin8nm | | | Elevators | | |
|---|---|---|---|---|---|---|---|---|---|
| Method | RMSE ($\downarrow$) | ESS($\boldsymbol{\theta}$) | ESS($\mathbf{y}$) | RMSE ($\downarrow$) | ESS($\boldsymbol{\theta}$) | ESS($\mathbf{y}$) | RMSE ($\downarrow$) | ESS($\boldsymbol{\theta}$) | ESS($\mathbf{y}$) |
| OLS | $4.93 \pm 0.56$ | 100 | 100 | $0.134 \pm 0.005$ | 100 | 100 | $0.004 \pm 0.000$ | 100 | 100 |
| Ridge | $4.22 \pm 0.11$ | 91.4 | 100 | $0.141 \pm 0.004$ | 60.0 | 100 | $0.004 \pm 0.000$ | 52.8 | 100 |
| GP | $\mathbf{4.04 \pm 0.10}$ | 5.1 | 100 | $0.214 \pm 0.345$ | 9.3 | 100 | $0.004 \pm 0.001$ | 8.8 | 100 |
| Student-t | $4.26 \pm 0.11$ | 100 | 98.1 | $0.143 \pm 0.004$ | 100 | 97.7 | $0.004 \pm 0.000$ | 100 | 96.5 |
| Huber | $4.17 \pm 0.11$ | 100 | 97.0 | $0.138 \pm 0.004$ | 100 | 95.9 | $0.004 \pm 0.000$ | 100 | 94.7 |
| EM | $4.12 \pm 0.11$ ($-0.31\%$) | 64.5 | 100 | $\mathbf{0.132 \pm 0.004}$ ($+0.08\%$) | 22.3 | 97.0 | $\mathbf{0.003 \pm 0.000}$ ($-0.83\%$) | 16.8 | 97.2 |
| MacKay | $4.15 \pm 0.11$ ($+0.14\%$) | 4.4 | 97.3 | $0.134 \pm 0.004$ ($+0.87\%$) | 19.4 | 90.9 | $\mathbf{0.003 \pm 0.000}$ ($+0.18\%$) | 14.3 | 93.6 |
| $\ell_2$-IRLS | $4.12 \pm 0.11$ ($-0.41\%$) | 62.2 | 99.1 | $0.133 \pm 0.004$ ($+0.29\%$) | 23.3 | 96.1 | $\mathbf{0.003 \pm 0.000}$ ($-0.86\%$) | 19.0 | 96.8 |
| $\ell_1$-IRLS | $4.14 \pm 0.11$ ($-0.11\%$) | 6.2 | 100 | $\mathbf{0.132 \pm 0.004}$ ($+0.04\%$) | 30.0 | 98.9 | $\mathbf{0.003 \pm 0.000}$ ($-1.58\%$) | 22.6 | 98.3 |
| Grad. (Primal) | $4.41 \pm 0.14$ ($+6.00\%$) | 5.7 | 42.3 | $0.141 \pm 0.005$ ($+5.94\%$) | 18.9 | 34.8 | $0.004 \pm 0.000$ ($+5.07\%$) | 16.1 | 38.1 |
| Grad. (Dual) | $4.41 \pm 0.14$ ($+6.03\%$) | 5.6 | 42.3 | $0.141 \pm 0.005$ ($+5.94\%$) | 18.5 | 34.8 | $0.004 \pm 0.000$ ($+5.10\%$) | 15.8 | 38.1 |

*Table 12.* Comparison of heteroscedastic SBL methods against sparse and robust baselines in predictive RMSE and effective support sizes (weights $\boldsymbol{\theta}$ and samples $\mathbf{y}$). Improvements over homoscedastic counterparts are shown as ($-x\%$). **Results on Power, Kin8nm, Elevators with 10% outlier contamination** (avg. over 10 trials, $\pm 1\sigma$).

| | Power | | | Kin8nm | | | Elevators | | |
|---|---|---|---|---|---|---|---|---|---|
| Method | RMSE ($\downarrow$) | ESS($\boldsymbol{\theta}$) | ESS($\mathbf{y}$) | RMSE ($\downarrow$) | ESS($\boldsymbol{\theta}$) | ESS($\mathbf{y}$) | RMSE ($\downarrow$) | ESS($\boldsymbol{\theta}$) | ESS($\mathbf{y}$) |
| OLS | $12.41 \pm 2.59$ | 100 | 100 | $0.171 \pm 0.005$ | 100 | 100 | $0.004 \pm 0.000$ | 100 | 100 |
| Ridge | $4.83 \pm 0.17$ | 97.0 | 100 | $0.153 \pm 0.003$ | 72.3 | 100 | $0.004 \pm 0.000$ | 68.3 | 100 |
| GP | $4.86 \pm 0.15$ | 19.4 | 100 | $0.156 \pm 0.006$ | 97.1 | 100 | $0.004 \pm 0.000$ | 27.0 | 100 |
| Student-t | $4.25 \pm 0.11$ | 100 | 92.9 | $0.144 \pm 0.004$ | 100 | 95.5 | $0.004 \pm 0.000$ | 100 | 94.3 |
| Huber | $4.20 \pm 0.11$ | 100 | 91.1 | $0.140 \pm 0.004$ | 100 | 92.1 | $0.004 \pm 0.000$ | 100 | 90.2 |
| EM | $\mathbf{4.19 \pm 0.14}$ ($-11.99\%$) | 51.4 | 91.1 | $\mathbf{0.136 \pm 0.004}$ ($-10.58\%$) | 20.2 | 88.8 | $\mathbf{0.003 \pm 0.000}$ ($-9.48\%$) | 16.5 | 89.4 |
| MacKay | $4.20 \pm 0.11$ ($-11.74\%$) | 2.9 | 87.1 | $0.137 \pm 0.004$ ($-10.34\%$) | 18.2 | 82.5 | $\mathbf{0.003 \pm 0.000}$ ($-9.30\%$) | 13.8 | 84.2 |
| $\ell_2$-IRLS | $4.21 \pm 0.18$ ($-11.55\%$) | 58.0 | 90.9 | $\mathbf{0.136 \pm 0.004}$ ($-10.65\%$) | 23.2 | 88.6 | $\mathbf{0.003 \pm 0.000}$ ($-9.59\%$) | 19.4 | 88.9 |
| $\ell_1$-IRLS | $4.40 \pm 0.15$ ($-19.43\%$) | 6.9 | 93.2 | $0.141 \pm 0.004$ ($-10.60\%$) | 23.3 | 92.7 | $0.004 \pm 0.000$ ($-11.88\%$) | 20.6 | 92.0 |
| Grad. (Primal) | $4.58 \pm 0.22$ ($-3.40\%$) | 8.8 | 38.6 | $0.150 \pm 0.005$ ($-1.86\%$) | 19.5 | 30.7 | $0.004 \pm 0.000$ ($-2.51\%$) | 16.6 | 33.3 |
| Grad. (Dual) | $4.55 \pm 0.18$ ($-3.97\%$) | 8.6 | 38.5 | $0.150 \pm 0.005$ ($-1.86\%$) | 19.1 | 30.6 | $0.004 \pm 0.000$ ($-2.51\%$) | 16.3 | 33.3 |

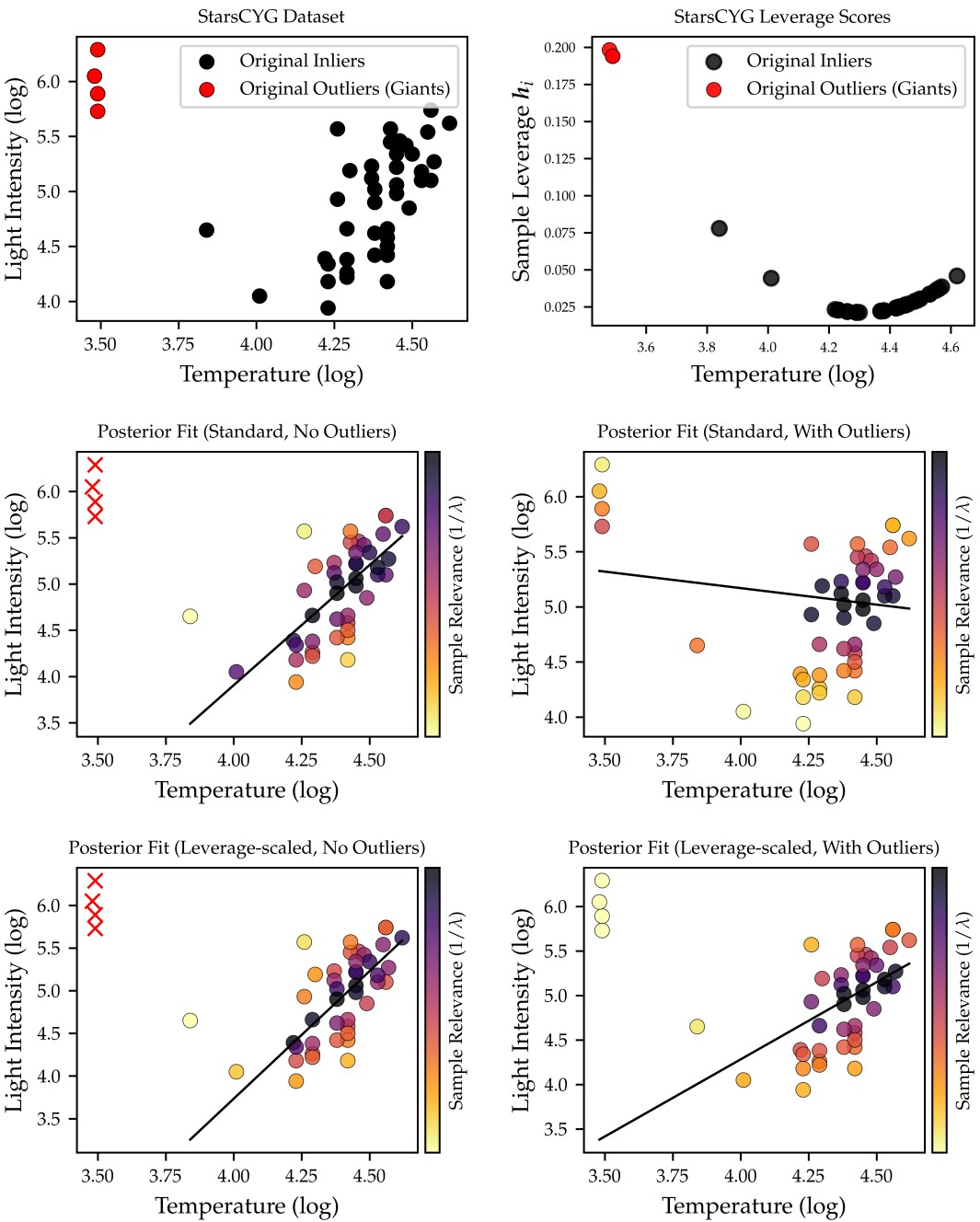

*Figure 6. Top row:* We run an additional experiment complementing § 4 with original *high-leverage* outliers representing giant stars contained in the *StarsCYG* dataset. *Center row:* Those four samples clearly exhibit strong leverage and are able to pull the model fit towards them unless excluded *a priori*. Here, we run joint ARD with standard EM updates on the single original feature. *Bottom row:* Motivated by the connection between $\lambda_i$ and the leverage-adjusted LOO residual $r_{-i,i}^2$ we test a 'studentized' EM noise update of the form $\lambda_i^{t+1} = r_i^2/(1 - h_i)^p$ which uses posterior leverage as a *multiplicative* magnifier taken to the power $p \geq 2$. While heuristic, this substantially reduces the influence of the high-leverage outliers and yields a fit closer to the uncontaminated case, suggesting potential for future investigation into leverage-adjusted updates obtained via ARD principles.

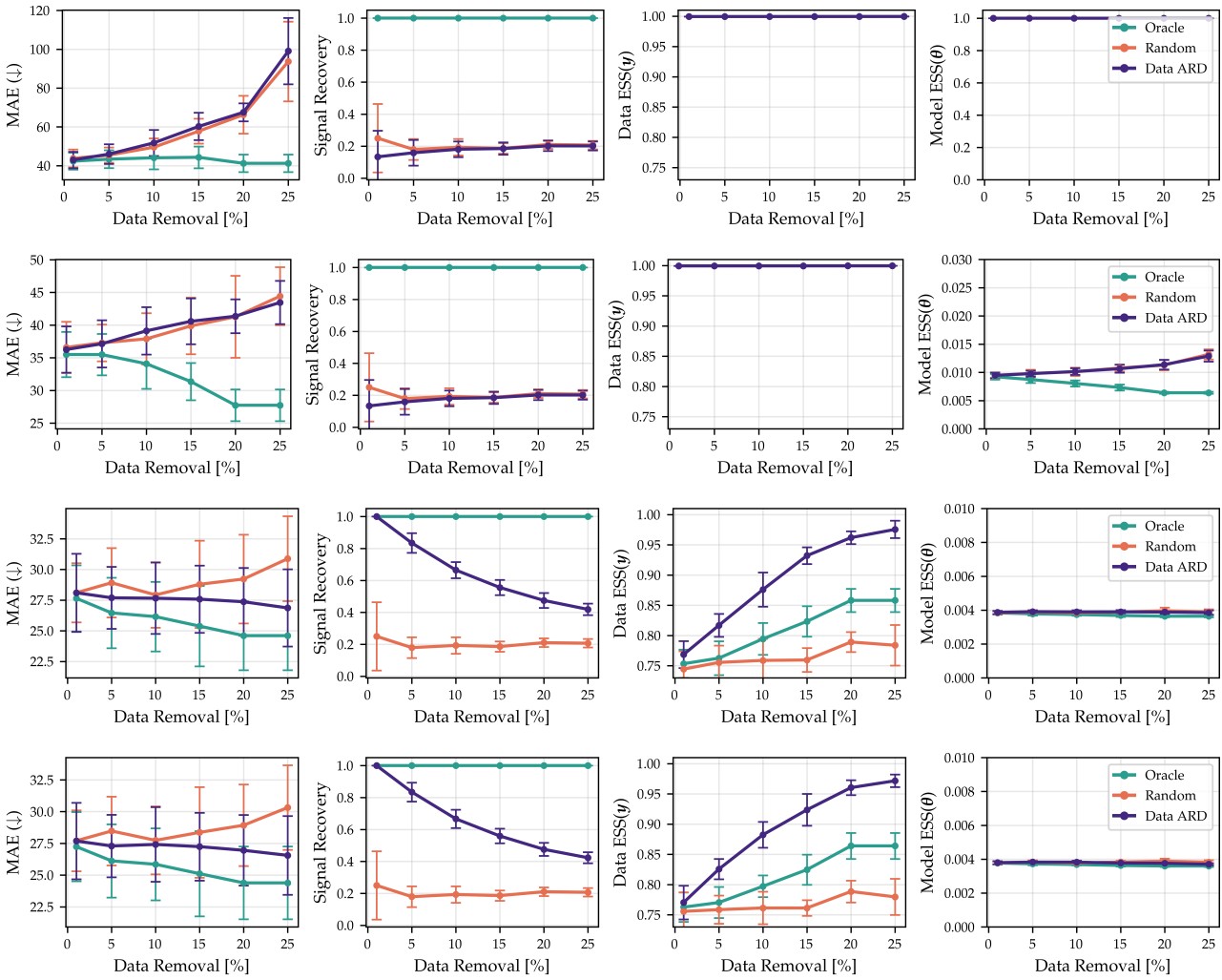

*Figure 7.* We report predictive performance (MAE), outlier recovery, ESS($\mathbf{y}$), and ESS($\boldsymbol{\theta}$) for the neural network experiment in § 5.4 across methods. ***Rows (top to bottom):*** **OLS, Ridge, MacKay, and EM** (avg. over 10 trials, $\pm 1\sigma$). As expected, OLS and Ridge exhibit poor outlier recovery and yield ESS($\mathbf{y}$) = 1.0 (no data sparsification), while Ridge attains feature sparsity comparable to the SBL approaches. For MacKay and EM, ESS($\mathbf{y}$) increases as larger fractions of high-noise points are removed, indicating self-consistency: once outliers are pruned, relevance becomes more evenly distributed across remaining samples, consistent with increasingly homogeneous inlier noise. EM and MacKay are selected here for their typically faster convergence and lower computational cost.

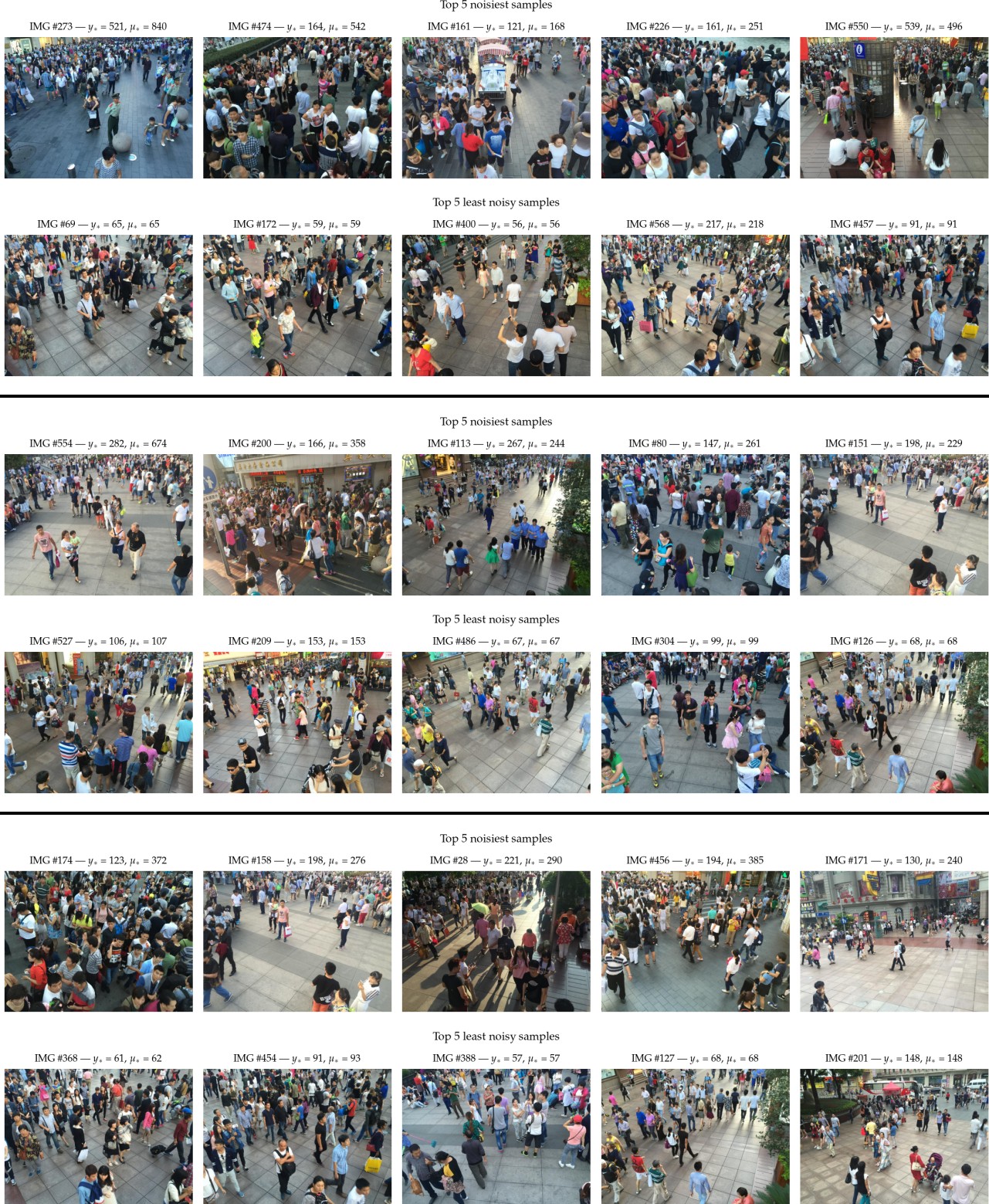

*Figure 8.* We provide additional qualitative examples complementing Fig. 5 by visualizing the five highest- and lowest-noise samples as ranked by $\lambda$, learned via EM for three distinct trials (2, 5, 8). High-noise images predominantly depict dense, high-count crowds, while low-noise examples correspond to simpler low-count scenes, indicating consistent retrieval of unreliable labels.

