# OpenReview forum: "Joint Model and Data Sparsification via the Marginal Likelihood"
_ICML.cc/2026/Conference — ICML 2026 regular_

### Official Review · Reviewer_PW2g · 2026-03-11

**Soundness:** 3
**Presentation:** 4
**Significance:** 2
**Originality:** 2
**Overall Recommendation:** 4
**Confidence:** 4

**Summary:**

This paper proposes a joint model and data sparsification framework within the Sparse Bayesian Learning (SBL) paradigm. The key idea is to extend the standard Automatic Relevance Determination (ARD) principle—which learns per-feature precision parameters ω to prune irrelevant features—by additionally introducing per-sample noise variances ε as first-class parameters. Both are optimized simultaneously via the marginal likelihood, yielding a single Bayesian objective that promotes feature sparsity (via ω → ∞) and data robustness (via large εᵢ for outliers). The authors derive closed-form heteroscedastic noise updates under several established SBL optimization schemes: expectation maximization (EM), MacKay fixed-point updates, ℓ₂-IRLS (via Fenchel conjugate majorization-minimization), ℓ₁-IRLS, and gradient-based methods. Interpretive connections are drawn between the learned noise parameters and classical influence diagnostics (leave-one-out residuals, leverage), as well as robust Gaussian process regression through the RVM kernel interpretation. Experiments span synthetic signal recovery, UCI tabular regression under contamination, sparse kernel regression (RVM on Boston), and a neural linear model for crowd counting with DINO features.

**Compliance With Llm Reviewing Policy:**

Affirmed.

**Key Questions For Authors:**

- Have you considered connections to the Robust PCA / Sparse+Low-Rank decomposition literature? The structural parallel—simultaneously identifying a
sparse signal component and sparse corruptions—seems worth discussing, even if the optimization landscapes differ.
- The EM noise update provides only an additive leverage correction (εᵢ = rᵢ² + εᵢhᵢ) compared to the multiplicative magnification of the LOO residual ((1
− hᵢ)⁻²). As noted in §4, this means high-leverage outliers may remain insufficiently downweighted. Have you explored studentized variants or iterative
leverage corrections to address this?
- How sensitive are the results to the choice of uninformative hyperpriors (a, b → 0)? In high-dimensional settings, improper priors on ε can lead to
degenerate solutions where all samples are deemed outliers.

**Limitations:**

This is a well-written paper addressing a relevant problem at the intersection of sparse recovery and robust estimation. The formulation is clean, the derivations are correct, and the experimental validation is adequate. However, the contribution is best characterized as a natural and relatively straightforward extension of existing SBL machinery to the heteroscedastic case, rather than a fundamentally new methodological advance. The restriction to linear-in-weights models and the modest empirical gains over simple robust baselines further limit the significance. I recommend a borderline/weak accept: the paper meets the quality bar for a venue like ICML in terms of writing and rigor, but the novelty is incremental given the substantial body of prior work on robust and heteroscedastic extensions of SBL and RVMs.

**Strengths And Weaknesses:**

## Strengths:

1. The problem formulation is clean and well-motivated. Treating data robustification as a sparsification task—symmetric to the standard model ARD
treatment of features—is an elegant conceptual contribution. The fact that this extension preserves Gaussian conjugacy and admits closed-form updates
within all major SBL optimization families (EM, MacKay, IRLS) is a genuine practical advantage over robust alternatives that require approximate inference
(e.g., Student-t likelihoods, GP-based heteroscedastic models).

2. The paper is clearly written and well-organized. The progression from the marginal likelihood objective (§3.1) through the optimization algorithms (§
3.2) to the interpretive analysis (§4) is logical and easy to follow. Table 2 provides a useful at-a-glance comparison of the algorithmic variants and
their convergence properties.

3. The connection between the EM noise update εᵢ = rᵢ² + xᵢᵀΣ_ω xᵢ and the classical PRESS leave-one-out residual rᵢ,₋ᵢ² = rᵢ²/(1 − hᵢ)² is insightful and
well-developed. This bridges the proposed Bayesian mechanism to established robust regression diagnostics in a non-trivial way.

4. The experimental evaluation is thorough in its coverage of settings (synthetic, tabular, kernel, neural) and includes appropriate baselines (OLS,
Ridge, GP, Student-t, Huber). The synthetic experiments with controlled sparsity and contamination ratios (Fig. 3) are particularly informative for
understanding the method's operating regime.

## Weaknesses:

1. The core contribution, while principled, is arguably incremental relative to the extensive existing literature on robust SBL and heteroscedastic
extensions. The idea of introducing per-sample noise variances in Bayesian linear models is not new per se—prior work on robust RVMs (Mitra et al., 2010;
Faul & Tipping, 2001), heteroscedastic RVMs (Khashabi et al., 2013), and SBL under heteroscedastic noise (Gerstoft et al., 2017; Sundin et al., 2015) has
explored closely related formulations. The paper would benefit from a more precise delineation of what is technically novel beyond the specific derivation
of closed-form updates for the joint (ω, ε) case across multiple SBL solvers.

2. The method is restricted to linear-in-weights models. While the authors acknowledge this limitation and point to the Bayesian Learning Rule (Khan &
Rue, 2023) as a potential extension path, the tentative BLR experiment (Table 4) shows mixed results and does not convincingly demonstrate scalability
beyond the linear setting. For an ICML audience, the inability to handle end-to-end neural network parameterizations limits the scope of impact.

3. The experimental gains, while consistent, are often modest in absolute terms. On the tabular benchmarks (Table 3), the improvements over Huber
regression—a simple and well-understood robust baseline—are not always statistically significant given the reported standard deviations. The crowd
counting experiment (§5.4) explicitly notes that "the injected corruptions are only mildly harmful," which somewhat undermines the practical motivation.

4. A natural question arises regarding the relationship to Robust PCA and Sparse PCA, where one similarly seeks sparsity along both the feature and sample
dimensions of a data matrix. In Robust PCA (Candès et al., 2011), the decomposition L + S separates a low-rank component from a sparse corruption matrix,
which conceptually parallels the joint ARD objective of separating signal (sparse ω) from contamination (sparse ε). Sparse PCA (Zou et al., 2006) further
imposes sparsity on the loading vectors. The authors do not discuss these connections, and doing so could both strengthen the positioning and suggest
additional baselines or problem settings (e.g., unsupervised or matrix-valued observations).

---

> ### Author Rebuttal · Authors · 2026-03-31
>
> Dear reviewer PW2g, we thank you for your time and insights and are glad you appreciated our “*clean and well-motivated*” work and “*elegant conceptual contribution*”. We address your remarks below.
>
> > *more precise delineation of what is technically novel […]*
> >
>
> We kindly refer to our response to reviewer aBQz (**Novelty and Contributions**).
>
> > *The method is restricted to linear-in-weights models. […] the tentative BLR experiment (Table 4) shows mixed results […]*
> >
>
> It is correct that closed-form updates rely on a linear-in-weights parametrization (§6). This limitation stems from the classical SBL solvers we extend but also persists in recent SBL-related proposals [ZH’25, WY’24, AS’21]. Nonetheless, our experiments show broad applicability through fixed nonlinear features, with explicit pointers towards future expansion to fully non-linear models. App. §B.8 details a plausible route to move beyond conjugacy by variational approximations with the BLR, while retaining ARD-type parametrizations. Table 4 points towards practical feasibility, but a full treatment is beyond this work’s scope.
>
> > *The experimental gains, while consistent, are often modest in absolute terms. […] improvements over Huber regression […] are not always statistically significant*
> >
>
> Our goal w.r.t. robust baselines (e.g., Huber) is not necessarily to beat them on RMSE. A well-working robustness mechanism should approximately reflect the injected outlier rate; indeed, Table 3 shows $\text{ESS}(y)\approx 90\%$ under 10% contamination, indicating both Huber and joint ARD downweight roughly the right amount. The added value of joint ARD is that it **simultaneously yields meaningful feature sparsity** and interpretable per-sample diagnostics, which robust-only baselines do not provide. In contrast, **predictive gains are visible against sparsity-only** baselines (e.g., homoscedastic SBL or GP/Ridge), and Table 3 reports these as $-X$% relative loss reductions.
>
> > *The crowd counting experiment (§5.4) explicitly notes that "the injected corruptions are only mildly harmful," which somewhat undermines the practical motivation.*
> >
>
> We agree the injected corruptions are intentionally mild, aiming to **mimic realistic annotation** **noise** in crowded scenes rather than contrived failures. Even so, Fig. 5 shows a consistent $\lambda$ ranking that highlights high-count images as unreliable, and its use as a selection criterion stabilizes MAE under increasing data scarcity, while random removal degrades. Thus the experiment supports our motivation for diagnostics and actionability, and does not inherently undermine it.
>
> > *connections to the Robust PCA / Sparse+Low-Rank decomposition literature?*
> >
>
> Thank you for the interesting suggestion. We agree on conceptual similarities (separating signal from corruptions), and the framing as a convex problem ([CE’11], Eq. 1.1) bears similarity to IRLS. However, we target supervised prediction rather than unsupervised matrix completion or recovery, and it is not obvious how to directly design a comparable baseline. PCA-then-regress schemes form a two-step approach rather than simultaneous identification and model fitting as we do, and may impose additional restrictions. For instance, Robust PCA [CE’11] requires distinct low-rank and corrupt components for identifiability, while in our general setting each sample may be an important contributor *and* corrupted, with ARD balancing model fit and robustness with continuous weights. That said, we are happy to add a related-works remark noting connections.
>
> > *The EM noise update provides only an additive leverage correction. […] Have you explored studentized variants […]?*
> >
>
> Yes, §4 discusses additive vs. multiplicative correction, and App. Fig. 6 provides a tentative ‘studentized’ variant to better handle high-leverage points. Deriving such leverage-magnified updates from canonical ARD principles is an interesting direction for future work.
>
> > *How sensitive are the results to the choice of uninformative hyperpriors?*
> >
>
> We discuss sensitivity w.r.t. practical design in App. §D.1, stressing challenges in heteroscedastic modelling due to high degrees of freedom. Strongly misspecified priors can slow convergence and recovery, but we found weakly-informative choices to work well across different algorithms and tasks, with no excessive ‘prior tuning’ necessary. We are happy to expand upon this and add an ablation on the choice of hyperprior settings for $\gamma, \lambda$.
>
> We thank you again and appreciate reconsideration of your score if your concerns have been addressed sufficiently.
>
> **Refs**
>
> [ZH’25] Zhang, H. et al. Efficient Network Automatic Relevance Determination. *ICML* (2025).
>
> [WY’24] See under reviewer aBQz.
>
> [AS’21] Ament, S. et al. Sparse Bayesian learning via stepwise regression. *ICML* (2021).
>
> [CE’11] Candès, E. et al. Robust principal component analysis?. *JACM* (2011).

---

> > ### Author Rebuttal · Reviewer_PW2g · 2026-04-03
> >
> > > we target supervised prediction rather than unsupervised matrix completion or recovery
> >
> > The loss function for what you call "unsupervised" matrix recovery is the denoising loss |X - A|_F^2 and the loss function for matrix completion is sum (X_ij - A_ij)^2 over observed i, j entries. Both of these are instances of a loss function l(X), and another example of loss function l(X) is |WX - Y|_2^2 for a regression problem or you can also write a classification problem with the same type of formalism. The penalty on X is what your work is about and the contributions of the RPCA work are essentially on the properties of that decomposition of X onto sparse + low-rank which still hold here. In that respect the connection is not artificial at all and deserves clear call out

---

> > > ### Author Response · Authors · 2026-04-04
> > >
> > > Thank you for the elaboration, we agree the drawn connection is not artificial when viewed through a shared regularization pattern. Both approaches encourage parsimonious signal representation (via a low-rank component resp. sparsity in ARD model parameters) and structured explanation of corruptions (via a sparse component resp. per-sample noise ARD), and also share overlap in IRLS-type solver procedures. We will examine the relevant Robust/Sparse PCA literature to make both analogies and differences to our ARD/SBL perspective more explicit (e.g., the task setting and assumptions on components). We appreciate the suggestion and will use it to further strengthen positioning and discussion.

---

### Official Review · Reviewer_m7QM · 2026-03-13

**Soundness:** 3
**Presentation:** 3
**Significance:** 2
**Originality:** 3
**Overall Recommendation:** 4
**Confidence:** 3

**Summary:**

The work extends the ARD principle by including "per-sample" noise variances. It then optimizes the noise variances jointly with model precision. This enables automatic downweighting of samples that are uninformative or harmful for model training. The estimate of variances and precision is formulated as a maximum likelihood estimation task. The objective leads to a sparse parameter estimates. Expectation maximization has been used to handle the marginalization task. The proposed scheme is tested on both synthetic data and classic datasets.

**Compliance With Llm Reviewing Policy:**

Affirmed.

**Final Justification:**

Considering my initial review, and the rebuttal, I believe that the work generally sound but it remains limited in contribution. I would hence keep my initial suggestion, i.e., weak accept.

**Key Questions For Authors:**

- In the numerical experiments, do you use gradient-based updates or a closed-form update for the per-sample variances?
- Does per-sample variance update impose higher computational complexity? Wouldn't it be helpful to discuss this complexity in the paper?
- The footnote on page 1 indicate that nonlinear features are also allowed, and using the linear model is for notational simplicity. How would that reflect for instance in $\theta$ update in page 4, assuming that the feature extraction is also a part of the parameterized model. Isn't the results only focuses on a single linear weighted transform?

**Limitations:**

- Model is restricted to be linearly weighted.
- No analysis of the proposed scheme has been provided.
- Numerical experiments are relatively limited.

**Strengths And Weaknesses:**

Strengths:
- The approach is a valid extension of the ARD frameowrk.
- The work has well presented its results. The EM derivations are generally sound.

Weaknesses:
- No analysis has been provided for the proposed scheme. No guarantee on convergence and no sensitivity analyses has been provides.
- The problem formulation has focused on a linear feature transform. Although nonlinear models are later evaluated in the experimental results.
- The work is quite classic with limited technical contribution. The extension of variance to per-sample variance is a basic extension, and the development of EM algorithm is rather straightforward. This limits the originality of the work.
- Numerical experiments are relatively simple. More experiments with outliers in the dataset would have been appreciated. Moreover, considering more advanced learning tasks is suggested.

---

> ### Author Rebuttal · Authors · 2026-03-31
>
> Dear reviewer m7QM, we thank you for your time and are glad you deem our work well presented and sound. We address your remarks below.
>
> > *[…] formulated as a maximum likelihood estimation task. […] Expectation maximization has been used to handle the marginalization task.*
> >
>
> To clarify, our optimization targets the marginal likelihood (Eq. 3) w.r.t. ARD parameters $(\gamma, \lambda)$, not the maximum likelihood w.r.t. model weights $\theta$. In practice, $\theta$ is updated alongside $(\gamma, \lambda)$ (see App. §C), either as the posterior mean or minimizer of an IRLS surrogate. Relatedly, EM is **only one of six considered solvers** for the same objective (EM, MacKay, L2-IRLS, L1-IRLS, primal/dual gradient). We do not claim preference for any one, and §5.1 and §5.2 compare all variants.
>
> > *[…] limited technical contribution. […] development of EM algorithm is rather straightforward.*
> >
>
> Please see our response to reviewer aBQz (**Novelty and Contributions**). We reiterate that EM is only one algorithm, and we derive algorithm-specific updates across all major SBL solvers (EM, MacKay, L2-IRLS, L1-IRLS), summarized in App. §B.7. Perhaps confusion arose because **we only wrote out the variance update once in §3.2 for space**, but updates are derived equally in each case. We will clarify this in the camera-ready.
>
> > *[…] focused on a linear feature transform. Although nonlinear models are later evaluated […] The footnote on page 1 indicate that nonlinear features are also allowed.*
> >
>
> We apologize if the footnote caused confusion. We use $X$ as a generic placeholder for either raw features or fixed (non-linear) transforms $\Phi(X)$ simply **to reduce visual clutter, and this does not affect any derivations.** As the reviewer points out, in experiments features come from multiple sources: raw numerical inputs (§5.1), random Fourier features (§5.2), an RBF kernel basis (§5.3), and pre-trained DINO-2 (§5.4). These features are mostly non-linear and treated as fixed (e.g., in §5.4 only the final linear layer is learned and sparsified via double ARD). No assumptions are made on the feature nature or origin.
>
> > *Numerical experiments are relatively simple. More experiments with outliers […] more advanced learning tasks is suggested.*
> >
>
> We agree more experiments are always possible. Our results already span **12 datasets** (mcycle, synthetic, 9 UCI/OpenML, ShanghaiTech), **multiple model designs** (polynomial, linear, RFF, RBF kernel, neural), and **18 methods** (6 SBL solvers in hetero- and homoscedastic, 6 more baselines). We also include smaller-scale corroborations on influence (Fig. 3), high-leverage (App. Fig. 6), and a variational BLR outlook (Table 4). Considering our conceptual and technical contributions, the breadth of experiments is considerable and **rarely seen in related works**, e.g. compare recent papers [WY’24, AS’21].
>
> > *In the numerical experiments, do you use gradient-based updates or a closed-form update […]?*
> >
>
> Relating to above, §5.1 and §5.2 **compare** **all six solvers**. In §5.3 and §5.4 we select one algorithm each (L2-IRLS and EM) for versatility and computational convenience. See more results for §5.3 under reviewer aBQz, and App. §E in general.
>
> > *No guarantee on convergence and no sensitivity analyses.*
> >
>
> Table 2 offers a high-level summary on optimization properties across SBL procedures. These follow directly from the underlying algorithms, whose convergence is studied more thoroughly in respective papers, e.g. [WD’07] for IRLS, [WJ’83] for EM, or [FA’02] targeting SBL. Our heteroscedastic extension does not alter the algorithmic form; **we inherit well-known optimization properties**. We will explicitly remark this inheritance in §3.2 citing standard works.
>
> > *Does per-sample variance update impose higher computational complexity?*
> >
>
> Yes, heteroscedasticity introduces $\mathcal{O}(n)$ additional parameters, but per-iteration **asymptotic cost is dominated by the same algebraic operations**. In both cases each (outer) iteration forms and factorizes $\Sigma_y$ or $\Sigma_{\theta}$, thus leading costs remain $\mathcal{O}(n^3)$ or $\mathcal{O}(d^3)$ with Cholesky. Only linear-time overhead is added for storing $\lambda$ and operations with $\Lambda^{-1}$. In practice, wall-clock time can increase due to potentially slower convergence, e.g. L1-IRLS faces a harder surrogate problem with additional regularization. We will add a note on complexity and wall-clock timing.
>
> We thank you again and appreciate reconsideration of your score if your concerns have been addressed.
>
> **Refs**
>
> [WY’24] See under reviewer aBQz.
>
> [AS’21] Ament, S. et al. Sparse Bayesian learning via stepwise regression. *ICML* (2021).
>
> [WD’07] Wipf, D. et al. A new view of automatic relevance determination. *NeurIPS* (2007).
>
> [WJ’83] Wu, J. On the convergence properties of the EM algorithm. *AnnStats* (1983).
>
> [FA’02] Faul, A. et al. Analysis of Sparse Bayesian Learning. *NeurIPS* (2002).

---

> > ### Author Rebuttal · Reviewer_m7QM · 2026-04-03
> >
> > While I keep my comments on the linearity of the model, I do acknowledge that my concerns with respect to my first two questions have been addressed as they have been partially due to confusion. I have hence edited my score to 4.

---

> > > ### Author Response · Authors · 2026-04-04
> > >
> > > We are glad our responses resolved earlier confusion and appreciate you raising your score. Regarding linearity of the model: it is entirely correct that the SBL model (Eq. 1) and closed-form updates rely on a linear-in-weights parametrization, as freely acknowledged in §6. This constraint is shared by recent SBL-related proposals [ZH’25, WY’24, AS’21, and others] and the wider literature on neural linear and conjugate Bayesian last-layer models, e.g. [OS’19, WJ’20, WJ’21, HJ’24]. While offering efficient updates and broad applicability through fixed nonlinear features (as shown in §5.4), an expansion to fully end-to-end nonlinear models is one of our principal angles for future work.
> > >
> > > **Additional References**
> > >
> > > [OS’19] Ober, S. et al. Benchmarking the Neural Linear Model for Regression. *AABI* (2019).
> > >
> > > [WJ’20] Watson, J. et al. Neural linear models with functional gaussian process priors. *AABI* (2020).
> > >
> > > [WJ’21] Watson, J. et al. Latent derivative Bayesian last layer networks. *AISTATS* (2021).
> > >
> > > [HJ’24] Harrison, J. et al. Variational Bayesian Last Layers. *ICLR* (2024).

---

### Official Review · Reviewer_aBQz · 2026-03-13

**Soundness:** 3
**Presentation:** 4
**Significance:** 2
**Originality:** 2
**Overall Recommendation:** 4
**Confidence:** 3

**Summary:**

This manuscript extends the automatic relevance determination (ARD) principle to the case of heteroscedastic noise and provides closed-form update rules for several established SBL optimization schemes. Furthermore, two complementary explanations (influence-based diagnostic and robust Gaussian process) for the generated noise updates are provided, linking them to data influence and the robustness of the Gaussian process.

**Compliance With Llm Reviewing Policy:**

Affirmed.

**Final Justification:**

See my acknowledgments, where I modified the score to weak acceptance.

**Key Questions For Authors:**

1. Does the conclusion regarding Student-t marginals described in Section 3.1 (from Tipping, 2001) hold true for the Gaussian marginals in the manuscript?

2. Why does the RVM in Section 5.3 use l2-IRLS instead of other algorithms (such as EM, MacKay, l1-IRLS, etc.)?

3. Why does the ARD in Section 5.4 use EM instead of other algorithms (such as MacKay, l2-IRLS, l1-IRLS, etc.)?

**Limitations:**

No, the new algorithms proposed in this manuscript merely extend i.i.d. noise to heteroscedastic noise, and they fall short in terms of innovation and contribution.

**Strengths And Weaknesses:**

• Soundness: The authors use different methods to update the model and the noise parameters. Experiments are conducted to verify the robustness, model fit, and model and data sparsity.

• Presentation: The submission is clearly written and well structured. The overall narrative is easy to follow.

• Significance: The work introduces heteroscedastic noise to account for model and data sparsity simultaneously.

• Originality: By introducing heteroscedastic noise to the model, the work provides two interpretations of the noise.

---

> ### Author Rebuttal · Authors · 2026-03-31
>
> Dear reviewer aBQz, we thank you for your time and comments and are glad that you found our work clearly written, theoretically sound, and empirically supported. We address your remarks below.
>
> > *merely extend i.i.d. noise to heteroscedastic noise […] fall short in terms of innovation and contribution*
> >
>
> **Comment on Novelty and Contributions**
>
> Our conceptual claim is **not that we propose** heteroscedastic modelling, but **that we show how** this can be operationalized by symmetric treatment of both model and noise parameters within a single, closed-form ARD objective; as reviewer PW2g states this forms *“an elegant conceptual contribution”*. Our idea naturally embeds robustness and sparsity in the ARD framework, does **not require any additional structural assumptions**, and holds for any feature basis. This is in contrast to prior works that require separate noise models which complicate the design (e.g., placing a GP on the noise [GP’17]), limit themselves to the RVM model (e.g., exploiting the RVM’s per-sample kernel basis [MK’10]), or need additional conditions for closed-form updates [FA’01]. We are happy to more thoroughly detail this distinction based on our discussion in §2.
>
> Technically, it is non-trivial to derive heteroscedastic update rules **across all major SBL optimization procedures** (EM, MacKay, L2-IRLS, L1-IRLS). App. B showcases a variety of mathematical tools are needed, from matrix algebra to convex conjugates. Our recovery of a canonical variance update does not stop there, but is **motivated and interpreted in two novel ways** in §4 (via influence and as a robust GP) that connect related but nonetheless disparate fields. As reviewer PW2g puts it, such results are *“insightful and well-developed […] in a non-trivial way”*.  This is also in clear contrast to other recent works on ARD/SBL, which operate on sparsity only and offer more narrow tailored solutions, e.g. [ZH’25] on a particular extension for correlated outputs or [WY’24] for a min-min relaxation result. We also take **several steps towards modern applicability** by showcasing neural network use, gradient-based updates, and an outlook on variational approximation via the Bayesian Learning Rule, including derivations in App. §B.8. Such clear efforts towards future development are generally not found in prior work.
>
> Thus, to the best of our knowledge, this is the first work that *(i)* places per-sample variances on equal footing with ARD weight precisions under a unified evidence objective, and *(ii)* validates the resulting joint mechanism across a broad set of standard SBL solvers and modern model instantiations, offering momentum for several future extensions.
>
> If the reviewer can detail more precisely why they believe our work to “*fall short*”, we will gladly address those particular concerns.
>
> > *Does the conclusion regarding Student-t marginals described in Section 3.1 […] hold true?*
> >
>
> Yes, under the Gaussian–Gamma hierarchy for weights, integrating out the precision induces a Student-t ‘marginal prior’ (with heavy tails and peak at zero), matching the standard ARD sparsity intuition (see [TM’01, §5.1]). Analogously, a Gaussian noise model with an inverse-Gamma variance hyperprior yields a Student-t marginal over noise, motivating robustification. The statement serves mostly as a theoretical motivator for these ARD effects enacted by its hierarchical formulation with hyperpriors.
>
> > *Why does the RVM in Section 5.3 use l2-IRLS? […] Why does the ARD in Section 5.4 use EM?*
> >
>
> In §5.3 we chose L2-IRLS mainly to avoid visual clutter in Fig. 4 and because it performs well empirically. For completeness, we provide the full Boston kernel tables from §5.3 via an anonymized link below. Across closed-form solvers, results are broadly comparable (esp. NLL). See also App. Tables 7, 8 for another comparison. In §5.4 we chose EM for computational convenience due to typically fast convergence. App. Fig. 7 also provides results for OLS, Ridge and MacKay.
>
> Anonymized Link: [https://limewire.com/d/BftRM#kRAug8D1Es](https://limewire.com/d/QavkT#fYjRqbgHXj)
>
> We thank you again and appreciate reconsideration of your score if your concerns have been addressed sufficiently.
>
> **References**
>
> [GP’17] Gerstoft, P. et al. Sparse bayesian learning for DOA estimation in heteroscedastic noise. *arXiv Preprint* (2017).
>
> [MK’10] Mitra, K. et al. Robust RVM regression using sparse outlier model. *CVPR* (2010).
>
> [FA’01] Faul, A. et al. A variational approach to robust regression. *ICANN* (2001).
>
> [ZH’25] Zhang, H. et al. Efficient Network Automatic Relevance Determination. *ICML* (2025).
>
> [WY’24] Wang, Y. et al. An Iterative Min-Min Optimization Method for Sparse Bayesian Learning. *ICML* (2024).
>
> [TM’01] Tipping, M. Sparse Bayesian learning and the relevance vector machine. *JMLR* (2001).

---

> > ### Author Rebuttal · Reviewer_aBQz · 2026-04-03
> >
> > Thanks to the authors for their detailed responses. Both model sparsification and data sparsification have been studied independently in the literature. This work seems to merely combine the ideas within a single objective, and uses five typical algorithms to solve it. This raises a concern for me regarding whether the contribution is sufficient, since both components and the optimization strategies are well-established. But the paper is well-structured and clearly written, with extensive experiments and visually appealing figures. So I decide to raise the score.

---

> > > ### Author Response · Authors · 2026-04-04
> > >
> > > Thank you for the exchange and we appreciate you raising your score. We recognize the remaining concern about how our contribution should be positioned relative to prior works that study model or data sparsification separately. We will sharpen our contribution statement and stress distinctions to existing literature more explicitly. Our intent is to provide a principled yet practical ARD formulation that combines robustness and sparsity within Sparse Bayesian Learning, and we believe this unification is a meaningful advance. We appreciate your positive assessment of the paper’s structure and experiments.

---

### Decision · Program_Chairs · 2026-04-30

**Decision:**

Accept (regular)

**Comment:**

All reviewers agree that the paper is technically sound, clearly written, and well-motivated, with a clean formulation that extends ARD to jointly capture feature and sample relevance through a unified marginal likelihood objective. They highlight the conceptual appeal of treating robustness as sparsification, the preservation of conjugacy with closed-form updates across multiple SBL solvers, and the solid empirical evaluation. The rebuttal helped clarify several concerns and resolved some points of confusion.

While reviewers noted that the contribution is somewhat incremental and raised limitations (e.g., lack of deeper analysis and restriction to linear-in-weights models), there is overall agreement that the paper provides a useful and principled extension of existing methods.